# FROM IMAGES TO SIGNALS: ARE LARGE VISION MODELS USEFUL FOR TIME SERIES ANALYSIS?

## ABSTRACT

Large Vision Models (LVMs) are emerging tools for transferring cross-modal knowledge to time series, but this potential is not well understood. This work addresses the gap by investigating LVMs for both high-level (classification) and low-level (forecasting) tasks. Our aim is to not only assess whether LVMs can succeed, but also reveal why they succeed or fall short. Through a comparative benchmark covering 4 LVMs, 8 imaging methods, 18 datasets, and 26 baselines, we identify the strengths and limitations of LVMs, as well as strategies for adapting them to time series modeling. Our findings indicate while LVMs are effective for time series classification, they face notable challenges in forecasting — the best LVM forecaster is limited to specific model types and imaging methods, exhibit biases toward forecasting periods, and struggle to leverage long look-back windows. We hope our findings can serve as both a cornerstone and a practical guide for advancing LVM- and multimodal-based solutions to different time series tasks.

## 1 INTRODUCTION

Time series analysis underpins applications in geoscience (Ardid et al., 2025), neuroscience (Caro et al., 2024), energy (Koprinska et al., 2018), healthcare (Morid et al., 2023), and smart city (Ma et al., 2017). Inspired by advances in sequence modeling for language, recent research has explored methods from Transformer (Wen et al., 2023) to Large Language Models (LLMs) (Jiang et al., 2024; Zhang et al., 2024) for time series. With the success of Large Vision Models (LVMs) such as `ViT` (Dosovitskiy et al., 2021), `BEiT` (Bao et al., 2022), and `MAE` (He et al., 2022), emerging work has begun to investigate their potential in this domain (Chen et al., 2025). In these approaches, time series are *imaged*, *i.e.*, transformed to certain image representations (*e.g.*, Fig. 1(a)) (Ni et al., 2025), then fed to an LVM to learn embeddings that can be probed for downstream tasks. The motivation of adapting LVMs, being pre-trained on vast images, to time series rests on two perspectives: (1) for *high-level* (*i.e.*, semantic level) tasks like classification, imaged time series can encode distinguishable temporal patterns as semantic cues recognizable by LVMs; and (2) for *low-level* (*i.e.*, numerical level) tasks like forecasting, the structural similarity between images and time series — where rows or columns of *continuous* pixels in an image resemble a univariate time series (UTS) — makes LVMs more naturally suited than LLMs, which operate on *discrete* tokens. Despite this promise, the deeper connections between LVMs and time series analysis remain largely underexplored.

To understand LVMs' role in time series tasks and inform future research — including multimodal models that integrate imaged time series (Zhong et al., 2025) — a thorough benchmark study is desired. To this end, we investigate LVMs on two representative tasks, time series classification (TSC) and time series forecasting (TSF). In a nutshell, our conclusion is: **pre-trained LVMs prove versatile for TSC — task relying on pattern comparison, but are constrained under TSF — task requiring numerical inference**. The current best LVM-based forecasters, although effective, remain confined to specific types of LVMs and imaging methods, exhibit biases toward forecasting periods, and struggle with long look-back windows. We envision our conclusion benefiting other high-level tasks (*e.g.*, retrieval, clustering) and low-level tasks (*e.g.*, imputation, anomaly detection). Unlike prior works that question the adoption of Transformer (Zeng et al., 2023a) and LLMs (Tan et al., 2024) in this field, we take a cautiously optimistic view, aiming to provide novel insights and caveats for selecting and adapting LVMs for the right time series tasks, so as to support future developments.

This work involves two LVMs that are supervisedly pre-trained, *i.e.*, `ViT` (Dosovitskiy et al., 2021) and `Swin` (Liu et al., 2021), and two LVMs that are self-supervisedly pre-trained, *i.e.*, `MAE` (He et al.,

2022) and `SimMIM` (Xie et al., 2022), along with 8 widely used methods for imaging time series (Ni et al., 2025). The selected 4 LVMs cover key properties such as different pre-training strategies and attention mechanisms (detailed in §4.1), underlying newer LVMs like `ViT-22B` (Dehghani et al., 2023), `DINOv2` (Oquab et al., 2024), and `VIS-MAE` (Liu et al., 2024). Another LVM, `LaVin-DiT` (Wang et al., 2025b), is also tested, but shows similar performance to the selected LVMs, thus is deferred to Appendix B.13. Our analysis involves 10 datasets for TSC and 8 datasets for TSF, all are widely used benchmarks (Bagnall et al., 2018; Wu et al., 2023; Tan et al., 2024). The results provide an overview on the effectiveness of LVMs, shedding light on what type of LVMs (*supervised vs. self-supervised*), which imaging method (*among 8 methods*), and what output design (*linear probing vs. pre-trained decoder*) fit which task (*classification vs. forecasting*).

To uncover LVMs' true potential, in-depth ablations are conducted. We compare their zero-shot and (fully/partially) fine-tuned performance with the same architectures trained from scratch, identifying the best adaptation strategy for TSC and TSF tasks, respectively. Testing with shuffled time steps shows that LVMs capture temporal modeling. As we observe TSF is more challenging than TSC to LVMs, further TSF-specific study is conducted, which reveals the best LVM forecaster is confined to a combination of a self-supervised LVM and a specific imaging method (*i.e.*, UVH in Fig. 1(a)). Intriguingly, we find that pre-trained decoders contribute more than encoders in forecasting, explaining the challenge for supervised LVMs. However, current best LVM forecasters carry an inductive bias: they tend to "combine past periods" as forecasts, making them prone to datasets with strong periodicity, highlighting an area to improve in the future. To sum up, our contributions are as follows:

- To the best of our knowledge, this is the first benchmark to comprehensively fine-tune and compare representative LVMs with time series models for both high-level and low-level tasks (§4.1).
- We summarize the current best ways to tweak LVMs (§3) and conduct a series of ablation analysis to assess whether LVMs are truly useful for TSC and TSF tasks, covering various aspects of the adapted LVMs, including their effectiveness in terms of pre-training, imaging, decoding, fine-tuning, architecture, temporal modeling, and computational costs (§4.2).
- We further investigate the challenge of using LVMs for forecasting by examining individual model components, potential inductive bias, and the impact of look-back windows (§4.3).

## 2 RELATED WORK

Our work share similar merits as (Zeng et al., 2023a; Tan et al., 2024; Zhou & Yu, 2025), each of which sheds important lights on a single time series task, *i.e.*, Transformers for TSF (Zeng et al., 2023a), LLMs for TSF (Tan et al., 2024), and LLMs for time series anomaly detection (TSAD) (Zhou & Yu, 2025). In contrast, our work is LVM-specific, covering more tasks with in-depth analysis. This work could be considered as a substantial complement to the prior works by adding a new lens to our understanding of large models' roles in the contemporary time series domain.

Vision models have been used for a variety of time series tasks, including classification (Li et al., 2023; Wu et al., 2023), forecasting (Zeng et al., 2023b; Yang et al., 2024), anomaly detection (Zhang et al., 2019; Wu et al., 2023), and generation (Li et al., 2022; Karami et al., 2024). Our work focuses on the recent development of using **pre-trained LVMs**, particularly Transformer-based models, for time series analysis. Image-pretrained CNNs have also been investigated in the past (Namura et al., 2024; Li et al., 2020), but are out of our scope due to their relatively smaller sizes. To apply LVMs to time series, the existing works typically transform time series to images by an imaging method (Ni et al., 2025). For example, `AST` (Gong et al., 2021) applies ImageNet-pretrained `DeiT` (Touvron et al., 2021) on filterbank spectrograms of audio signals, which are basically UTS, for TSC. `ViTST` (Li et al., 2023) uses pre-trained `Swin` (Liu et al., 2021) for classifying lineplots of time series. These works have inspired a series of efforts in pre-training `ViT` architectures with imaged time series data, such as `SSAST` on AudioSet-2M (Gong et al., 2022), `ViTime` on synthetic data (Yang et al., 2024), and `Brain-JEPA` on brain time series (Dong et al., 2024). In contrast to TSC, TSF task has less efforts in using LVMs, possibly because LVMs are less adept at low-level tasks than high-level tasks. The most salient method is `VisionTS` (Chen et al., 2025), which adapts a self-supervisedly pre-trained LVM *i.e.*, `MAE` (He et al., 2022), to zero-shot and few-shot TSF. In our work, in addition to `MAE`, we include another self-supervised LVM – `SimMIM` (Xie et al., 2022).

Recently, vision-language models (VLMs), such as `LLaVA` (Liu et al., 2023a), `CLIP` (Radford et al., 2021), `ViLT` (Kim et al., 2021), *etc.*, which involve pre-trained vision encoders, have been explored

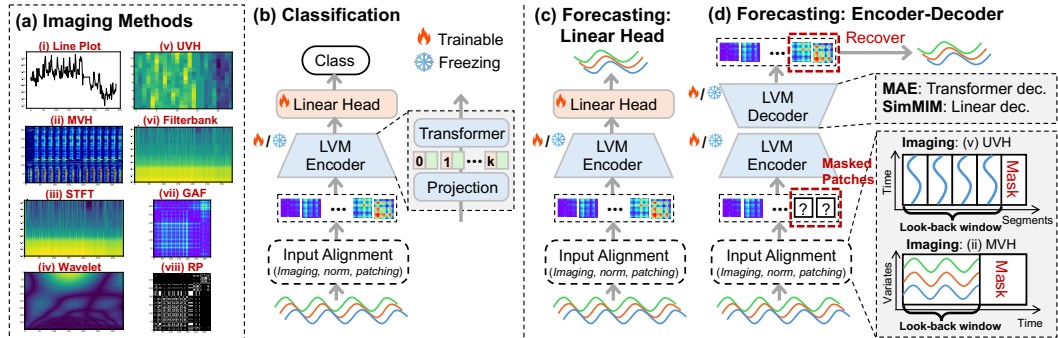

Figure 1: An overview of (a) different imaging methods, (b) LVM-based time series classification, (c) LVM with linear head for forecasting, (d) LVM encoder-decoder for forecasting. In (a), MVH encodes MTS, others encode UTS. (b)(c) apply to all LVMs (ViT, Swin, MAE, SimMIM) in this study. (d) applies to MAE and SimMIM with UVH/MVH images. Table 1 summarizes their applicability.

for TSC (Wimmer & Rekabsaz, 2023; Prithyani et al., 2024), TSAD (Zhuang et al., 2024), and TSF (Zhong et al., 2025). However, the effectiveness of sole LVMs in time series analysis has not yet been well understood. As such, we focus on LVMs in this work, and leave VLMs for future work. More discussions are in Appendix E.

## 3 METHODS FOR USING LVMS IN TIME SERIES ANALYSIS

In this work, we assess LVMs' innate ability in time series analysis by keeping the main architecture intact but making a few necessary tweaks for **cross-modal adaptation**. Additionally, we introduce two ablations that will be used in §4 to evaluate whether LVMs' architecture is over-complex.

**Input Alignment.** The input to a pre-trained LVM should be a normalized 3-channel image of a predefined size. Fitting time series to LVMs' input requires (1) imaging time series; (2) resizing the imaged time series to fit the channel/size requirement; and (3) normalizing the image.

For (1), we employ 8 imaging methods outlined by (Ni et al., 2025). As Fig. 1(a) illustrates, **Line Plot** draws a 2D image with $x$-axis representing time steps and $y$-axis representing time-wise values. **UVH** (Univariate Heatmap) divides a UTS, $\mathbf{x} \in \mathbb{R}^T$, into $\lfloor T/L \rfloor$ segments of length $L$ — a period obtained using Fast Fourier Transform (FFT) on $\mathbf{x}$ — and stack them to a 2D image of size $L \times \lfloor T/L \rfloor$. **MVH** (Multivariate Heatmap) visualizes the matrix of a multivariate time series (MTS), $\mathbf{X} \in \mathbb{R}^{d \times T}$, with $x$-axis representing $T$ time steps and $y$-axis representing $d$ variates. **STFT** (Short-Time Fourier Transform), **Wavelet** (Wavelet Transform) and **Filterbank** are three methods for transforming $\mathbf{x}$ to a **Spectrogram** with $x$-axis as time and $y$-axis as frequency/scale. **GAF** (Gramian Angular Field) and **RP** (Recurrence Plot) produce square matrices with both $x$- and $y$-axis representing time, but they encode different temporal patterns. For brevity, we refer readers to (Ni et al., 2025) and our summary in Appendix F.1 and F.2 for a more detailed introduction about the 8 imaging methods.

For (2), *i.e.*, image resizing, following (Gong et al., 2021; Chen et al., 2025), we first resize an imaged time series to fit the size defined by LVMs' pre-training data using bilinear interpolation. Then, we align the resized images to meet the 3-channel requirement by duplicating each resized image (per variate) three times to form a gray image. For (3), *i.e.*, image normalization, since the adopted LVMs, *i.e.*, ViT, Swin, MAE, SimMIM, standardize each pre-training image, we normalize each imaged time series in the same manner for consistency: $\mathbf{I}_{\text{norm}} = [\mathbf{I} - \text{mean}(\mathbf{I})]/\text{standard-deviation}(\mathbf{I})$, where $\mathbf{I}$ is the input image and $\mathbf{I}_{\text{norm}}$ is the normalized one. As shown in Fig. 1(b)-(d), the normalized image is then divided into a number of patches as specified by each LVM before feeding to the LVM.

**Task-Specific Augmentation.** For **TSC task**, as shown in Fig. 1(b), we linearly probe each LVM's encoder. For ViT and Swin, this implies replacing their classification layers by a new linear layer tailored to a specific downstream TSC task. For MAE and SimMIM, this means their reconstruction decoders are replaced by a linear classification layer. As most imaging methods encode UTS (except for MVH), the image of each variate is fed to the LVM individually. The output patch embeddings of all variates are concatenated before delivering to the last linear layer. For MVH, there is a single image of all variates, thus it does not need variate-concatenation.

For **TSF task**, we employ two frameworks from the literature. Fig. 1(c) trains a linear forecaster (Zeng et al., 2023b; Yang et al., 2024), Fig. 1(d) uses LVMs' reconstruction decoders for forecasting (Chen et al., 2025). Because only MAE and SimMIM in our study have such decoders, Fig. 1(d) is applied to them. Fig. 1(c) applies to ViT and Swin. For both frame-

| Task | Imaging | ViT | Swin | MAE | SimMIM |
|---|---|---|---|---|---|
| Classification | All | (b) | (b) | (b) | (b) |
| Forecasting | UVH,MVH | (c) | (c) | (d) | (d) |
| Forecasting | Other | (c) | (c) | (c) | (c) |

Table 1: LVM framework summary. (b)(c)(d) indicates the frameworks in Fig. 1.

works, we adopt the "variate-independence" assumption that is widely used in TSF (Nie et al., 2023), *i.e.*, each variate is forecasted independently. This applies to all imaging methods except for MVH, by which all variates appear in the same image thus are forecasted once. Additionally, the framework in Fig. 1(d) adds a mask subsequent to the look-back window part in the image, then it reconstructs the masked patches and recovers forecasts from the reconstructed patches. This requires input images to preserve raw time series values in pixels. Among the 8 imaging methods, only MVH and UVH preserve time series values. Thus, this framework is applied to MVH and UVH[1]. The framework in Fig. 1(c) can be applied to all imaging types. Table 1 summarizes how frameworks (b)(c)(d) in Fig. 1 apply to different LVMs.

**Ablations.** To assess whether LVM architecture is over-complex, we add two ablation models. Both models keep the projection layer in LVM encoder, but replace the Transformer by a simpler layer. The first ablation replaces the Transformer by a linear layer, named as W/O-LVM. The second ablation uses a single randomly initialized multi-head attention layer, named as LVM2ATTN. Both ablations use a linear head to avoid complex decoders. They are applicable to all 8 imaging types and both of the two tasks. An illustration of the ablation models can be found in Appendix B.6.

## 4 EXPERIMENTS

**Datasets.** Our experiments are conducted on widely used benchmarks. For TSC, following (Wu et al., 2023; Zhou et al., 2023), we use 10 datasets from UEA Archive (Bagnall et al., 2018), covering gesture/action/audio recognition, heartbeat-based diagnosis, and other real-world tasks. For TSF, we use 8 datasets including ETT (Electricity Transformer Temperature) (Zhou et al., 2021), encompassing ETTh1, ETTh2, ETTm1, ETTm2, Weather (Wu et al., 2021), Illiness (Wu et al., 2021), Traffic (Wu et al., 2021), and Electricity (Trindade, 2015). For both tasks, all of the time series are MTS. We defer detailed data descriptions to Appendix A.1.

**Evaluation Metrics.** For TSC, following (Wu et al., 2023; Zhou et al., 2023), we report classification accuracy of the compared methods. For TSF, following (Nie et al., 2023; Zeng et al., 2023a; Tan et al., 2024), mean squared error (MSE) and mean absolute error (MAE) are used to evaluate performance. Definitions of the metrics are deferred to Appendix A.3.

**Models.** We base our experiments on representative LVMs, including two supervised LVMs: (1) ViT (Dosovitskiy et al., 2021), (2) Swin (Liu et al., 2021), and two self-supervised LVMs: (3) MAE (He et al., 2022), (4) SimMIM (Xie et al., 2022). They are implemented as per Table 1 for different tasks. Following (Wu et al., 2023; Zhou et al., 2023), we include 18 classification baselines ranging from XGBoost to LLMs. Moreover, 8 SOTA forecasting baselines are compared. Since we aim to assess LVMs' *cross-modal knowledge*, the baselines focus on models without pre-trained knowledge and models with other cross-modal knowledge (*e.g.*, LLMs). The baseline methods are presented in Fig. 4 and Table 2, and described in Appendix A.2. The implementation details of the LVMs, including checkpoint selection, hyperparameters, and running environments are described in Appendix A.4.

**Structure of Experiments.** Next, through a series of research questions (RQs), we will assess (i) whether LVMs are effective in TSC and TSF tasks (RQ1 - RQ3, §4.1); (ii) whether the best-performing LVMs (as identified in (i)) are truely useful (RQ4 - RQ7, §4.2); and (iii) why TSF poses more challenges to LVMs (RQ8 - RQ10, §4.3). RQ3 uses all of the datasets for an overall comparison, while other analyses are based on 4 UEA classification datasets (FaceDetection, Handwriting, SpokenArabicDigits, and UWaveGestureLibrary) and 4 forecasting datasets (ETTh1, ETTm1, Weather, and Illiness) for conciseness. It is noteworthy that (i) is not a simple reflection of existing works since none of the existing works formally compares fine-tuned LVMs across the two tasks. As another contribution, the RQs also provide a guide on how to configure LVMs for the right task.

---

[1]GAF can be applied as it has an inverse function, but is largely limited for reasons described in Appendix B.2.

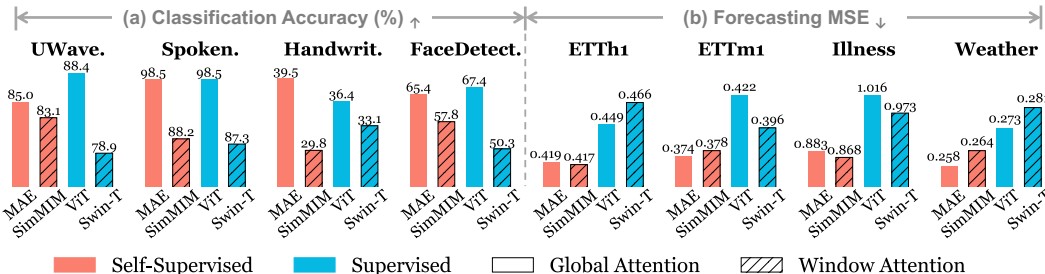

Figure 2: Comparison of 4 LVMs on TSC (accuracy) and TSF (MSE). ↑ (↓) indicates a higher (lower) value is better. Two taxonomies of the LVMs: (1) supervised (ViT, Swin) *vs.* self-supervised (MAE, SimMIM), (2) using global attention (ViT, MAE) *vs.* window-based attention (Swin, SimMIM).

### 4.1 WHETHER LVMS ARE EFFECTIVE IN TIME SERIES TASKS?

This section first determines the best combination of LVMs and imaging methods in TSC and TSF tasks (RQ1 and RQ2), then compares the best settings with SOTA non-LVM baselines to assess an overall effectiveness (RQ3).

*[RQ1] What type of LVM best fits TSC (TSF) task?* Fig. 2 compares the 4 LVMs in TSC and TSF tasks. From Fig. 2, we observe (1) **supervised LVMs and self-supervised LVMs show comparable accuracies in classification**, while (2) **self-supervised LVMs are remarkably better at forecasting than supervised LVMs**. (1) is consistent with the comparable performance of the two kinds of LVMs in classifying images (He et al., 2022). (2) attributes to the continuous nature of pixels and time series, which enables self-supervised LVMs to transfer their ability in reconstructing masked pixels to predict (masked) time series, as proposed by (Chen et al., 2025). Moreover, in Fig. 2(a), we observe SimMIM and Swin underperform (SimMIM uses Swin backbone). This is because they use *window-based local attention* mechanism. Compared to the *global attention* used by MAE and ViT, local attention implicitly assumes *translation invariance* – a model's ability to recognize an object in an image regardless of its location (Lenc & Vedaldi, 2015). This assumption, however, does not hold in imaged time series since different locations correspond to different time-steps/frequencies. A pattern that appears at different time steps may lead to different classes. By overlooking spatial differences, SimMIM and Swin fail to identify some time/frequency-sensitive patterns.

*[RQ2] Which imaging method best fits TSC (TSF) task?*

Fig. 3 presents the critical difference (CD) diagrams (Han et al., 2022) on the average rank of the 8 imaging methods on TSC and TSF tasks (lower rank is better). From Fig. 3(a), **GAF fits the classification best**, with close performance to MVH and RP, indicating their abilities in encoding distinguishable semantic patterns. Line Plot remarkably un-

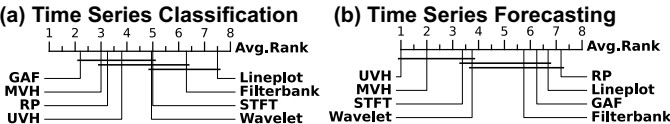

Figure 3: Average rank of different imaging methods in (a) TSC task, and (b) TSF task. Lower rank is better. The detailed results can be found in Appendix B.2.

derperforms, thus may not fit this task. **For forecasting**, from Fig. 3(b), when used in conjunction with the reconstruction framework in Fig. 1(d), **UVH demonstrates best performance**, followed by MVH, suggesting their suitability in numerical level tasks by leveraging LVMs' knowledge acquired from reconstructing masked pixels during pre-training (further analysis in Appendix B.2).

*[RQ3] Can LVMs achieve SOTA performance?* Fig. 4 and Table 2 present the overall performance of the compared methods, where ViT and MAE are selected for their best performance as a supervised LVM and a self-supervised LVM, respectively. Here, ViT and MAE are set up with their best imaging methods — GAF for TSC and UVH for TSF. On average, LVMs were fine-tuned on each dataset with 20 epochs for TSC and 8 epochs for TSF upon early stopping. Our experiments follow standard protocols of TSC (Zhou et al., 2023) and TSF (Tan et al., 2024).

| Method | MAE | | ViT | | Time-LLM | | GPT4TS | | CALF | | Dlinear | | PatchTST | | TimesNet | | FEDformer | | Autoformer | |
|---|---|---|---|---|---|---|---|---|---|---|---|---|---|---|---|---|---|---|---|---|
| Metrics | MSE | MAE | MSE | MAE | MSE | MAE | MSE | MAE | MSE | MAE | MSE | MAE | MSE | MAE | MSE | MAE | MSE | MAE | MSE | MAE |
| ETTh1 | 0.409 | 0.419 | 0.445 | 0.449 | 0.418 | 0.432 | 0.418 | 0.421 | 0.432 | 0.431 | 0.423 | 0.437 | 0.413 | 0.431 | 0.458 | 0.450 | 0.440 | 0.460 | 0.496 | 0.487 |
| ETTh2 | 0.357 | 0.390 | 0.389 | 0.411 | 0.361 | 0.396 | 0.354 | 0.389 | 0.351 | 0.384 | 0.431 | 0.447 | 0.330 | 0.379 | 0.414 | 0.427 | 0.437 | 0.449 | 0.450 | 0.459 |
| ETTm1 | 0.345 | 0.374 | 0.409 | 0.422 | 0.356 | 0.377 | 0.363 | 0.378 | 0.396 | 0.391 | 0.357 | 0.379 | 0.351 | 0.381 | 0.400 | 0.406 | 0.448 | 0.452 | 0.588 | 0.517 |
| ETTm2 | 0.268 | 0.327 | 0.300 | 0.337 | 0.261 | 0.316 | 0.254 | 0.311 | 0.283 | 0.323 | 0.267 | 0.334 | 0.255 | 0.315 | 0.291 | 0.333 | 0.305 | 0.349 | 0.327 | 0.371 |
| Weather | 0.225 | 0.258 | 0.234 | 0.273 | 0.244 | 0.270 | 0.227 | 0.255 | 0.251 | 0.274 | 0.240 | 0.300 | 0.226 | 0.264 | 0.259 | 0.287 | 0.309 | 0.360 | 0.338 | 0.382 |
| Illness | 1.837 | 0.883 | 2.179 | 1.016 | 2.018 | 0.894 | 1.871 | 0.852 | 1.700 | 0.869 | 2.169 | 1.041 | 1.443 | 0.798 | 2.139 | 0.931 | 2.847 | 1.144 | 3.006 | 1.161 |
| Traffic | 0.386 | 0.256 | 0.430 | 0.343 | 0.422 | 0.281 | 0.421 | 0.274 | 0.444 | 0.284 | 0.434 | 0.295 | 0.391 | 0.264 | 0.620 | 0.336 | 0.610 | 0.376 | 0.628 | 0.379 |
| Electricity | 0.159 | 0.250 | 0.173 | 0.266 | 0.165 | 0.259 | 0.170 | 0.263 | 0.176 | 0.266 | 0.166 | 0.264 | 0.162 | 0.253 | 0.193 | 0.295 | 0.214 | 0.327 | 0.227 | 0.338 |
| # Wins | 9 | | 0 | | 0 | | 3 | | 0 | | 0 | | 4 | | 0 | | 0 | | 0 | |

Table 2: Model comparison in TSF. The results are averaged over different prediction lengths. See Table 15 in Appendix B.4 for full results. Red and Blue numbers are the the best and second best results. # Wins is the number of times the method performed best.

From Fig. 4, both `ViT` and `MAE` outperform the baselines, suggesting **both supervised and self-supervised LVMs appear effective in high-level (*i.e.*, semantic level) TSC task**. This is consistent with their ability in classifying regular images (He et al., 2022). From Table 2, across the 8 datasets and 2 metrics, `MAE` outperforms non-LVM baselines in 9/16 cases, while `ViT` doesn't show evident superiority over non-LVM baselines, which may be caused by its classification-based pre-training. The results suggest **LVMs' distinct abilities in TSF, conveying that more challenges may appear in low-level (*i.e.*, numerical level) tasks**. Taking a closer look at Table 2, despite `ViT`'s inferior performance, it is comparable to `DLinear` in many cases. It implies that although `ViT` is pre-trained for image classification, linearly probing it is adequate to produce reasonable forecasting results, showing a potential in cross-task/modality knowledge transfer.

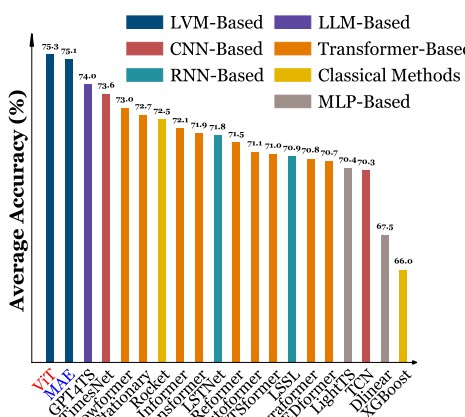

Figure 4: Model comparison in TSC. The results are averaged over 10 UEA datasets. See Table 13 in Appendix B.3 for full results.

## 4.2 ARE LVMs TRUELY USEFUL FOR TIME SERIES TASKS?

This section aims to uncover whether LVMs' effectiveness in Fig. 4 and Table 2 truly come from their pre-trained parameters, architectures, and temporal modeling ability. Unless otherwise noted, the best-performing LVM is used for TSC, *i.e.*, `ViT` with GAF imaging (*ref.* Fig. 4), and TSF, *i.e.*, `MAE` with UVH imaging (*ref.* Table 2), respectively.

***[RQ4] Are the pre-trained parameters in LVMs useful in time series tasks?*** To answer this question, we compare three kinds of ablations: (1) training LVMs from scratch; (2) freezing LVM's parameters; and (3) fine-tuning LVMs with a few epochs. Since different tasks may need different fine-tuning strategies, we include a series of fine-tuning ablations that progressively freeze the key components in the Transformer block of LVMs. Fig. 5 shows the key components. To sum up, our ablations in this study include (a) fine-tune all parameters; (b) fine-tune all parameters but freeze `CLS` token and `Mask` token; (c) fine-tune MLP and norm layers only; (d) fine-tune norm layer only; (e) freeze all parameters; and (f) randomly initialize an LVM and train it from scratch.

Table 3 (upper panel) summarizes the results. For TSC, we observe that "freeze all" is better than "train from scratch" in all cases, suggesting **LVMs indeed transfer useful knowledge**. Fine-tuning all parameters with a few epochs always improves over "freeze all" cases, further validating effective knowledge transfer. Comparing (a) and (c), **fine-tuning MLP & norm layers is the minimal fine-tuning effort for achieving "full fine-tuning"-like performance in TSC.** For TSF, surprisingly, neither of "freeze-all" case nor fine-tuning all parameters consistently outperforms training from scratch, while **only fine-tuning the norm layer significantly boosts the performance in TSF**. This may be caused by the low-level nature of the forecasting task. The model needs to predict numerical values, which is largely influenced by normalization, while fine-tuning more than necessary may lead

| Task | | TSC Task (accuracy (%)$_\uparrow$) | | | | TSF Task (MSE$_\downarrow$) | | | |
|---|---|---|---|---|---|---|---|---|---|
| Dataset | | UWave. | Spoken. | Handwrit. | FaceDetect. | ETTh1 | ETTm1 | Illiness | Weather |
| RQ4 | (a) All parameters | 88.4 | **98.5** | **36.4** | **67.4** | 0.558 | 0.399 | 1.781 | 0.273 |
| | (b) All but `CLS` & `Mask` | 87.5 | 98.2 | 35.2 | 66.3 | 0.530 | 0.408 | 1.783 | 0.275 |
| | (c) MLP & norm | **88.7** | 98.4 | 35.5 | 67.1 | 0.532 | 0.396 | 1.737 | 0.264 |
| | (d) Norm | 81.6 | 98.0 | 28.5 | 65.2 | **0.409** | **0.345** | 1.837 | **0.225** |
| | (e) Freeze (zero-shot in TSF) | 84.0 | **98.5** | 27.8 | 66.7 | 0.452 | 0.420 | 2.037 | 0.308 |
| | (f) Train from scratch | 73.4 | 97.0 | 24.3 | 65.0 | 0.475 | 0.372 | **1.723** | 0.241 |
| RQ5 | w/o-LVM | 78.6 | 96.4 | 22.4 | 64.1 | 0.423 | 0.376 | 2.291 | 0.255 |
| | LVM2ATTN | 80.1 | 96.5 | 20.7 | 66.2 | 0.428 | 0.357 | 2.108 | 0.254 |

Table 3: Ablation analysis of LVMs. For classification, higher accuracy indicates better performance. For forecasting, lower MSE is preferred. Full results are in Appendices B.5 and B.6.

| Task | | Classification | | | | Forecasting | | | |
|---|---|---|---|---|---|---|---|---|---|
| Dataset | | UWave. | Spoken. | Handwrit. | FaceDetect. | ETTh1 | ETTm1 | Illiness | Weather |
| Sf-All | w/o-LVM | 78.2% | 49.7% | 81.7% | 19.3% | 76.2% | 98.4% | 116.4% | 24.1% |
| | LVM2ATTN | 86.4% | 50.6% | 89.9% | 22.4% | 79.7% | 117.1% | 109.1% | 24.4% |
| | LVM | 80.7% | 84.7% | 91.5% | 29.2% | 83.8% | 118.4% | 162.8% | 44.5% |
| Sf-Half | w/o-LVM | 6.6% | 12.4% | 74.6% | 10.8% | 14.4% | 28.3% | 41.6% | 2.4% |
| | LVM2ATTN | 8.7% | 11.6% | 83.6% | 11.3% | 19.5% | 44.8% | 69.3% | 2.4% |
| | LVM | 36.4% | 30.2% | 86.5% | 9.3% | 14.5% | 48.2% | 21.3% | 9.6% |
| Ex-Half | w/o-LVM | 98.8% | 82.2% | 83.5% | 22.8% | 13.0% | 145.3% | 11.0% | 34.0% |
| | LVM2ATTN | 98.9% | 82.3% | 87.0% | 24.6% | 9.1% | 158.3% | 27.9% | 35.5% |
| | LVM | 59.4% | 89.9% | 97.0% | 9.2% | 14.2% | 242.3% | 23.0% | 67.2% |
| Masking | w/o-LVM | -1.0% | 3.1% | 22.3% | -1.2% | 47.3% | 58.5% | 94.1% | 33.4% |
| | LVM2ATTN | 1.0% | 3.6% | 20.3% | 2.7% | 46.0% | 70.3% | 127.8% | 33.6% |
| | LVM | 29.0% | 41.8% | 56.0% | 7.4% | 47.5% | 58.4% | 128.9% | 49.6% |

Table 4: Performance drop of the compared models under different temporal perturbations. Red color marks the largest drop for each perturbation strategy. Full results are in Appendix B.7.

to overfitting. This is in contrast to classification, where the learning of high-level semantic patterns is influenced by more layers than normalization, thus fine-tuning more parameters is beneficial.

**[RQ5] How useful are LVMs' architectures?** In [RQ4], training LVMs from scratch may overfit the small training datasets due to their complex architectures. To examine it, we run the two simpler models introduced in §3, *i.e.*, w/o-LVM and LVM2ATTN. From Table 3 (bottom panel), We observe training from scratch does not consistently outperform the simple models, implying the LVM's architecture itself is over-complex. However, since training from scratch is no worse than the simpler models, the overfitting issue is not serious. Moreover, the "freeze all" and all fine-tuning cases (a)-(d) outperform the simple models in TSC. Fine-tuning case (d) consistently outperforms the simple models in TSF. These results indicate **LVMs' architectures are not over-complex when used as a container of transferrable knowledge from pre-training**.

**[RQ6] Can LVMs capture temporal order of time series?** Temporal order plays a critical role in time series analysis. It is of significant

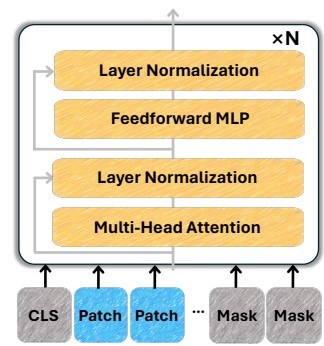

Figure 5: Key components in LVMs' Transformer block.

interests to understand whether LVMs can capture temporal information. To this end, following (Tan et al., 2024), we perturb the temporal order by four methods (1) **Sf-All**: randomly shuffle all of the time points; (2) **Sf-Half**: randomly shuffle the first half of the time points; (3) **Ex-Half**: swap the first and second halves of the time points; and (4) **Masking**: randomly mask 50% time points. Table 4 summarizes the relative performance drop. Following (Zeng et al., 2023a; Tan et al., 2024), simple models are compared for their effectiveness in capturing temporal order. From Table 4, we can see that LVMs always have a performance drop under temporal perturbations. Moreover, they are more vulnerable to temporal perturbations than the ablations. This implies **LVMs are very likely making effective use of temporal patterns in time series during their inferences**.

| Method | | LVM | | | 1st Baseline (task specific) | | | 2nd Baseline (task specific) | | |
|---|---|---|---|---|---|---|---|---|---|---|
| Task | Dataset | # Param (M) | Train (min) | Inference(ms) | # Param (M) | Train (min) | Inference(ms) | # Param (M) | Time (min) | Inference(ms) |
| TSC | UWave. | 89.43 | 2.83 | 11.52 | 82.23 | 1.19 | 57.61 | 2.42 | 0.39 | 1.69 |
| | Handwrit. | 97.59 | 5.18 | 23.72 | 83.62 | 1.33 | 50.51 | 2.47 | 0.51 | 0.78 |
| TSF | ETTh1 | 111.91 | 9.99 | 4.32 | 3.75 | 0.52 | 0.18 | 85.02 | 10.46 | 0.50 |
| | Weather | 111.91 | 207.83 | 1.50 | 6.90 | 16.97 | 0.10 | 86.64 | 94.10 | 0.35 |

Table 5: Computational costs of LVMs and two best baselines in TSC (`GPT4TS`, `TimesNet`) and TSF (`PatchTST`, `GPT4TS`). The forecasting costs are measured with prediction length 96.

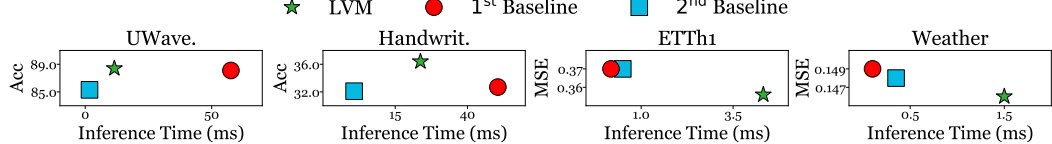

Figure 6: Inference time *vs.* performance of compared methods on TSC (accuracy) using UWaveGesture, SpokenArabicDigits, and TSF (MSE) using ETTh1, Weather. Full results are in Appendix B.8.

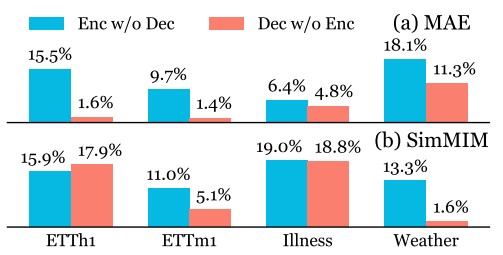

Figure 7: Forecasting performance drop (%) of (a) MAE and (b) SimMIM when only using encoder (blue) and decoder (red).

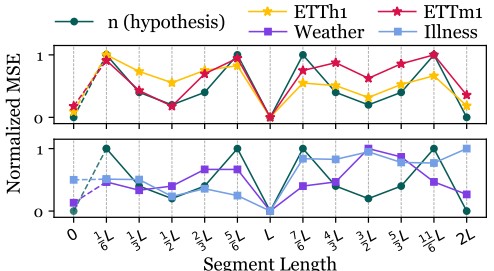

Figure 8: Forecasting performance of MAE *w.r.t.* varying segment length used in UVH imaging. $n$ (green) estimates the difficulty of forecasting.

***[RQ7] What are the computational costs of LVMs?*** We evaluate the training and inference time of LVMs. Training time is measured when a model converges with early stopping. Inference time is estimated by the average runtime per test sample. Table 5 compares LVMs with the best two baselines in TSC (Fig. 4) – `GPT4TS`, `TimesNet`, and TSF (Table 2) – `PatchTST`, `GPT4TS`. From Table 5, LVMs have more parameters than the baselines. On average, LVMs take 3x (16x) training time than the best TSC (TSF) baseline, primarily due to their larger sizes of trainable parameters. For inference, LVMs are 4x faster than the best TSC baseline, but are 20x slower than the best TSF baseline. This is incurred by both the parameter size and the extra costs to imaging time series. Fig. 6 shows inference time *vs.* performance. Compared to the best baselines, **LVMs show reasonable costs in TSC, but trade the computational overhead for better performance in TSF**.

## 4.3 WHY FORECASTING POSES CHALLENGES TO LVMS?

This section aims to reveal the cause of the challenges to LVM forecasters as observed in §4.1.

***[RQ8] Which component of LVMs contributes more to forecasting?*** Usually, pre-trained encoders are considered as general feature extractors and widely used in knowledge transfer, while decoders are task-specific thus are often re-trained in downstream tasks. Yet the conclusion looks counterintuitive when adapting LVMs to TSF. Fig. 7 shows the performance drop of two ablations relative to `MAE` and `SimMIM`: (1) **Enc w/o Dec** fine-tunes the pre-trained encoder but re-trains the decoder; (2) **Dec w/o Enc** fine-tunes the pre-trained decoder but re-trains the encoder. Both ablations are fine-tuned until convergence. From Fig. 7, **Enc w/o Dec** drops more than **Dec w/o Enc**, implying the **pre-trained decoders play more important roles than the encoders in TSF**. This is because LVMs' decoders aim to reconstruct pixel values, thus fitting the low-level TSF task (more analysis in Appendix E). Surprisingly, `SimMIM`'s decoder is merely a linear layer that only occupies 3.8% of all parameters, which however overwhelms its much larger encoder, further underscoring the essential role of LVMs' pre-trained decoders in TSF. This explains **supervised LVMs' difficulty in TSF** (Fig. 2) **as they don't have pre-trained decoders**.

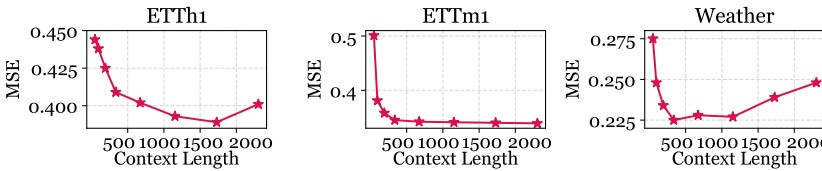

Figure 10: TSF performance (MSE) of MAE with varying look-back window (or context) lengths.

***[RQ9] Will period-based imaging method induce any bias?*** In Table 2, the best LVM forecaster is `MAE` with UVH imaging. As shown in Fig. 1(a)(d), UVH is a period-based imaging method – it stacks length-$L$ segments of a UTS $\mathbf{x}$ into a 2D image of size $L \times \lfloor T/L \rfloor$, where $L$ is a period. **We find this method leads to an inductive bias towards "forecasting periods".** In Fig. 8, we evaluate `MAE`'s forecasting performance by changing the segment length from $\frac{1}{6}L$ to $\frac{12}{6}L$, where the MSE values are min-max normalized to range $[0,1]$. In Fig. 8, an estimated MSE is added at 0 by averaging the MSEs at $L$ and $2L$ since length-0 is not computable. This (and the green lines) will be used later. From Fig. 8, `MAE`'s best performance occurs at $L$ and $2L$, implying (1) the datasets show strong periodicity; and (2) `MAE` tends to infer future by "combining" past segments. When past segments do not coincide with periods, *i.e.*, $\neq L$ or $2L$, `MAE` fails to forecast accurately.

Interestingly, following the UVH imaging method, we can estimate the difficulty of TSF for `MAE` by using the segment length. Basically, the difficulty highly correlates with *how long a segment can reoccur*, measured by the number of segments between the two occurrences (Fig. 9). If the two occurrences are far apart, it is more difficult for `MAE` to capture periodic patterns. More formally, if we divide the UTS into length-$\frac{i}{k}L$ segments, *e.g.*, in Fig. 8, $k = 6$, $i = [1, ..., 12]$, the following Lemma tells how to infer the number of segments before a specific segment reoccurs.

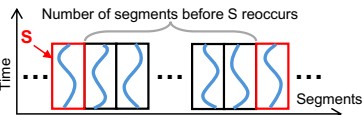

Figure 9: An illustration of UVH.

**Lemma 1.** *Let $\mathbf{x}$ be a UTS with a perfect period $L$, i.e., $\mathbf{x}_t = \mathbf{x}_{t+L}$. If $\mathbf{x}$ is divided into length-$\frac{i}{k}L$ segments, where $i, k \in \mathbb{N}^+$, the smallest number of segments, $n$, before any segment reoccurs, i.e., $\mathbf{x}_t = \mathbf{x}_{t+n\cdot(i/k)L}$, is given by $n = \frac{k}{GCD(i,k)}$, where GCD is the greatest common divisor.*

The proof of Lemma 1 is in Appendix C. Lemma 1 states we can calculate $n$ given $i$ and $k$. To validate the correlation between $n$ and the difficulty of TSF, we calculate $n$ in Fig. 8, and normalize it to range $[0,1]$. $n$ is small when $\frac{i}{k} = 1, 2 \rightarrow n = 1$ or $\frac{i}{k} = \frac{1}{2}, \frac{3}{2} \rightarrow n = 2$, leading to an "M"-shape curve (green). Its coincidence with the MSEs on ETTh1 and ETTm1 datasets validates our estimation of TSF difficulty, implying `MAE` "combines past" to forecast future. In contrast, the MSEs on Weather and Illness datasets align less with the $n$-values, likely due to their weaker periodic patterns. We've evaluated additional datasets with weak periodicity in Appendix B.11.

***[RQ10] Can LVMs make effective use of look-back windows?*** Ideally, longer look-back windows facilitate forecasting (Zeng et al., 2023a). We assess `MAE` with different look-back windows in Fig. 10. The Illness dataset is excluded due to its short time series (966 time steps in total). From Fig. 10, `MAE`'s **performance improves up to a window length of 1000, after which it plateaus or declines**. This may result from image transformation. Fixed-size image in pre-trained LVMs has a pixel limit and may constrain the information captured from excessively long time series (more analysis in Appendix E). Fortunately, contemporary LVMs can handle moderately long windows well (1000 may be considered sufficient in many cases). Future models may extend this capability further.

## 5 CONCLUSION AND FUTURE DIRECTIONS

In this work, we explore the potential of LVMs for time series analysis in both high-level (classification) and low-level (forecasting) tasks. By experiments with various LVMs and ablations, we offer insights into **whether and how image-pretrained LVMs benefit time series tasks**, hopefully helping ease their adoption across research and applications. Our forecasting-specific analysis highlights key limitations of current LVM forecasters, underscoring the need for **improving encoder utilization**, **addressing inductive bias**, **handling longer look-back windows**, and **diversifying benchmark datasets**. We hope this study complements existing research and lays the groundwork for future developments on multi-modal, agentic time series analysis.

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

# A  EXPERIMENTAL SETUP

## A.1  BENCHMARKS

**Time Series Classification**. For TSC, following (Wu et al., 2023; Zhou et al., 2023), our experiments are conducted on 10 multivariate benchmark datasets from UEA archive (Bagnall et al., 2018), which span diverse domains, including chemical analysis, cognitive neuroscience, gesture recognition, biomedical signal processing, speech recognition and traffic analysis. The datasets are preprocessed following (Zerveas et al., 2021). Table 6 summarizes the statistics of the datasets.

| Dataset | Variates | Series Length | Dataset Size | Classes |
|---|---|---|---|---|
| EthanolConcentration | 3 | 1751 | (261, 263, 263) | 4 |
| FaceDetection | 144 | 62 | (5890, 3524, 3524) | 2 |
| Handwriting | 3 | 152 | (150, 850, 850) | 26 |
| Heartbeat | 61 | 405 | (204, 205,205) | 2 |
| Japanese Vowels | 12 | 29 | (270, 370, 370) | 9 |
| PEMS-SF | 963 | 144 | (267, 173, 173) | 7 |
| SelfRegulationSCP1 | 6 | 896 | (268, 293, 293) | 2 |
| SelfRegulationSCP2 | 7 | 1152 | (200, 180, 180) | 2 |
| SpokenArabicDigits | 13 | 93 | (6599, 2199, 2199) | 10 |
| UWaveGestureLibrary | 3 | 315 | (120, 320, 320) | 8 |

Table 6: Statistics of the datasets for TSC. "Dataset Size" is organized in (Train, Validation, Test).

**Time Series Forecasting**. For TSF, following (Zhou et al., 2021; Wu et al., 2021; Nie et al., 2023; Zeng et al., 2023a; Tan et al., 2024; Chen et al., 2025), our experiments are conducted on 8 widely used benchmark datasets. The four ETT datasets (ETTh1, ETTh2, ETTm1, ETTm2) record oil temperature from two electric transformers, sampled at 15-minute and hourly intervals. The Weather dataset collects measurements of meteorological indicators in Germany every 10 minutes. The Illness dataset keeps weekly counts of patients and the influenza-like illness ratio from the United States. The Traffic dataset measures hourly road occupancy rates from sensors on San Francisco freeways. The Electricity dataset records hourly electricity consumption of Portuguese clients. Table 7 summarizes the statistics of the datasets.

| Dataset | # Variates | Series Length | Dataset Size | Frequency |
|---|---|---|---|---|
| ETTh1 | 7 | 17420 | (8545, 2881, 2881) | Hourly |
| ETTh2 | 7 | 17420 | (8545, 2881, 2881) | Hourly |
| ETTm1 | 7 | 69680 | (34465, 11521, 11521) | 15 mins |
| ETTm2 | 7 | 69680 | (34465, 11521, 11521) | 15 mins |
| Weather | 321 | 52696 | (36792, 5271, 10540) | 10 mins |
| Illness | 7 | 966 | (617, 74, 170) | Weekly |
| Traffic | 862 | 17544 | (12185, 1757, 3509) | Hourly |
| Electricity | 21 | 26304 | (18317, 2633, 5261) | Hourly |

Table 7: Statistics of the datasets for TSF. "Dataset Size" is organized in (Train, Validation, Test).

## A.2  BASELINES

For TSC, following (Zhou et al., 2023), 18 conventional and SOTA baselines are included. For TSF, following (Nie et al., 2023; Tan et al., 2024; Chen et al., 2025), 8 representative LLM-based, Transfomer-based, and non-Transformer-based baselines are included. Since several baselines are used in both TSC and TSF tasks (*e.g.*, GPT4TS, Autoformer, Dlinear, *etc.*), there are 21 distinct baselines, which are described as follows.

- GPT4TS (Zhou et al., 2023) is a foundation model built on GPT for various of time series tasks.
- Time-LLM (Jin et al., 2023) implements reprogramming to align time series with language so as to leverage pre-trained LLMs.

- CALF (Liu et al., 2025b) is built upon LLMs by designing a cross attention and feature regularization loss to align time series with language.
- PatchTST (Nie et al., 2023) divides time series into subsequence-based patches, which is then modeled as tokens through Transformer encoders with channel independence strategy.
- Flowformer (Huang et al., 2022) introduces a linear-time attention mechanism named Flow-Attention without using specific inductive biases for time series forecasting.
- Informer (Zhou et al., 2021) is a Transformer-based model that designs a ProbSparse attention mechanism to reduce time complexity on long time series.
- Transformer (Vaswani et al., 2017) is the most traditional encoder-decoder structure which can model time series with attention mechanism.
- Stationary (Liu et al., 2022b) combines series stationarization and de-stationary attention to solve the over-stationarization problem in time series forecasting.
- Reformer (Kitaev et al., 2020) applies locality-sensitive hashing and reversible residual layers to improve the efficiency of using Transformers to model long time series.
- Autoformer (Wu et al., 2021) replaces the attention block of Transformer with the Auto-Correlation mechanism which can enhance both efficiency and accuracy.
- ETSformer (Woo et al., 2022) decomposes an input time series into interpretable components with exponential smoothing attention and frequency attention for time series forecasting.
- Pyraformer (Liu et al., 2022a) designs a pyramidal attention module with inter-scale tree structures and intra-scale neighboring connections to capture multi-resolution temporal dependencies.
- FEDformer (Zhou et al., 2022) combines seasonal-trend decomposition with a frequency-enhanced Transformer to capture both global patterns and detailed structures in time series.
- Rocket (Dempster et al., 2020) achieves accurate time series classification by using linear classifiers with random convolutional kernels.
- XGBoost (Chen & Guestrin, 2016) is an efficient implementation of gradient boost decision trees for both classification and regression tasks.
- Dlinear (Zeng et al., 2023a) is a linear model that decomposes an input time series into seasonal component and trend component, and then models them with linear layers.
- LightTS (Zhang et al., 2022) is an efficient MLP-based architecture for multivariate time series forecasting by leveraging interval and continuous down-sampling to preserve temporal patterns.
- TimesNet (Wu et al., 2023) transforms time series into a 2D image-like representation using period-based patching, and then models the transformed time series with inception blocks.
- TCN (Franceschi et al., 2019) is a type of convolutional neural network that use causal, dilated convolutions with residual connections to model the temporal dependencies in time series.
- LSTNet (Lai et al., 2018) integrates RNNs and CNNs to capture temporal patterns in time series.
- LSSL (Gu et al., 2021) is proposed based on a new parameterization for state space model to capture the long-term dependencies in time series.

It is noteworthy that time series foundation models (TSFMs) such as Chronos-2 (Ansari et al., 2025) and Moirai2 (Liu et al., 2025a) encode *intra-modal knowledge, i.e.,* pre-trained knowledge from large-scale time series data, which is not orthogonal to LVMs' *cross-modal knowledge*. For example, one future research could be fine-tuning image-pretrained LVMs on large-scale time series data for integrating both cross-modal and intra-modal knowledge for enhancing performance.

## A.3 EVALUATION METRICS

For TSC, following (Wu et al., 2023; Zhou et al., 2023), accuracy (in percentage) is used as the evaluation metric. For TSF, following (Nie et al., 2023; Zeng et al., 2023a; Tan et al., 2024; Chen et al., 2025), Mean Squared Error (MSE) and Mean Absolute Error (MAE) are used as the evaluation metrics. Eq equation 1 defines MSE and MAE.

$$\mathbf{MSE} = \frac{1}{D \cdot T} \sum_{d=1}^{D} \sum_{t=1}^{T} \|\hat{\mathbf{Y}}_{dt} - \mathbf{Y}_{dt}\|_2^2, \quad \mathbf{MAE} = \frac{1}{D \cdot T} \sum_{d=1}^{D} \sum_{t=1}^{T} \|\hat{\mathbf{Y}}_{dt} - \mathbf{Y}_{dt}\|_1 \quad (1)$$

where $\hat{\mathbf{Y}} \in \mathbb{R}^{D \times T}$ stands for the prediction at $T$ future time steps of $D$ variates, $\mathbf{Y}$ stands for the ground truth, $\|\cdot\|_2$ is $\ell_2$ norm, and $\|\cdot\|_1$ is $\ell_1$ norm.

Following (Nie et al., 2023; Zeng et al., 2023a; Tan et al., 2024), for fair comparison, we adopt the standard evaluation protocol. In particular, the look-back window length is set to $H = 336$. The prediction lengths is set to $T \in \{96, 192, 336, 720\}$ for all datasets except for Illness dataset. For Illness dataset, because of its limited total length of 966 time steps, shorter look-back window of $H = 104$ and prediction lengths $T \in \{24, 36, 48, 60\}$ are employed by following (Nie et al., 2023; Zeng et al., 2023a; Tan et al., 2024). Unless otherwise noted, this configuration is applied to all of the experiments on TSF.

### A.4 Implementation Details

As described in §4, 4 pre-trained LVMs have been included in our experiments. For ViT and Swin, we use the checkpoints `ViT_B_16_Weights.IMAGENET1K_V1` and `Swin_B_Weights.IMAGENET1K_V1` respectively from *PyTorch*, which are pre-trained on $224 \times 224 \times 3$ sized images. For MAE, we use the checkpoint released by *Meta Research*[2], which is pre-trained on $224 \times 224 \times 3$ sized images with ViT-Base backbone. For SimMIM, we use the checkpoint released by *Microsoft*[3], which is pre-trained on $192 \times 192 \times 3$ sized images with Swin-Base backbone.

For TSC task, we fine-tune the LVMs using Adam optimizer with learning rate 0.0001 and batch size 32. The training runs up to a maximum of 30 epochs on the training set. Early stopping is applied after 8 consecutive epochs of no improvement is observed on the validation set.

For TSF task, we use Adam optimizer with learning rate 0.0001. For ETT and Illness datasets, the batch size is set to 32. For Weather, Traffic and Electricity datasets, the batch size is set to 256. The training runs up to 20 epochs on the training set. Early stopping is applied after 3 consecutive epochs of no improvement is observed on the validation set.

All experiments are repeated three times, and the final result is obtained by taking the average. Unless otherwise noted, the above training configuration is applied to all experiments.

The experiments are conducted on NVIDIA RTX 6000 Ada Generation GPUs with 48GB memory. All implementations are based on PyTorch 2.6.0 and utilize CUDA 12.4 for training.

### A.5 Imaging Methods

In this section, we elaborate Gramian Angular Field (GAF) and Univariate Heatmap (UVH), as they are the most frequently used imaging methods in our experiments. For more details about GAF, UVH, and other imaging methods, we refer readers to (Ni et al., 2025).

**Gramian Angular Field (GAF).** Given a univariate time series $\mathbf{x} = [x_1, ..., x_T] \in \mathbb{R}^{1 \times T}$, where $x_i$ $(1 \leq i \leq T)$ is the value at time step $i$, GAF applies Min-Max scaling to normalize each $x_i$ to $\hat{x}_i \in [0, 1]$. This normalization allows each time step to be mapped into polar coordinates with angular component $\phi_i = \arccos(\hat{x}_i)$ and radial component $r_i = i/N$, where $N$ is a constant factor.

In Gramian Sum Angular Field (GSAF), the $(i, j)$-th entry encodes the temporal correlation between time steps $i$ and $j$, which is computed as $\cos(\phi_i + \phi_j)$ and can be further expanded as following.

$$cos(\phi_i + \phi_j) = \hat{x}_i \hat{x}_j - \sqrt{1 - \hat{x}_i^2}\sqrt{1 - \hat{x}_j^2} \tag{2}$$

The resulting GAF is a matrix of size $T \times T$, with $(i, j)$-th entry defined as $\cos(\phi_i + \phi_j)$, which captures the pairwise temporal correlations among all time steps. For a multivariate time series $\mathbf{X} \in \mathbb{R}^{d \times T}$, the resulting GAF consists of $d$ individual $T \times T$ matrices.

**Univariate Heatmap (UVH).** Given a univariate time series $\mathbf{x} \in \mathbb{R}^{1 \times T}$, UVH applies Fast Fourier Transform (FFT) to compute the Fourier coefficient of each frequency component $f_i$, where $f_i \in [1, \lfloor T/2 \rfloor]$. Then it identifies the dominant frequency $f_L$ with the largest coefficient amplitude, and

---

[2] https://github.com/facebookresearch/mae
[3] https://github.com/microsoft/SimMIM

sets the potential period length as $L = \lceil T/f_L \rceil$. Next, $\mathbf{x}$ is left-padded to a length-$\hat{T}$ time series $\hat{\mathbf{x}}$, where $\hat{T}$ is a multiple of $L$. The padded time series $\hat{\mathbf{x}}$ is subsequently reshaped into a 2D image of size $L \times \hat{T}/L$ by stacking it subsequences of length $L$.

**Segment length selection for UVH**. To identify the best segment length for UVH, FFT is applied on a long look-back window of 1152 time steps on all datasets except for Illness dataset, where 104 time steps is used to accommodate its short time series. Table 8 summarizes the top-3 potential periods with the highest Fourier coefficients on each TSF dataset, along with the segment length $L$ used in the subsequent experiments involving UVH imaging method.

| | ETTh1, ETTh2 | ETTm1, ETTm2 | Weather | Illness | Traffic | Electricity |
|---|---|---|---|---|---|---|
| **Top 3 Period** | {24, 576, 384} | {96, 576, 384} | {144, 72, 576} | {52, 26, 17} | {24, 12, 168} | {24, 164, 82} |
| **Segment Length L** | 24 | 96 | 144 | 52 | 24 | 24 |

Table 8: Top-3 potential periods by FFT and segment lengths for UVH on 8 TSF datasets.

## B  FULL EXPERIMENTAL RESULTS

### B.1  FULL RESULTS OF RQ1: WHAT TYPE OF LVM BEST FITS TSC (TSF) TASK?

The detailed performance comparison between self-supervised LVMs and supervised LVMs using the best imaging method on TSC (*i.e.*, GAF) and TSF (*i.e.*, UVH) tasks are provided in Table 9 and Table 10, respectively. For TSC, supervised and self-supervised LVMs perform comparably, while for TSF, self-supervised LVMs outperform their supervised counterparts.

| **Dataset** | MAE | SimMiM | ViT | Swin |
|---|---|---|---|---|
| UWaveGestureLibrary | 85.0 | 83.1 | 88.4 | 78.9 |
| SpokenArabicDigits | 98.5 | 88.2 | 98.5 | 87.3 |
| Handwriting | 39.5 | 29.8 | 36.4 | 33.1 |
| FaceDetection | 65.4 | 57.8 | 67.4 | 50.3 |
| Average | 72.1 | 64.7 | 72.6 | 62.4 |

Table 9: Accuracy (%) comparison between self-supervised LVMs and supervised LVMs on TSC benchmark datasets. Red numbers indicate the best performance for each dataset.

### B.2  FULL RESULTS OF RQ2: WHICH IMAGING METHOD BEST FITS TSC (TSF) TASK?

This section provides detailed performance comparison of 8 imaging methods, including GAF, MVH, RP, STFT, Wavelet (Wave.), Filterbank (Filter.), UVH, and Line Plot. The best LVMs for TSC (*i.e.*, ViT) and TSF (*i.e.*, MAE) are used. Table 11 and Table 12 summarize the results for TSC and TSF, respectively. For TSF, UVH demonstrates a clear advantage on the 4 datasets in Table 12. For TSC, all benchmark datasets are used in Table 11 because the 4 datasets outlined in §4.2 are insufficient to confidently rank the compared methods using critical difference (CD) diagram (Fig. 3). Using all datasets improves confidence and helps identify the best imaging method (*i.e.*, GAF).

In Table 12, GAF (GAF$^\dagger$) represents applying GAF with framework (c) (framework (d)) in Fig. 1. For GAF$^\dagger$, we follow (Wang & Oates, 2015) to use its inverse function to recover forecasted values from the reconstructed images by the framework in Fig. 1(d). Notably, GAF$^\dagger$ scales all time series values within [0, 1] by min-max normalization to compute polar coordinates during its imaging process. The normalization uses the minimum and maximum values from the look-back window, which are used to recover any predicted values. This imposes a constraint on the predicted values, *i.e.*, the predicted values must remain within the upper and lower bounds of the look-back window, which is irrational in TSF, leading to a significant limitation and performance degradation as demonstrated in Table 12. As such, we use GAF, which outperforms GAF$^\dagger$ in Table 12, in the CD diagram in Fig. 3(b). The key limitation of GAF (and RP, STFT, Wavelet, Filterbank, Lineplot) in TSF is because they don't preserve the original time series values in their images. Without knowing historical time

| Dataset | Model | Self-Supervised | | | | Supervised | | | |
| | Model | MAE | | SimMIM | | ViT | | Swin | |
| | Metrics | MSE | MAE | MSE | MAE | MSE | MAE | MSE | MAE |
|---|---|---|---|---|---|---|---|---|---|
| ETTh1 | 96 | 0.356 | 0.383 | 0.362 | 0.383 | 0.398 | 0.401 | 0.407 | 0.429 |
| | 192 | 0.395 | 0.406 | 0.407 | 0.412 | 0.439 | 0.445 | 0.442 | 0.458 |
| | 336 | 0.417 | 0.424 | 0.422 | 0.417 | 0.462 | 0.458 | 0.467 | 0.481 |
| | 720 | 0.467 | 0.463 | 0.462 | 0.455 | 0.479 | 0.491 | 0.470 | 0.497 |
| | Average | 0.409 | 0.419 | 0.413 | 0.417 | 0.445 | 0.449 | 0.447 | 0.466 |
| ETTm1 | 96 | 0.284 | 0.333 | 0.311 | 0.350 | 0.344 | 0.384 | 0.308 | 0.360 |
| | 192 | 0.328 | 0.363 | 0.335 | 0.367 | 0.414 | 0.425 | 0.350 | 0.381 |
| | 336 | 0.357 | 0.384 | 0.356 | 0.382 | 0.411 | 0.427 | 0.385 | 0.407 |
| | 720 | 0.411 | 0.417 | 0.400 | 0.413 | 0.466 | 0.451 | 0.430 | 0.437 |
| | Average | 0.345 | 0.374 | 0.351 | 0.378 | 0.409 | 0.422 | 0.368 | 0.396 |
| Weather | 96 | 0.146 | 0.191 | 0.148 | 0.196 | 0.162 | 0.219 | 0.163 | 0.216 |
| | 192 | 0.194 | 0.238 | 0.196 | 0.243 | 0.196 | 0.244 | 0.214 | 0.262 |
| | 336 | 0.243 | 0.275 | 0.244 | 0.276 | 0.250 | 0.286 | 0.270 | 0.298 |
| | 720 | 0.318 | 0.328 | 0.340 | 0.340 | 0.329 | 0.342 | 0.345 | 0.348 |
| | Average | 0.225 | 0.258 | 0.232 | 0.264 | 0.234 | 0.273 | 0.248 | 0.281 |
| Illness | 24 | 1.977 | 0.921 | 1.934 | 0.902 | 1.989 | 0.941 | 1.990 | 0.942 |
| | 36 | 1.812 | 0.872 | 1.754 | 0.825 | 2.123 | 1.002 | 2.003 | 0.951 |
| | 48 | 1.743 | 0.856 | 1.715 | 0.867 | 2.200 | 1.032 | 2.084 | 0.991 |
| | 60 | 1.816 | 0.881 | 1.673 | 0.877 | 2.404 | 1.087 | 2.128 | 1.007 |
| | Average | 1.837 | 0.883 | 1.769 | 0.868 | 2.179 | 1.016 | 2.051 | 0.973 |

Table 10: MSE and MAE Comparison between self-supervised LVMs and supervised LVMs on TSF datasets. Red numbers indicate the best performance for each prediction length per dataset.

series values, LVMs cannot effectively forecasting future values. MVH preserves time series values, thus is better than the above methods. However, it underperforms UVH. The limitation of MVH is its mixture of variates in the image without a principled ordering of the variates.

| Dataset | GAF | MVH | RP | STFT | Wave. | Filter. | UVH | Lineplot |
|---|---|---|---|---|---|---|---|---|
| EthanolConcentration | 49.4 | 30.7 | 43.7 | 31.9 | 27.3 | 28.1 | 28.5 | 25.2 |
| FaceDetection | 67.4 | 68.3 | 65.5 | 61.1 | 63.9 | 64.7 | 67.7 | 50.3 |
| Handwriting | 36.4 | 30.8 | 45.1 | 28.2 | 34.0 | 22.3 | 25.8 | 15.9 |
| Heartbeat | 74.6 | 77.5 | 71.7 | 74.7 | 72.6 | 73.1 | 78.0 | 53.7 |
| Japanese Vowels | 98.3 | 97.8 | 87.8 | 94.8 | 94.9 | 97.0 | 96.4 | 65.7 |
| PEMS-SF | 84.2 | 87.2 | 80.1 | 68.5 | 84.7 | 71.2 | 88.1 | 73.4 |
| SelfRegulationSCP1 | 97.2 | 90.4 | 98.6 | 90.7 | 76.7 | 55.6 | 91.8 | 85.3 |
| SelfRegulationSCP2 | 58.8 | 53.3 | 54.4 | 52.7 | 54.4 | 52.2 | 52.8 | 44.5 |
| SpokenArabicDigits | 98.5 | 97.5 | 98.4 | 97.9 | 96.1 | 95.0 | 97.0 | 68.1 |
| UWaveGestureLibrary | 88.4 | 88.7 | 91.8 | 86.2 | 86.3 | 52.1 | 84.3 | 74.0 |
| Average | 75.3 | 72.2 | 73.7 | 68.7 | 69.1 | 61.1 | 71.0 | 55.6 |

Table 11: Accuracy (%) comparison of 8 imaging methods on TSC benchmark datasets. Red numbers indicate the best performance for each dataset.

### B.3 FULL RESULTS OF RQ3: TIME SERIES CLASSIFICATION

Table 13 provides the full results of the compared methods on 10 benchmark datasets for TSC. The LVM results are averaged over 3 runs. The corresponding standard deviations reported in Table 14.

| Imaging Method | GAF | | GAF† | | MVH | | RP | | STFT | | Wave. | | Filter. | | UVH | | Lineplot | |
|---|---|---|---|---|---|---|---|---|---|---|---|---|---|---|---|---|---|---|
| Dataset / Metrics | MSE | MAE | MSE | MAE | MSE | MAE | MSE | MAE | MSE | MAE | MSE | MAE | MSE | MAE | MSE | MAE | MSE | MAE |
| ETTh1 96 | 0.986 | 0.783 | 1.224 | 0.850 | 0.484 | 0.471 | 0.969 | 0.771 | 0.534 | 0.533 | 0.621 | 0.582 | 0.820 | 0.684 | 0.356 | 0.383 | 0.902 | 0.751 |
| ETTh1 192 | 1.004 | 0.797 | 1.227 | 0.854 | 0.575 | 0.517 | 0.971 | 0.775 | 0.621 | 0.587 | 0.650 | 0.600 | 0.864 | 0.707 | 0.395 | 0.406 | 1.204 | 0.894 |
| ETTh1 336 | 1.038 | 0.820 | 1.214 | 0.857 | 0.623 | 0.546 | 0.989 | 0.788 | 0.602 | 0.573 | 0.681 | 0.616 | 0.827 | 0.693 | 0.417 | 0.424 | 1.223 | 0.901 |
| ETTh1 720 | 1.008 | 0.812 | 1.190 | 0.863 | 0.737 | 0.612 | 1.062 | 0.825 | 0.669 | 0.621 | 0.699 | 0.633 | 0.858 | 0.720 | 0.467 | 0.463 | 1.150 | 0.852 |
| ETTh1 Average | 1.009 | 0.803 | 1.214 | 0.856 | 0.605 | 0.537 | 0.998 | 0.790 | 0.607 | 0.579 | 0.663 | 0.608 | 0.842 | 0.701 | 0.409 | 0.419 | 1.120 | 0.850 |
| ETTm1 96 | 0.836 | 0.729 | 0.956 | 0.676 | 0.310 | 0.352 | 0.849 | 0.719 | 0.420 | 0.470 | 0.449 | 0.490 | 0.793 | 0.648 | 0.284 | 0.333 | 0.842 | 0.735 |
| ETTm1 192 | 0.830 | 0.717 | 0.967 | 0.685 | 0.386 | 0.400 | 0.865 | 0.726 | 0.466 | 0.496 | 0.504 | 0.524 | 0.798 | 0.649 | 0.328 | 0.363 | 0.840 | 0.726 |
| ETTm1 336 | 0.853 | 0.725 | 0.988 | 0.697 | 0.393 | 0.402 | 0.872 | 0.728 | 0.506 | 0.519 | 0.532 | 0.535 | 0.883 | 0.690 | 0.357 | 0.384 | 0.841 | 0.726 |
| ETTm1 720 | 0.865 | 0.726 | 1.107 | 0.779 | 0.488 | 0.467 | 0.928 | 0.754 | 0.543 | 0.536 | 0.586 | 0.563 | 0.899 | 0.703 | 0.411 | 0.417 | 0.872 | 0.741 |
| ETTm1 Average | 0.846 | 0.724 | 1.005 | 0.709 | 0.394 | 0.405 | 0.879 | 0.732 | 0.484 | 0.505 | 0.518 | 0.528 | 0.843 | 0.673 | 0.345 | 0.374 | 0.849 | 0.732 |
| Illness 24 | 5.066 | 1.591 | 6.172 | 2.618 | 2.326 | 0.976 | 5.106 | 1.594 | 5.049 | 1.591 | 4.270 | 1.484 | 7.863 | 2.056 | 1.977 | 0.921 | 4.993 | 1.508 |
| Illness 36 | 5.236 | 1.628 | 5.497 | 2.627 | 2.152 | 0.919 | 5.309 | 1.629 | 5.143 | 1.598 | 4.293 | 1.487 | 8.169 | 2.122 | 1.812 | 0.872 | 5.147 | 1.593 |
| Illness 48 | 5.118 | 1.600 | 5.218 | 2.448 | 2.111 | 0.966 | 5.381 | 1.643 | 5.010 | 1.574 | 4.190 | 1.451 | 7.144 | 1.962 | 1.743 | 0.856 | 5.039 | 1.541 |
| Illness 60 | 5.349 | 1.641 | 5.299 | 2.239 | 2.118 | 0.968 | 5.586 | 1.685 | 5.164 | 1.601 | 4.045 | 1.430 | 7.193 | 1.986 | 1.816 | 0.881 | 5.235 | 1.601 |
| Illness Average | 5.192 | 1.615 | 5.547 | 2.483 | 2.177 | 0.957 | 5.346 | 1.638 | 5.092 | 1.591 | 4.200 | 1.463 | 7.592 | 2.032 | 1.837 | 0.883 | 5.104 | 1.561 |
| Weather 96 | 0.581 | 0.554 | 0.961 | 0.592 | 0.153 | 0.202 | 0.647 | 0.610 | 0.202 | 0.294 | 0.224 | 0.312 | 0.515 | 0.488 | 0.146 | 0.191 | 0.588 | 0.561 |
| Weather 192 | 0.598 | 0.567 | 0.995 | 0.614 | 0.194 | 0.241 | 0.649 | 0.607 | 0.251 | 0.336 | 0.273 | 0.354 | 0.516 | 0.488 | 0.194 | 0.238 | 0.604 | 0.574 |
| Weather 336 | 0.593 | 0.558 | 1.039 | 0.637 | 0.239 | 0.275 | 0.674 | 0.619 | 0.294 | 0.364 | 0.330 | 0.388 | 0.505 | 0.484 | 0.243 | 0.275 | 0.601 | 0.568 |
| Weather 720 | 0.611 | 0.574 | 1.051 | 0.644 | 0.337 | 0.344 | 0.640 | 0.593 | 0.364 | 0.413 | 0.411 | 0.433 | 0.513 | 0.499 | 0.318 | 0.328 | 0.617 | 0.582 |
| Weather Average | 0.596 | 0.563 | 1.012 | 0.622 | 0.231 | 0.266 | 0.653 | 0.607 | 0.278 | 0.352 | 0.310 | 0.372 | 0.512 | 0.490 | 0.225 | 0.258 | 0.603 | 0.571 |

Table 12: MSE and MAE comparison of 8 imaging methods on TSF benchmark datasets. Red numbers indicate the best performance for each dataset. GAF represents applying GAF with the framework in Fig. 1(c). GAF† represents applying GAF with the framework in Fig. 1(d).

| Dataset | MAE | ViT | XGBoost | Rocket | LSTNet | LSSL | TCN | Trans. | Re. | In. | Pyra. | Auto. | Station. | FED. | ETS. | Flow. | Dlinear | LightTS | TimesNet | GPT4TS |
|---|---|---|---|---|---|---|---|---|---|---|---|---|---|---|---|---|---|---|---|---|
| EthanolConcentration | 41.4 | 49.4 | 43.7 | 45.2 | 39.9 | 31.1 | 28.9 | 32.7 | 31.9 | 31.6 | 30.8 | 31.6 | 32.7 | 31.2 | 28.1 | 33.8 | 32.6 | 29.7 | 35.7 | 34.2 |
| FaceDetection | 65.4 | 67.4 | 63.3 | 64.7 | 65.7 | 66.7 | 52.8 | 67.3 | 68.6 | 67.0 | 65.7 | 68.4 | 68.0 | 66.0 | 66.3 | 67.6 | 68.0 | 67.5 | 68.6 | 69.2 |
| Handwriting | 39.5 | 36.4 | 15.8 | 58.8 | 25.8 | 24.6 | 53.3 | 32.0 | 27.4 | 32.8 | 29.4 | 36.7 | 31.6 | 28.0 | 32.5 | 33.8 | 27.0 | 26.1 | 32.1 | 32.7 |
| Heartbeat | 86.8 | 74.6 | 73.2 | 75.6 | 77.1 | 72.7 | 75.6 | 76.1 | 77.1 | 80.5 | 75.6 | 74.6 | 73.7 | 73.7 | 71.2 | 77.6 | 75.1 | 75.1 | 78.0 | 77.2 |
| Japanese Vowels | 95.4 | 98.3 | 86.5 | 96.2 | 98.1 | 98.4 | 98.9 | 98.7 | 97.8 | 98.9 | 98.4 | 96.2 | 98.7 | 95.9 | 98.9 | 96.2 | 96.2 | 98.4 | 98.6 | |
| PEMS-SF | 84.4 | 84.2 | 98.3 | 75.1 | 86.7 | 86.1 | 68.8 | 82.1 | 82.7 | 81.5 | 83.2 | 82.7 | 87.3 | 80.9 | 86.0 | 83.8 | 75.1 | 88.4 | 89.6 | 87.9 |
| SelfRegulationSCP1 | 95.2 | 97.2 | 84.6 | 90.8 | 84.0 | 90.8 | 84.6 | 92.2 | 90.4 | 90.1 | 88.1 | 84.0 | 89.4 | 88.7 | 89.6 | 92.5 | 87.3 | 89.8 | 91.8 | 93.2 |
| SelfRegulationSCP2 | 59.4 | 58.8 | 48.9 | 53.3 | 52.8 | 52.2 | 55.6 | 53.9 | 56.7 | 53.3 | 53.3 | 50.6 | 57.2 | 54.4 | 55.0 | 56.1 | 50.5 | 51.1 | 57.2 | 59.4 |
| SpokenArabicDigits | 98.5 | 98.5 | 69.6 | 71.2 | 100.0 | 100.0 | 95.6 | 98.4 | 97.0 | 100.0 | 99.6 | 100.0 | 100.0 | 100.0 | 100.0 | 98.8 | 81.4 | 100.0 | 99.0 | 99.2 |
| UWaveGestureLibrary | 85.0 | 88.4 | 75.9 | 94.4 | 87.8 | 85.9 | 88.4 | 85.6 | 85.6 | 85.6 | 83.4 | 85.9 | 87.5 | 85.3 | 85.0 | 86.6 | 82.1 | 80.3 | 85.3 | 88.1 |
| Average | 75.1 | 75.3 | 66.0 | 72.5 | 71.8 | 70.9 | 70.3 | 71.9 | 71.5 | 72.1 | 70.8 | 71.1 | 72.7 | 70.7 | 71.0 | 73.0 | 67.5 | 70.4 | 73.6 | 74.0 |
| # Wins | 2 | 3 | 1 | 1 | 1 | 1 | 1 | 0 | 0 | 1 | 0 | 1 | 2 | 1 | 1 | 0 | 0 | 1 | 0 | 2 |

Table 13: Accuracy (%) of the compared methods in TSC on 10 benchmark datasets. Red numbers are the the best results. # Wins is the number of times the method performs the best.

| Dataset | MAE | ViT |
|---|---|---|
| EthanolConcentration | 41.4 ± 0.5 | 49.4 ± 0.9 |
| FaceDetection | 65.4 ± 1.2 | 67.4 ± 1.5 |
| Handwriting | 39.5 ± 1.5 | 36.4 ± 1.3 |
| Heartbeat | 86.8 ± 2.1 | 74.6 ± 0.6 |
| Japanese Vowels | 95.4 ± 0.3 | 98.3 ± 0.3 |
| PEMS-SF | 84.4 ± 0.4 | 84.2 ± 0.5 |
| SelfRegulationSCP1 | 95.2 ± 0.6 | 97.2 ± 0.9 |
| SelfRegulationSCP2 | 59.4 ± 1.5 | 58.8 ± 1.3 |
| SpokenArabicDigits | 98.5 ± 0.5 | 98.5 ± 0.5 |
| UWaveGestureLibrary | 85.0 ± 1.7 | 88.4 ± 1.4 |

Table 14: Standard deviation of LVMs on TSC datasets.

## B.4 Full Results of RQ3: Time Series Forecasting

Table 15 provides the full result of the compared methods on 8 benchmark datasets for TSF. The results of LVMs are averaged over 3 runs with standard deviations reported in Table 16.

## B.5 Full Results of RQ4: Are the pre-trained parameters in LVMs useful in time series tasks?

Table 17 and Table 18 provide the results of comparing different fine-tuning strategies on TSC and TSF tasks, respectively. In this ablation analysis, we progressively freeze the components of the Transformer blocks in LVMs (Fig. 5) with the following settings: (a) Fine-tune all parameters; (b) Fine-tune all parameters but freeze CLS token and Mask token; (c) Fine-tune MLP and norm layers

| | Method | MAE | | ViT | | Time-LLM | | GPT4TS | | CALF | | Dlinear | | PatchTST | | TimesNet | | FEDformer | | Autoformer | |
|---|---|---|---|---|---|---|---|---|---|---|---|---|---|---|---|---|---|---|---|---|---|
| | Metrics | MSE | MAE | MSE | MAE | MSE | MAE | MSE | MAE | MSE | MAE | MSE | MAE | MSE | MAE | MSE | MAE | MSE | MAE | MSE | MAE |
| ETTh1 | 96 | 0.356 | 0.383 | 0.398 | 0.401 | 0.376 | 0.402 | 0.370 | 0.389 | 0.370 | 0.393 | 0.375 | 0.399 | 0.370 | 0.399 | 0.384 | 0.402 | 0.376 | 0.419 | 0.449 | 0.459 |
| | 192 | 0.395 | 0.406 | 0.439 | 0.445 | 0.407 | 0.421 | 0.412 | 0.413 | 0.429 | 0.426 | 0.405 | 0.416 | 0.413 | 0.421 | 0.436 | 0.429 | 0.420 | 0.448 | 0.500 | 0.482 |
| | 336 | 0.417 | 0.424 | 0.462 | 0.458 | 0.430 | 0.438 | 0.448 | 0.431 | 0.451 | 0.440 | 0.439 | 0.443 | 0.422 | 0.436 | 0.491 | 0.469 | 0.459 | 0.465 | 0.521 | 0.496 |
| | 720 | 0.467 | 0.463 | 0.479 | 0.491 | 0.457 | 0.468 | 0.441 | 0.449 | 0.476 | 0.466 | 0.472 | 0.490 | 0.447 | 0.466 | 0.521 | 0.500 | 0.506 | 0.507 | 0.514 | 0.512 |
| ETTh2 | 96 | 0.297 | 0.341 | 0.302 | 0.355 | 0.286 | 0.346 | 0.280 | 0.335 | 0.284 | 0.336 | 0.289 | 0.353 | 0.274 | 0.336 | 0.340 | 0.374 | 0.358 | 0.397 | 0.346 | 0.388 |
| | 192 | 0.356 | 0.386 | 0.394 | 0.411 | 0.361 | 0.391 | 0.348 | 0.380 | 0.353 | 0.378 | 0.383 | 0.418 | 0.339 | 0.379 | 0.402 | 0.414 | 0.429 | 0.439 | 0.456 | 0.452 |
| | 336 | 0.371 | 0.402 | 0.423 | 0.429 | 0.390 | 0.414 | 0.380 | 0.405 | 0.361 | 0.394 | 0.448 | 0.465 | 0.329 | 0.380 | 0.452 | 0.452 | 0.496 | 0.487 | 0.482 | 0.486 |
| | 720 | 0.403 | 0.430 | 0.438 | 0.449 | 0.405 | 0.434 | 0.406 | 0.436 | 0.406 | 0.428 | 0.605 | 0.551 | 0.379 | 0.422 | 0.462 | 0.468 | 0.463 | 0.474 | 0.515 | 0.511 |
| ETTm1 | 96 | 0.284 | 0.333 | 0.344 | 0.384 | 0.291 | 0.341 | 0.300 | 0.340 | 0.323 | 0.350 | 0.299 | 0.343 | 0.290 | 0.342 | 0.338 | 0.375 | 0.379 | 0.419 | 0.505 | 0.475 |
| | 192 | 0.328 | 0.363 | 0.414 | 0.425 | 0.341 | 0.369 | 0.343 | 0.368 | 0.375 | 0.376 | 0.335 | 0.365 | 0.332 | 0.369 | 0.374 | 0.387 | 0.426 | 0.441 | 0.553 | 0.496 |
| | 336 | 0.357 | 0.384 | 0.411 | 0.427 | 0.359 | 0.379 | 0.376 | 0.386 | 0.411 | 0.401 | 0.369 | 0.386 | 0.366 | 0.392 | 0.410 | 0.411 | 0.445 | 0.459 | 0.621 | 0.537 |
| | 720 | 0.411 | 0.417 | 0.466 | 0.451 | 0.433 | 0.419 | 0.431 | 0.416 | 0.476 | 0.438 | 0.425 | 0.421 | 0.416 | 0.420 | 0.478 | 0.450 | 0.543 | 0.490 | 0.671 | 0.561 |
| ETTm2 | 96 | 0.173 | 0.258 | 0.179 | 0.265 | 0.162 | 0.248 | 0.163 | 0.249 | 0.177 | 0.255 | 0.167 | 0.269 | 0.165 | 0.255 | 0.187 | 0.267 | 0.203 | 0.287 | 0.255 | 0.339 |
| | 192 | 0.231 | 0.297 | 0.262 | 0.319 | 0.235 | 0.304 | 0.222 | 0.291 | 0.245 | 0.300 | 0.224 | 0.303 | 0.220 | 0.292 | 0.249 | 0.309 | 0.269 | 0.328 | 0.281 | 0.340 |
| | 336 | 0.282 | 0.340 | 0.346 | 0.371 | 0.280 | 0.329 | 0.273 | 0.327 | 0.309 | 0.341 | 0.281 | 0.342 | 0.274 | 0.329 | 0.321 | 0.351 | 0.325 | 0.366 | 0.339 | 0.372 |
| | 720 | 0.386 | 0.413 | 0.411 | 0.392 | 0.366 | 0.382 | 0.357 | 0.376 | 0.402 | 0.395 | 0.397 | 0.421 | 0.362 | 0.385 | 0.408 | 0.403 | 0.421 | 0.415 | 0.433 | 0.432 |
| Weather | 96 | 0.146 | 0.191 | 0.162 | 0.219 | 0.155 | 0.199 | 0.148 | 0.188 | 0.168 | 0.207 | 0.176 | 0.237 | 0.149 | 0.198 | 0.172 | 0.220 | 0.217 | 0.296 | 0.266 | 0.336 |
| | 192 | 0.194 | 0.238 | 0.196 | 0.244 | 0.223 | 0.261 | 0.192 | 0.230 | 0.216 | 0.251 | 0.220 | 0.282 | 0.194 | 0.241 | 0.219 | 0.261 | 0.276 | 0.336 | 0.307 | 0.367 |
| | 336 | 0.243 | 0.275 | 0.250 | 0.286 | 0.251 | 0.279 | 0.246 | 0.273 | 0.271 | 0.292 | 0.265 | 0.319 | 0.245 | 0.282 | 0.280 | 0.306 | 0.339 | 0.380 | 0.359 | 0.395 |
| | 720 | 0.318 | 0.328 | 0.329 | 0.342 | 0.320 | 0.328 | 0.320 | 0.328 | 0.350 | 0.345 | 0.333 | 0.362 | 0.314 | 0.334 | 0.365 | 0.359 | 0.403 | 0.428 | 0.419 | 0.428 |
| Illness | 24 | 1.977 | 0.921 | 1.989 | 0.941 | 1.792 | 0.807 | 1.869 | 0.823 | 1.460 | 0.788 | 2.215 | 1.081 | 1.319 | 0.754 | 2.317 | 0.934 | 3.228 | 1.260 | 3.483 | 1.287 |
| | 36 | 1.812 | 0.872 | 2.123 | 1.002 | 1.833 | 0.833 | 1.853 | 0.854 | 1.573 | 0.837 | 1.963 | 0.963 | 1.430 | 0.834 | 1.972 | 0.920 | 2.679 | 1.080 | 3.103 | 1.148 |
| | 48 | 1.743 | 0.856 | 2.200 | 1.032 | 2.269 | 1.012 | 1.886 | 0.855 | 2.130 | 0.890 | 2.130 | 0.963 | 1.553 | 0.815 | 2.238 | 0.940 | 2.622 | 1.078 | 2.669 | 1.085 |
| | 60 | 1.816 | 0.881 | 2.404 | 1.087 | 2.177 | 0.925 | 1.877 | 0.877 | 1.982 | 0.962 | 2.368 | 1.096 | 1.470 | 0.788 | 2.027 | 0.928 | 2.857 | 1.157 | 2.770 | 1.125 |
| Traffic | 96 | 0.346 | 0.232 | 0.403 | 0.330 | 0.392 | 0.267 | 0.396 | 0.264 | 0.416 | 0.274 | 0.410 | 0.282 | 0.360 | 0.249 | 0.593 | 0.321 | 0.587 | 0.366 | 0.613 | 0.388 |
| | 192 | 0.376 | 0.245 | 0.411 | 0.334 | 0.409 | 0.271 | 0.412 | 0.268 | 0.430 | 0.276 | 0.423 | 0.287 | 0.379 | 0.256 | 0.617 | 0.336 | 0.604 | 0.373 | 0.616 | 0.382 |
| | 336 | 0.389 | 0.252 | 0.429 | 0.335 | 0.434 | 0.296 | 0.421 | 0.273 | 0.451 | 0.286 | 0.436 | 0.296 | 0.392 | 0.264 | 0.629 | 0.336 | 0.621 | 0.383 | 0.622 | 0.337 |
| | 720 | 0.432 | 0.293 | 0.477 | 0.371 | 0.451 | 0.291 | 0.455 | 0.291 | 0.478 | 0.301 | 0.466 | 0.315 | 0.432 | 0.286 | 0.640 | 0.350 | 0.626 | 0.382 | 0.660 | 0.408 |
| Electricity | 96 | 0.127 | 0.217 | 0.152 | 0.244 | 0.137 | 0.233 | 0.141 | 0.239 | 0.147 | 0.240 | 0.140 | 0.237 | 0.129 | 0.222 | 0.168 | 0.272 | 0.193 | 0.308 | 0.201 | 0.317 |
| | 192 | 0.148 | 0.237 | 0.164 | 0.249 | 0.152 | 0.247 | 0.158 | 0.253 | 0.163 | 0.254 | 0.153 | 0.249 | 0.157 | 0.240 | 0.184 | 0.289 | 0.201 | 0.315 | 0.222 | 0.334 |
| | 336 | 0.163 | 0.253 | 0.173 | 0.275 | 0.169 | 0.267 | 0.172 | 0.266 | 0.178 | 0.270 | 0.169 | 0.267 | 0.163 | 0.259 | 0.198 | 0.300 | 0.214 | 0.329 | 0.231 | 0.338 |
| | 720 | 0.199 | 0.293 | 0.202 | 0.294 | 0.200 | 0.290 | 0.207 | 0.293 | 0.215 | 0.300 | 0.203 | 0.301 | 0.197 | 0.290 | 0.220 | 0.320 | 0.246 | 0.355 | 0.254 | 0.361 |
| # Wins | | 28 | | 0 | | 5 | | 14 | | 1 | | 0 | | 20 | | 0 | | 0 | | 0 | |

Table 15: MSE and MAE evaluation of the compared methods in TSF on benchmark datasets. Red (Blue) numbers are the best (second best) results on each prediction length per dataset. # Wins is the number of times the method performs the best.

| | Method | MAE | | ViT | |
|---|---|---|---|---|---|
| | Metrics | MSE | MAE | MSE | MAE |
| ETTh1 | 96 | $0.356 \pm 0.001$ | $0.383 \pm 0.005$ | $0.398 \pm 0.011$ | $0.401 \pm 0.012$ |
| | 192 | $0.395 \pm 0.001$ | $0.406 \pm 0.001$ | $0.439 \pm 0.005$ | $0.445 \pm 0.003$ |
| | 336 | $0.417 \pm 0.001$ | $0.424 \pm 0.001$ | $0.462 \pm 0.004$ | $0.458 \pm 0.004$ |
| | 720 | $0.467 \pm 0.012$ | $0.463 \pm 0.010$ | $0.479 \pm 0.011$ | $0.491 \pm 0.008$ |
| ETTh2 | 96 | $0.297 \pm 0.000$ | $0.341 \pm 0.004$ | $0.302 \pm 0.001$ | $0.355 \pm 0.000$ |
| | 192 | $0.356 \pm 0.005$ | $0.386 \pm 0.011$ | $0.394 \pm 0.001$ | $0.411 \pm 0.001$ |
| | 336 | $0.371 \pm 0.003$ | $0.402 \pm 0.004$ | $0.423 \pm 0.003$ | $0.429 \pm 0.001$ |
| | 720 | $0.403 \pm 0.001$ | $0.430 \pm 0.005$ | $0.438 \pm 0.005$ | $0.449 \pm 0.002$ |
| ETTm1 | 96 | $0.284 \pm 0.003$ | $0.333 \pm 0.004$ | $0.344 \pm 0.001$ | $0.384 \pm 0.002$ |
| | 192 | $0.328 \pm 0.001$ | $0.363 \pm 0.002$ | $0.414 \pm 0.003$ | $0.425 \pm 0.003$ |
| | 336 | $0.357 \pm 0.001$ | $0.384 \pm 0.001$ | $0.411 \pm 0.002$ | $0.427 \pm 0.007$ |
| | 720 | $0.411 \pm 0.002$ | $0.417 \pm 0.001$ | $0.466 \pm 0.003$ | $0.451 \pm 0.002$ |
| ETTm2 | 96 | $0.173 \pm 0.005$ | $0.258 \pm 0.004$ | $0.179 \pm 0.003$ | $0.265 \pm 0.004$ |
| | 192 | $0.231 \pm 0.004$ | $0.297 \pm 0.003$ | $0.262 \pm 0.002$ | $0.319 \pm 0.001$ |
| | 336 | $0.282 \pm 0.001$ | $0.340 \pm 0.004$ | $0.346 \pm 0.001$ | $0.371 \pm 0.003$ |
| | 720 | $0.386 \pm 0.002$ | $0.413 \pm 0.003$ | $0.411 \pm 0.002$ | $0.392 \pm 0.004$ |
| Weather | 96 | $0.146 \pm 0.000$ | $0.191 \pm 0.002$ | $0.162 \pm 0.001$ | $0.219 \pm 0.003$ |
| | 192 | $0.194 \pm 0.001$ | $0.238 \pm 0.002$ | $0.196 \pm 0.002$ | $0.244 \pm 0.003$ |
| | 336 | $0.243 \pm 0.000$ | $0.275 \pm 0.001$ | $0.250 \pm 0.001$ | $0.286 \pm 0.000$ |
| | 720 | $0.318 \pm 0.001$ | $0.328 \pm 0.001$ | $0.329 \pm 0.002$ | $0.342 \pm 0.002$ |
| Illness | 24 | $1.977 \pm 0.017$ | $0.921 \pm 0.003$ | $1.989 \pm 0.011$ | $0.941 \pm 0.004$ |
| | 36 | $1.812 \pm 0.014$ | $0.872 \pm 0.009$ | $2.123 \pm 0.006$ | $1.002 \pm 0.003$ |
| | 48 | $1.743 \pm 0.029$ | $0.856 \pm 0.012$ | $2.200 \pm 0.009$ | $1.032 \pm 0.005$ |
| | 60 | $1.816 \pm 0.022$ | $0.881 \pm 0.008$ | $2.404 \pm 0.018$ | $1.087 \pm 0.011$ |
| Traffic | 96 | $0.346 \pm 0.004$ | $0.232 \pm 0.003$ | $0.403 \pm 0.003$ | $0.330 \pm 0.002$ |
| | 192 | $0.376 \pm 0.006$ | $0.245 \pm 0.002$ | $0.411 \pm 0.001$ | $0.334 \pm 0.000$ |
| | 336 | $0.389 \pm 0.004$ | $0.252 \pm 0.003$ | $0.429 \pm 0.002$ | $0.335 \pm 0.005$ |
| | 720 | $0.432 \pm 0.002$ | $0.293 \pm 0.005$ | $0.477 \pm 0.004$ | $0.371 \pm 0.002$ |
| Electricity | 96 | $0.127 \pm 0.001$ | $0.217 \pm 0.000$ | $0.152 \pm 0.001$ | $0.244 \pm 0.001$ |
| | 192 | $0.148 \pm 0.004$ | $0.237 \pm 0.000$ | $0.164 \pm 0.003$ | $0.249 \pm 0.001$ |
| | 336 | $0.163 \pm 0.001$ | $0.253 \pm 0.002$ | $0.173 \pm 0.002$ | $0.275 \pm 0.003$ |
| | 720 | $0.199 \pm 0.002$ | $0.293 \pm 0.001$ | $0.202 \pm 0.001$ | $0.294 \pm 0.003$ |

Table 16: Standard deviation of LVMs on TSF datasets.

only; (d) Fine-tune norm layers only; (e) Freeze all parameters (*i.e.*, zero-shot); and (f) Randomly initialize an LVM and train it from scratch. From Table 17, for TSC, fully fine-tuning all parameters yields the best performance. From Table 18, for TSF, fine-tuning only the norm layer leads to better performance than other settings.

| Dataset | (a) | (b) | (c) | (d) | (e) | (f) |
|---|---|---|---|---|---|---|
| UWaveGestureLibrary | 88.4 | 87.5 | 88.7 | 81.6 | 84.0 | 73.4 |
| SpokenArabicDigits | 98.5 | 98.2 | 98.4 | 98.0 | 98.5 | 97.0 |
| Handwriting | 36.4 | 35.2 | 35.5 | 28.5 | 27.8 | 24.3 |
| FaceDetection | 67.4 | 66.3 | 67.1 | 65.2 | 66.7 | 65.0 |

Table 17: Accuracy (%) comparison of different fine-tuning strategies for on TSC benchmark datasets. Red numbers indicate the best performance for each dataset.

| Fine-tuning Strategy | | (a) | | (b) | | (c) | | (d) | | (e) | | (f) | |
|---|---|---|---|---|---|---|---|---|---|---|---|---|---|---|
| Dataset | Metrics | MSE | MAE | MSE | MAE | MSE | MAE | MSE | MAE | MSE | MAE | MSE | MAE |
| ETTh1 | 96 | 0.512 | 0.448 | 0.481 | 0.435 | 0.477 | 0.418 | 0.356 | 0.383 | 0.426 | 0.397 | 0.412 | 0.431 |
| | 192 | 0.511 | 0.453 | 0.520 | 0.455 | 0.526 | 0.456 | 0.395 | 0.406 | 0.448 | 0.417 | 0.462 | 0.462 |
| | 336 | 0.610 | 0.512 | 0.537 | 0.484 | 0.584 | 0.497 | 0.417 | 0.424 | 0.478 | 0.439 | 0.489 | 0.479 |
| | 720 | 0.598 | 0.523 | 0.581 | 0.526 | 0.539 | 0.493 | 0.467 | 0.463 | 0.454 | 0.453 | 0.536 | 0.514 |
| | Average | 0.558 | 0.484 | 0.530 | 0.475 | 0.532 | 0.466 | 0.409 | 0.419 | 0.452 | 0.427 | 0.475 | 0.472 |
| ETTm1 | 96 | 0.303 | 0.334 | 0.320 | 0.348 | 0.306 | 0.338 | 0.284 | 0.333 | 0.394 | 0.370 | 0.323 | 0.367 |
| | 192 | 0.385 | 0.385 | 0.389 | 0.385 | 0.385 | 0.378 | 0.328 | 0.363 | 0.404 | 0.381 | 0.344 | 0.383 |
| | 336 | 0.409 | 0.403 | 0.419 | 0.407 | 0.420 | 0.402 | 0.357 | 0.384 | 0.421 | 0.398 | 0.375 | 0.403 |
| | 720 | 0.500 | 0.461 | 0.503 | 0.461 | 0.474 | 0.444 | 0.411 | 0.417 | 0.462 | 0.426 | 0.446 | 0.445 |
| | Average | 0.399 | 0.396 | 0.408 | 0.400 | 0.396 | 0.391 | 0.345 | 0.374 | 0.420 | 0.394 | 0.372 | 0.400 |
| Illness | 24 | 1.888 | 0.818 | 1.683 | 0.789 | 2.043 | 0.818 | 1.977 | 0.921 | 2.227 | 0.971 | 1.719 | 0.799 |
| | 36 | 1.542 | 0.781 | 1.632 | 0.801 | 1.573 | 0.775 | 1.812 | 0.872 | 2.023 | 0.932 | 1.541 | 0.753 |
| | 48 | 1.682 | 0.829 | 1.839 | 0.845 | 1.548 | 0.783 | 1.743 | 0.856 | 1.947 | 0.920 | 1.687 | 0.817 |
| | 60 | 2.012 | 0.859 | 1.977 | 0.921 | 1.783 | 0.860 | 1.816 | 0.881 | 1.952 | 0.939 | 1.944 | 0.880 |
| | Average | 1.781 | 0.822 | 1.783 | 0.839 | 1.737 | 0.809 | 1.837 | 0.883 | 2.037 | 0.941 | 1.723 | 0.812 |
| Weather | 96 | 0.172 | 0.213 | 0.174 | 0.213 | 0.171 | 0.208 | 0.146 | 0.191 | 0.274 | 0.280 | 0.154 | 0.201 |
| | 192 | 0.225 | 0.259 | 0.233 | 0.263 | 0.225 | 0.256 | 0.194 | 0.238 | 0.284 | 0.294 | 0.199 | 0.245 |
| | 336 | 0.298 | 0.302 | 0.296 | 0.304 | 0.293 | 0.303 | 0.243 | 0.275 | 0.311 | 0.316 | 0.265 | 0.292 |
| | 720 | 0.397 | 0.363 | 0.397 | 0.364 | 0.367 | 0.361 | 0.318 | 0.328 | 0.364 | 0.354 | 0.344 | 0.350 |
| | Average | 0.273 | 0.284 | 0.275 | 0.286 | 0.264 | 0.282 | 0.225 | 0.258 | 0.308 | 0.311 | 0.241 | 0.272 |

Table 18: MSE and MAE comparison of different fine-tuning strategies on TSF benchmark datasets. Red numbers indicate the best performance for each dataset.

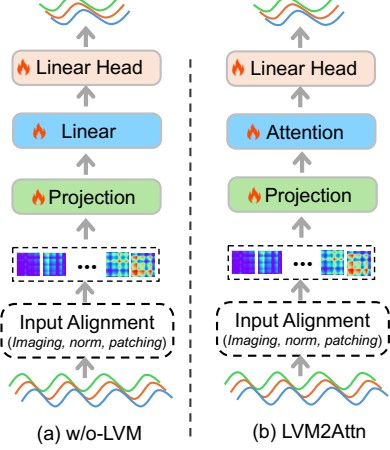

(a) w/o-LVM          (b) LVM2Attn

Figure 11: Illustration of LVM's ablation models. (a) is the model W/O-LVM, which replaces the Transformer blocks in LVMs with a linear layer. (b) is the model LVM2ATTN, which replaces the Transformer blocks in LVMs with a single mult-head attention layer.

### B.6 FULL RESULTS OF RQ5: HOW USEFUL ARE LVMS' ARCHITECTURES?

Table 19 and Table 20 provide the results of comparing LVMs' architecture and two ablation models, W/O-LVM and LVM2ATTN, on TSC and TSF tasks, respectively. Fig. 11 illustrates the ablation models. Both models keep the projection layer in LVM encoder. The model W/O-LVM replaces the Transformer blocks with a linear layer. The model LVM2ATTN replaces the Transformer blocks with a single multi-head self-attention layer. Other components including input alignment and the linear head remain unchanged. In this comparison, all models are trained from scratch without using pre-trained parameters. From Table 19 and Table 20, without pre-trained knowledge, LVMs perform on par with W/O-LVMand LVM2ATTN on both TSC and TSF tasks. However, as demonstrated in Table 17 and Table 18, with pre-training parameters, LVMs outperform both ablation models.

| Dataset | LVMs | W/O-LVM | LVM2ATTN |
|---|---|---|---|
| UWaveGestureLibrary | 73.4 | 78.6 | 80.1 |
| SpokenArabicDigits | 97.0 | 96.4 | 96.5 |
| Handwriting | 24.3 | 22.4 | 20.7 |
| FaceDetection | 65.0 | 64.1 | 66.2 |

Table 19: Accuracy (%) comparison between LVM architecture and ablation models on TSC benchmark datasets. Red numbers indicate the best performance for each dataset.

| Model | | LVMs | | W/O-LVM | | LVM2ATTN | |
|---|---|---|---|---|---|---|---|
| Dataset | Metrics | MSE | MAE | MSE | MAE | MSE | MAE |
| ETTh1 | 96 | 0.412 | 0.431 | 0.392 | 0.410 | 0.391 | 0.417 |
| | 192 | 0.462 | 0.462 | 0.418 | 0.426 | 0.414 | 0.435 |
| | 336 | 0.489 | 0.479 | 0.441 | 0.443 | 0.438 | 0.452 |
| | 720 | 0.536 | 0.514 | 0.441 | 0.465 | 0.469 | 0.485 |
| | Average | 0.475 | 0.472 | 0.423 | 0.436 | 0.428 | 0.447 |
| ETTm1 | 96 | 0.323 | 0.367 | 0.322 | 0.364 | 0.298 | 0.354 |
| | 192 | 0.344 | 0.383 | 0.353 | 0.381 | 0.338 | 0.380 |
| | 336 | 0.375 | 0.403 | 0.388 | 0.401 | 0.376 | 0.401 |
| | 720 | 0.446 | 0.445 | 0.440 | 0.432 | 0.416 | 0.427 |
| | Average | 0.372 | 0.400 | 0.376 | 0.395 | 0.357 | 0.391 |
| Illness | 24 | 1.719 | 0.799 | 2.280 | 1.034 | 1.990 | 0.909 |
| | 36 | 1.541 | 0.753 | 2.224 | 1.018 | 1.913 | 0.899 |
| | 48 | 1.687 | 0.817 | 2.296 | 1.039 | 2.105 | 0.964 |
| | 60 | 1.944 | 0.880 | 2.364 | 1.052 | 2.423 | 1.033 |
| | Average | 1.723 | 0.812 | 2.291 | 1.036 | 2.108 | 0.951 |
| Weather | 96 | 0.154 | 0.201 | 0.188 | 0.243 | 0.184 | 0.240 |
| | 192 | 0.199 | 0.245 | 0.226 | 0.273 | 0.226 | 0.271 |
| | 336 | 0.265 | 0.292 | 0.270 | 0.302 | 0.271 | 0.303 |
| | 720 | 0.344 | 0.350 | 0.336 | 0.347 | 0.335 | 0.346 |
| | Average | 0.241 | 0.272 | 0.255 | 0.291 | 0.254 | 0.290 |

Table 20: MSE and MAE comparison between LVM architecture and ablation models on TSF benchmark datasets. Red numbers indicate the best performance for each dataset.

### B.7 FULL RESULTS OF RQ6: CAN LVMS CAPTURE TEMPORAL ORDER OF TIME SERIES?

Four kinds of perturbation, **Sf-All**, **Sf-Half**, **Ex-Half** and **Masking**, are applied to the time series to compare the performance drop of LVMs, W/O-LVM, and LVM2ATTN on both TSC and TSF tasks. Table 21 and Table 22 summarize the results. As can be seen, LVMs are more vulnerable to temporal perturbations than the ablation models.

| Model | | LVMs | | | | w/o-LVM | | | | LVM2Attn | | | |
|---|---|---|---|---|---|---|---|---|---|---|---|---|---|
| Dataset | Perturbation | Shuffle All | Shuffle Half | Ex-half | Masking | Shuffle All | Shuffle Half | Ex-half | Masking | Shuffle All | Shuffle Half | Ex-half | Masking |
| UWaveGestureLibrary | Accuracy(%) | 17.1 | 56.2 | 35.9 | 62.8 | 17.1 | 73.4 | 0.9 | 79.4 | 10.9 | 73.1 | 0.9 | 79.3 |
| | Performance Drop | 80.7% | 36.4% | 59.4% | 29.0% | 78.2% | 6.6% | 98.8% | -1.0% | 86.4% | 8.7% | 98.9% | 1.0% |
| SpokenArabicDigits | Accuracy(%) | 15.1 | 68.8 | 9.9 | 57.3 | 48.5 | 84.4 | 17.2 | 93.4 | 47.7 | 85.3 | 17.2 | 93.0 |
| | Performance Drop | 84.7% | 30.2% | 89.9% | 41.8% | 49.7% | 12.4% | 82.2% | 3.1% | 50.6% | 11.6% | 82.2% | 3.6% |
| Handwriting | Accuracy(%) | 3.1 | 4.9 | 1.1 | 16.0 | 4.1 | 5.7 | 3.7 | 17.4 | 2.1 | 3.4 | 2.7 | 16.5 |
| | Performance Drop | 91.5% | 86.5% | 97.0% | 56.0% | 81.7% | 74.6% | 83.5% | 22.3% | 89.9% | 83.6% | 87.0% | 20.3% |
| FaceDetection | Accuracy(%) | 47.7 | 61.1 | 61.2 | 62.4 | 51.7 | 57.2 | 49.5 | 64.9 | 51.4 | 58.7 | 49.9 | 64.4 |
| | Performance Drop | 29.2% | 9.3% | 9.2% | 7.4% | 19.3% | 10.8% | 22.8% | -1.2% | 22.4% | 11.3% | 24.6% | 2.7% |

Table 21: Comparison of accuracy (%) and performance drop (%) between LVMs and the ablation models under temporal perturbations on the TSC benchmark datasets. Red numbers indicate the largest performance drop for each dataset.

| Model | | LVMs | | | | | | | | w/o-LVM | | | | | | | | LVM2ATTN | | | | | | | |
|---|---|---|---|---|---|---|---|---|---|---|---|---|---|---|---|---|---|---|---|---|---|---|---|---|---|
| Perturbation | | Sf-All | | Sf-Half | | Ex-half | | Masking | | Sf-All | | Sf-Half | | Ex-half | | Masking | | Sf-All | | Sf-Half | | Ex-half | | Masking | |
| Dataset | Metrics | MSE | MAE | MSE | MAE | MSE | MAE | MSE | MAE | MSE | MAE | MSE | MAE | MSE | MAE | MSE | MAE | MSE | MAE | MSE | MAE | MSE | MAE | MSE | MAE |
| ETTh1 | 96 | 0.747 | 0.588 | 0.369 | 0.393 | 0.457 | 0.437 | 0.551 | 0.534 | 0.746 | 0.582 | 0.437 | 0.438 | 0.483 | 0.460 | 0.608 | 0.559 | 0.741 | 0.589 | 0.442 | 0.449 | 0.456 | 0.448 | 0.577 | 0.554 |
| | 192 | 0.734 | 0.584 | 0.443 | 0.446 | 0.462 | 0.444 | 0.578 | 0.550 | 0.751 | 0.590 | 0.487 | 0.468 | 0.481 | 0.461 | 0.621 | 0.567 | 0.776 | 0.622 | 0.515 | 0.502 | 0.458 | 0.452 | 0.608 | 0.578 |
| | 336 | 0.733 | 0.595 | 0.486 | 0.469 | 0.453 | 0.442 | 0.612 | 0.577 | 0.736 | 0.591 | 0.503 | 0.479 | 0.470 | 0.460 | 0.625 | 0.574 | 0.769 | 0.626 | 0.537 | 0.504 | 0.468 | 0.467 | 0.643 | 0.602 |
| | 720 | 0.765 | 0.631 | 0.587 | 0.549 | 0.480 | 0.476 | 0.664 | 0.582 | 0.740 | 0.613 | 0.509 | 0.507 | 0.472 | 0.484 | 0.635 | 0.597 | 0.779 | 0.660 | 0.554 | 0.518 | 0.479 | 0.491 | 0.669 | 0.632 |
| | Avg. Drop | 83.8% | 43.5% | 14.5% | 10.4% | 14.2% | 7.6% | 47.5% | 34.2% | 76.2% | 36.4% | 14.4% | 8.5% | 13.0% | 7.1% | 47.3% | 31.8% | 79.7% | 39.7% | 19.5% | 10.3% | 9.1% | 4.0% | 46.0% | 32.3% |
| ETTm1 | 96 | 0.732 | 0.561 | 0.441 | 0.440 | 1.127 | 0.691 | 0.504 | 0.508 | 0.731 | 0.561 | 0.441 | 0.430 | 0.929 | 0.629 | 0.567 | 0.538 | 0.779 | 0.611 | 0.442 | 0.447 | 0.895 | 0.625 | 0.577 | 0.554 |
| | 192 | 0.721 | 0.562 | 0.512 | 0.462 | 1.146 | 0.704 | 0.534 | 0.519 | 0.731 | 0.563 | 0.463 | 0.444 | 0.894 | 0.618 | 0.589 | 0.547 | 0.768 | 0.585 | 0.436 | 0.442 | 0.929 | 0.639 | 0.525 | 0.526 |
| | 336 | 0.736 | 0.568 | 0.522 | 0.492 | 1.163 | 0.724 | 0.552 | 0.533 | 0.731 | 0.568 | 0.485 | 0.457 | 0.895 | 0.622 | 0.586 | 0.547 | 0.730 | 0.569 | 0.464 | 0.454 | 0.873 | 0.622 | 0.552 | 0.537 |
| | 720 | 0.780 | 0.587 | 0.556 | 0.526 | 1.221 | 0.745 | 0.570 | 0.547 | 0.753 | 0.582 | 0.529 | 0.484 | 0.919 | 0.636 | 0.616 | 0.562 | 0.772 | 0.721 | 0.743 | 0.585 | 0.939 | 0.656 | 0.771 | 0.628 |
| | Avg. Drop | 118.4% | 53.0% | 48.2% | 28.4% | 242.3% | 92.2% | 58.4% | 41.4% | 98.4% | 44.6% | 28.3% | 15.2% | 145.3% | 59.3% | 58.5% | 39.5% | 117.1% | 59.3% | 44.8% | 23.2% | 158.3% | 63.4% | 70.3% | 44.0% |
| Illness | 24 | 4.794 | 1.578 | 2.426 | 1.064 | 2.465 | 1.045 | 4.169 | 1.386 | 5.220 | 1.674 | 3.091 | 1.251 | 2.529 | 1.098 | 4.394 | 1.507 | 4.712 | 1.613 | 3.449 | 1.287 | 2.942 | 1.219 | 4.768 | 1.572 |
| | 36 | 4.719 | 1.572 | 2.240 | 1.006 | 2.256 | 0.995 | 4.128 | 1.372 | 4.966 | 1.634 | 3.181 | 1.281 | 2.505 | 1.095 | 4.388 | 1.486 | 4.240 | 1.523 | 3.517 | 1.132 | 2.648 | 1.136 | 4.683 | 1.533 |
| | 48 | 4.665 | 1.561 | 2.108 | 0.964 | 2.157 | 0.974 | 4.113 | 1.373 | 4.685 | 1.583 | 3.240 | 1.294 | 2.487 | 1.089 | 4.428 | 1.480 | 4.179 | 1.515 | 3.615 | 1.359 | 2.463 | 1.070 | 4.689 | 1.540 |
| | 60 | 5.094 | 1.622 | 2.138 | 0.962 | 2.161 | 0.999 | 4.374 | 1.422 | 4.947 | 1.632 | 3.464 | 1.335 | 2.648 | 1.129 | 4.574 | 1.521 | 4.349 | 1.523 | 3.597 | 1.352 | 2.629 | 1.099 | 4.940 | 1.578 |
| | Avg. Drop | 162.8% | 79.5% | 21.3% | 13.2% | 23.0% | 13.7% | 128.9% | 57.4% | 116.4% | 57.5% | 41.6% | 24.6% | 11.0% | 6.5% | 94.1% | 44.7% | 109.1% | 62.9% | 69.3% | 34.8% | 27.9% | 19.5% | 127.8% | 64.0% |
| Weather | 96 | 0.258 | 0.316 | 0.162 | 0.211 | 0.329 | 0.351 | 0.278 | 0.371 | 0.261 | 0.312 | 0.189 | 0.246 | 0.295 | 0.329 | 0.290 | 0.379 | 0.261 | 0.313 | 0.189 | 0.246 | 0.298 | 0.331 | 0.290 | 0.380 |
| | 192 | 0.283 | 0.329 | 0.206 | 0.249 | 0.336 | 0.354 | 0.296 | 0.371 | 0.291 | 0.331 | 0.230 | 0.276 | 0.320 | 0.342 | 0.315 | 0.392 | 0.288 | 0.329 | 0.232 | 0.280 | 0.315 | 0.339 | 0.315 | 0.393 |
| | 336 | 0.318 | 0.349 | 0.260 | 0.291 | 0.358 | 0.365 | 0.327 | 0.398 | 0.320 | 0.349 | 0.286 | 0.319 | 0.334 | 0.351 | 0.346 | 0.411 | 0.319 | 0.347 | 0.278 | 0.312 | 0.339 | 0.353 | 0.339 | 0.405 |
| | 720 | 0.396 | 0.411 | 0.363 | 0.357 | 0.391 | 0.388 | 0.384 | 0.439 | 0.370 | 0.380 | 0.341 | 0.354 | 0.382 | 0.382 | 0.376 | 0.428 | 0.371 | 0.381 | 0.341 | 0.353 | 0.387 | 0.385 | 0.376 | 0.428 |
| | Avg. Drop | 44.50% | 38.97% | 9.57% | 7.44% | 67.20% | 45.88% | 49.58% | 57.17% | 24.06% | 18.68% | 2.43% | 2.49% | 33.98% | 21.74% | 33.42% | 39.75% | 24.43% | 19.11% | 2.44% | 2.70% | 35.49% | 22.70% | 33.58% | 40.18% |

Table 22: Comparison of performance drop (%) between the LVM and its ablated variants under time series perturbations on the TSF benchmark datasets. Red numbers indicate the largest performance drop among the three variants.

## B.8 FULL RESULTS OF RQ7: WHAT ARE THE COMPUTATIONAL COSTS OF LVMS?

Fig. 12 presents the accuracy and inference efficiency comparison between LVMs and the two best-performing baselines on TSC task. Fig. 13 (Fig. 14) presents the MSE (MAE) and inference efficiency comparisons between LVMs and the two best-performing baselines on TSF task. In general, LVMs can yield improved performance with higher costs of inference time.

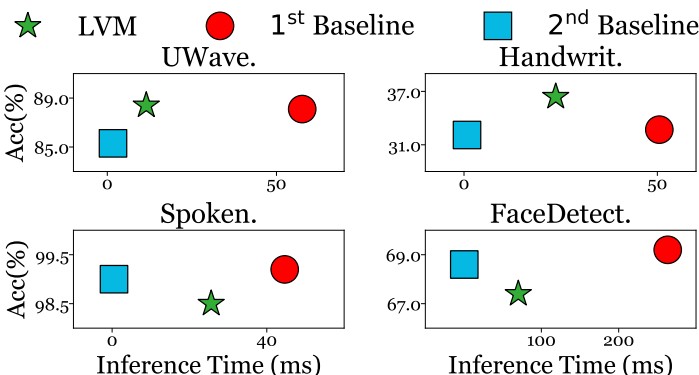

Figure 12: Accuracy *vs.* inference time of the compared methods on TSC benchmark datasets. Green marker stands for LVM, Red marker stands for GPT4TS and Blue marker stands for TimesNet.

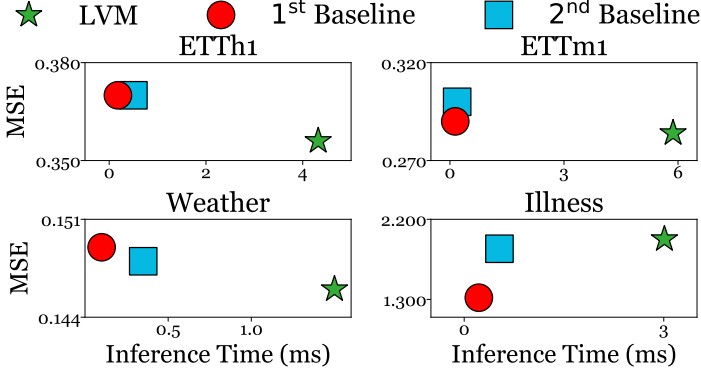

Figure 13: MSE *vs.* inference time of the compared methods on TSF benchmark datasets. Green marker stands for LVM, Red marker stands for PatchTST and Blue marker stands for GPT4TS.

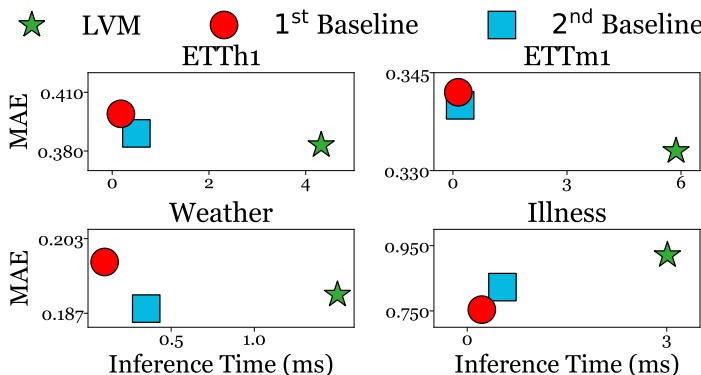

Figure 14: MAE *vs.* inference time of the compared methods on TSF benchmark datasets. Green marker stands for LVM, Red marker stands for PatchTST and Blue marker stands for GPT4TS.

### B.9 FULL RESULTS OF RQ8: WHICH COMPONENT OF LVMs CONTRIBUTES MORE TO FORECASTING

Table 23 provides the detailed results on MSE and MAE of the two ablations, **Enc w/o Dec** and **Dec w/o Enc**, of self-supervised LVMs on TSF benchmark datasets. From Table 23, **Enc w/o Dec** shows inferior performance to **Dec w/o Enc**, highlighting the importance of the pre-trained decoders of LVMs in TSF.

| Model | | MAE | | | | | | SimMIM | | | | | |
|---|---|---|---|---|---|---|---|---|---|---|---|---|---|
| | | Pre-trained | | Enc w/o Dec | | Dec w/o Enc | | Pre-trained | | Enc w/o Dec | | Dec w/o Enc | |
| Dataset | Metrics | MSE | MAE | MSE | MAE | MSE | MAE | MSE | MAE | MSE | MAE | MSE | MAE |
| ETTh1 | 96 | 0.356 | 0.383 | 0.420 | 0.423 | 0.396 | 0.401 | 0.362 | 0.383 | 0.466 | 0.426 | 0.412 | 0.418 |
| | 192 | 0.395 | 0.406 | 0.445 | 0.446 | 0.399 | 0.414 | 0.407 | 0.412 | 0.496 | 0.455 | 0.457 | 0.446 |
| | 336 | 0.417 | 0.424 | 0.489 | 0.484 | 0.441 | 0.433 | 0.422 | 0.417 | 0.499 | 0.474 | 0.581 | 0.520 |
| | 720 | 0.467 | 0.463 | 0.582 | 0.543 | 0.426 | 0.451 | 0.462 | 0.455 | 0.505 | 0.481 | 0.564 | 0.526 |
| | Average | 0.409 | 0.419 | 0.484 | 0.474 | 0.416 | 0.425 | 0.413 | 0.417 | 0.492 | 0.459 | 0.504 | 0.478 |
| ETTm1 | 96 | 0.284 | 0.333 | 0.324 | 0.363 | 0.295 | 0.335 | 0.311 | 0.350 | 0.320 | 0.347 | 0.299 | 0.348 |
| | 192 | 0.328 | 0.363 | 0.361 | 0.387 | 0.330 | 0.364 | 0.335 | 0.367 | 0.377 | 0.377 | 0.344 | 0.378 |
| | 336 | 0.357 | 0.384 | 0.398 | 0.414 | 0.365 | 0.388 | 0.356 | 0.382 | 0.411 | 0.401 | 0.403 | 0.419 |
| | 720 | 0.411 | 0.417 | 0.446 | 0.440 | 0.409 | 0.416 | 0.400 | 0.413 | 0.468 | 0.442 | 0.431 | 0.433 |
| | Average | 0.345 | 0.374 | 0.382 | 0.401 | 0.350 | 0.376 | 0.351 | 0.378 | 0.394 | 0.392 | 0.369 | 0.395 |
| Illness | 24 | 1.977 | 0.921 | 1.946 | 0.842 | 1.774 | 0.841 | 1.934 | 0.902 | 2.314 | 0.944 | 2.034 | 0.899 |
| | 36 | 1.812 | 0.872 | 1.981 | 0.895 | 1.918 | 0.876 | 1.754 | 0.825 | 2.434 | 1.045 | 2.198 | 0.983 |
| | 48 | 1.743 | 0.856 | 1.967 | 0.855 | 2.061 | 0.943 | 1.715 | 0.867 | 2.008 | 0.869 | 2.209 | 0.960 |
| | 60 | 1.816 | 0.881 | 1.956 | 0.858 | 1.969 | 0.950 | 1.673 | 0.877 | 1.979 | 0.865 | 2.275 | 0.997 |
| | Average | 1.837 | 0.883 | 1.963 | 0.863 | 1.931 | 0.903 | 1.769 | 0.868 | 2.184 | 0.931 | 2.179 | 0.960 |
| Weather | 96 | 0.146 | 0.191 | 0.168 | 0.210 | 0.155 | 0.201 | 0.148 | 0.196 | 0.166 | 0.208 | 0.150 | 0.200 |
| | 192 | 0.194 | 0.238 | 0.237 | 0.263 | 0.209 | 0.248 | 0.196 | 0.243 | 0.228 | 0.257 | 0.199 | 0.246 |
| | 336 | 0.243 | 0.275 | 0.299 | 0.306 | 0.274 | 0.298 | 0.244 | 0.276 | 0.294 | 0.297 | 0.251 | 0.284 |
| | 720 | 0.318 | 0.328 | 0.396 | 0.372 | 0.378 | 0.361 | 0.340 | 0.340 | 0.382 | 0.357 | 0.343 | 0.342 |
| | Average | 0.225 | 0.258 | 0.275 | 0.288 | 0.254 | 0.277 | 0.232 | 0.264 | 0.268 | 0.280 | 0.236 | 0.268 |

Table 23: MSE and MAE comparison of self-supervised LVMs with either the pre-trained encoder (**Dec w/o Enc**) or decoder (**Enc w/o Dec**) excluded on TSF benchmark datasets.

### B.10 FULL RESULTS OF RQ9: WILL PERIOD-BASED IMAGING METHOD INDUCE ANY BIAS?

Fig. 15 provides the forecasting performance of an LVM (*i.e.*, MAE) in terms of metrics MAE *w.r.t.* segment length that varies from $\frac{1}{6}L$ to $\frac{12}{6}L$. The LVM generally achieves the best performance when segment length is a multiple of the period, *i.e.* $L$ or $2L$, which is caused by the inductive bias as discussed in RQ8 In §4.3.

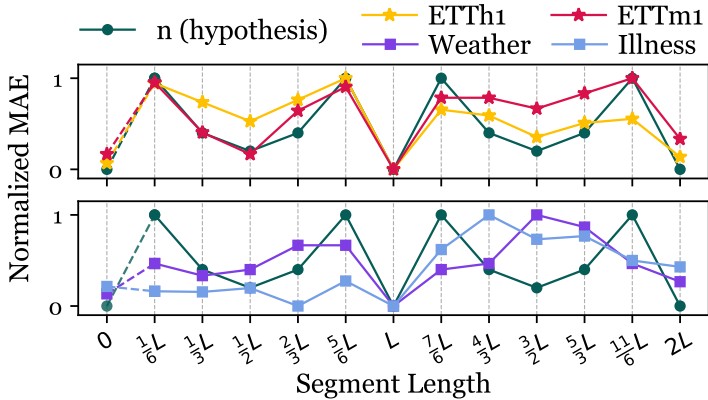

Figure 15: Forecasting performance (MAE) of an LVM *w.r.t.* varying segment length used in UVH imaging. $n$ (green) estimates the difficulty of forecasting.

## B.11 EVALUATION OF LVMs ON DATASETS WITH WEAK PERIODICITY

In this section, we compare the best LVM forecaster (*i.e.*, `MAE`) with the best non-LVM forecaster (*i.e.*, `PatchTST` according to Table 2) on datasets weak periodicity, including Weather, Illness and two additional datasets – Exchange (Lai et al., 2018) and Solar (Liu et al., 2023b). Fig. 16 visualizes sample time series from these datasets, which demonstrate weak periodicity. Table 24 summarize the results, from which we can observe that `MAE` does not show advantage over the non-LVM baseline on non-periodic datasets. This confirms our analysis of LVMs' bias toward periodicity in RQ9 (§4.3). The findings also call for diversifying benchmark datasets for TSF task.

| Model | Dataset | Weather | | Illness | | Exchange | | Solar | | # Wins |
| --- | --- | --- | --- | --- | --- | --- | --- | --- | --- | --- |
| | Metric | MSE | MAE | MSE | MAE | MSE | MAE | MSE | MAE | |
| MAE | 96 | **0.146** | **0.191** | 1.977 | 0.921 | 0.099 | 0.224 | 0.190 | **0.245** | |
| | 192 | 0.194 | **0.238** | 1.812 | 0.872 | **0.199** | 0.321 | 0.206 | **0.257** | |
| | 336 | **0.243** | **0.275** | 1.743 | 0.856 | 0.383 | 0.453 | 0.214 | **0.265** | 12 |
| | 720 | 0.318 | **0.328** | 1.816 | 0.881 | 0.937 | 0.729 | 0.235 | 0.299 | |
| | Avearge | **0.225** | **0.258** | 1.837 | 0.883 | 0.405 | 0.432 | 0.211 | 0.267 | |
| PatchTST | 96 | 0.149 | 0.198 | **1.319** | **0.754** | **0.092** | **0.213** | **0.185** | 0.251 | |
| | 192 | **0.193** | 0.241 | **1.430** | **0.834** | 0.207 | **0.235** | **0.194** | 0.263 | |
| | 336 | 0.245 | 0.282 | **1.553** | **0.815** | **0.376** | **0.451** | **0.213** | 0.274 | 29 |
| | 720 | **0.314** | 0.334 | **1.470** | **0.788** | **0.858** | **0.692** | **0.213** | 0.275 | |
| | Avearge | **0.225** | 0.264 | **1.443** | **0.798** | **0.383** | **0.398** | **0.201** | **0.266** | |

Table 24: MSE and MAE comparison of LVM forcaster and non-LVM forecaster on TSF datasets with weak periodicity.

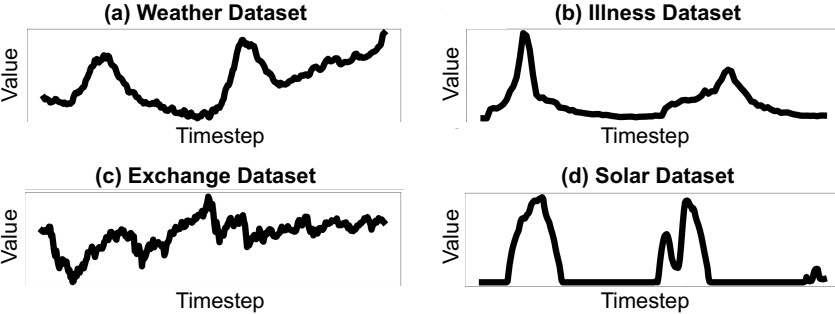

Figure 16: An illustration of random samples in a look-back window (length=336) from (a) Weather dataset; (b) Illness dataset; (c) Exchange dataset; and (d) Solar dataset, which have weak periodicity.

## B.12 FULL RESULTS OF RQ10: CAN LVMs MAKE EFFECTIVE USE OF LOOK-BACK WINDOWS?

Table 25 presents the MSE and MAE performance of LVMs across varying look-back window lengths, ranging from 48 to 2304. As discussed in RQ9, LVMs exhibits limited ability in fully leveraging the information of look-back window when the window length exceeds approximately 1000 time steps. The Illness dataset is omitted in Table 25 because its time series are of short lengths, with only 966 time steps in total.

## B.13 COMPARISON WITH LATEST DIFFUSION-BASED LVM

Table 26 and Table 27 present comparisons with the latest diffusion LVM (Wang et al., 2025b) on both TSC and TSF tasks. For each task, the best basic LVM is used to compare with `LaVin-DiT`. The results show no clear advantage over the general LVMs applied in this paper.

## C PROOF OF LEMMA 1

In this section, we provide the proof for Lemma 1.

| Look-back Window | | 48 | | 96 | | 192 | | 336 | | 720 | | 1152 | | 1728 | | 2304 | |
|---|---|---|---|---|---|---|---|---|---|---|---|---|---|---|---|---|
| Dataset | Metrics | MSE | MAE | MSE | MAE | MSE | MAE | MSE | MAE | MSE | MAE | MSE | MAE | MSE | MAE | MSE | MAE |
| ETTh1 | 96 | 0.376 | 0.395 | 0.373 | 0.390 | 0.364 | 0.383 | 0.356 | 0.383 | 0.347 | 0.375 | 0.347 | 0.376 | 0.344 | 0.376 | 0.373 | 0.402 |
| | 192 | 0.440 | 0.431 | 0.424 | 0.418 | 0.411 | 0.412 | 0.395 | 0.406 | 0.385 | 0.405 | 0.384 | 0.402 | 0.391 | 0.408 | 0.399 | 0.417 |
| | 336 | 0.474 | 0.450 | 0.471 | 0.445 | 0.456 | 0.437 | 0.417 | 0.424 | 0.408 | 0.418 | 0.410 | 0.418 | 0.395 | 0.413 | 0.408 | 0.423 |
| | 720 | 0.485 | 0.477 | 0.482 | 0.471 | 0.469 | 0.465 | 0.467 | 0.463 | 0.468 | 0.460 | 0.432 | 0.440 | 0.425 | 0.442 | 0.424 | 0.442 |
| | Average | 0.444 | 0.438 | 0.438 | 0.431 | 0.425 | 0.424 | 0.409 | 0.419 | 0.402 | 0.415 | 0.393 | 0.409 | 0.389 | 0.410 | 0.401 | 0.421 |
| ETTm1 | 96 | 0.443 | 0.413 | 0.316 | 0.353 | 0.304 | 0.345 | 0.284 | 0.333 | 0.279 | 0.324 | 0.280 | 0.332 | 0.277 | 0.322 | 0.285 | 0.326 |
| | 192 | 0.476 | 0.431 | 0.373 | 0.390 | 0.333 | 0.365 | 0.328 | 0.363 | 0.322 | 0.358 | 0.321 | 0.361 | 0.321 | 0.355 | 0.318 | 0.350 |
| | 336 | 0.512 | 0.457 | 0.385 | 0.400 | 0.370 | 0.390 | 0.357 | 0.384 | 0.356 | 0.381 | 0.362 | 0.383 | 0.352 | 0.378 | 0.346 | 0.374 |
| | 720 | 0.574 | 0.489 | 0.449 | 0.438 | 0.426 | 0.429 | 0.411 | 0.417 | 0.411 | 0.414 | 0.399 | 0.413 | 0.411 | 0.414 | 0.407 | 0.416 |
| | Average | 0.501 | 0.448 | 0.381 | 0.395 | 0.358 | 0.382 | 0.345 | 0.374 | 0.342 | 0.369 | 0.341 | 0.372 | 0.340 | 0.367 | 0.339 | 0.367 |
| Weather | 96 | 0.200 | 0.237 | 0.167 | 0.209 | 0.152 | 0.196 | 0.146 | 0.191 | 0.142 | 0.188 | 0.144 | 0.194 | 0.143 | 0.193 | 0.141 | 0.195 |
| | 192 | 0.236 | 0.267 | 0.212 | 0.249 | 0.200 | 0.240 | 0.194 | 0.238 | 0.188 | 0.235 | 0.189 | 0.237 | 0.195 | 0.242 | 0.200 | 0.253 |
| | 336 | 0.293 | 0.307 | 0.268 | 0.290 | 0.254 | 0.280 | 0.243 | 0.275 | 0.247 | 0.281 | 0.242 | 0.279 | 0.272 | 0.302 | 0.278 | 0.307 |
| | 720 | 0.370 | 0.358 | 0.346 | 0.340 | 0.330 | 0.333 | 0.318 | 0.328 | 0.334 | 0.341 | 0.332 | 0.339 | 0.344 | 0.349 | 0.372 | 0.357 |
| | Average | 0.275 | 0.292 | 0.248 | 0.272 | 0.234 | 0.262 | 0.225 | 0.258 | 0.228 | 0.261 | 0.227 | 0.262 | 0.239 | 0.272 | 0.248 | 0.278 |

Table 25: The MSE and MAE performance of LVMs across different look-back window lengths on TSF benchmark datasets.

| Dataset | ViT | LaVin-DiT |
|---|---|---|
| UWaveGestureLibrary | 88.4 | 84.2 |
| SpokenArabicDigits | 98.5 | 97.9 |
| Handwriting | 36.4 | 36.7 |
| FaceDetection | 67.4 | 67.0 |
| Average | 72.6 | 71.5 |

Table 26: The comparison on performance of diffusion-based LVM on TSC benchmark datasets

| | MAE | | LaVin-DiT | |
|---|---|---|---|---|
| Dataset | MSE | MAE | MSE | MAE |
| ETTh1 | 0.409 | 0.419 | 0.403 | 0.416 |
| ETTm1 | 0.345 | 0.374 | 0.349 | 0.377 |
| Weather | 0.225 | 0.258 | 0.231 | 0.259 |
| Illness | 1.837 | 0.883 | 1.733 | 0.859 |

Table 27: The comparison on performance of diffusion-based LVM on TSF benchmark datasets

*Proof.* Given $\mathbf{x}$ is perfectly periodic, $x_t = x_{t+\alpha \cdot L}$ holds when $\alpha \in \mathbb{N}^+$ and $L$ is the period. The smallest number of segments $n$ before any segment reoccurs, *i.e.*, $\mathbf{x}_t = \mathbf{x}_{t+n\cdot(i/k)L}$, indicates $n \cdot (i/k) \in \mathbb{N}^+$. Hence, the proof of Lemma 1 is equivalent to prove $n = \frac{k}{\mathrm{GCD}(i,k)}$ as the smallest natural number such that $k$ divides $n \cdot i$, denoted as $k \mid n \cdot i$.

Set $d = \mathrm{GCD}(i,k)$ as the greatest common divisor of $i$ and $k$. The following is based on the definition of greated common divisor:

$$i = d \cdot i' \tag{3}$$
$$k = d \cdot k' \tag{4}$$
$$\mathrm{GCD}(i', k') = 1 \tag{5}$$

where $i', k' \in \mathbb{N}^+$. As $k$ divides $n \cdot i$, we have

$$
\begin{aligned}
k \mid n \cdot i &\Rightarrow d \cdot k' \mid d \cdot n \cdot i' \\
&\Rightarrow k' \mid n \cdot i' \\
&\Rightarrow k' \mid n
\end{aligned}
\tag{6}
$$

The first step in Eq. equation 6 is expanded with Eq. equation 3 and Eq. equation 4. The second step cancels the common factor $d$ from both sides of with the divisibility relation unchanged. The last step follows Eq. equation 5. To satisfy Eq. equation 6, the smallest $n$ is $n = k'$. Finally, expand $k'$ with

Eq. equation 4, we reach

$$n = k' = \frac{k}{d} = \frac{k}{\text{GCD}(i, k)}$$

□

# D   VISUALIZATION RESULTS

## D.1   VISUALIZATION OF GAF ON TSC TASK

To have a sense about what temporal patterns can be recognized by LVMs for TSC, we visualize the images of GAF method on the Handwriting and UWaveGestureLibrary datasets in Fig. 17 and Fig. 18, respectively. The examples are randomly sampled from five different classes on both datasets. From Fig. 17 and Fig. 18, we can observe clear visual patterns that distinguish the GAF images from different classes, which highlight the effectiveness of GAF as a way to encode time series for LVMs to process for TSC.

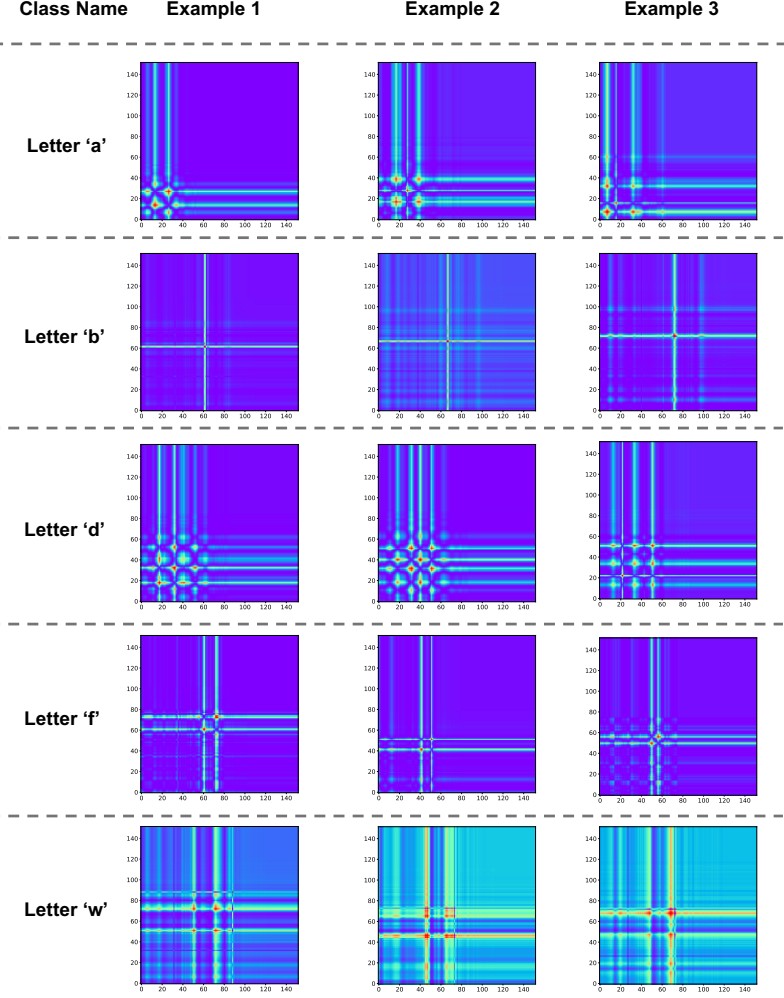

Figure 17: Examples of GAF images on the first channel of multivariate time series with 152 time steps randomly drawn from five classes in the Handwriting dataset.

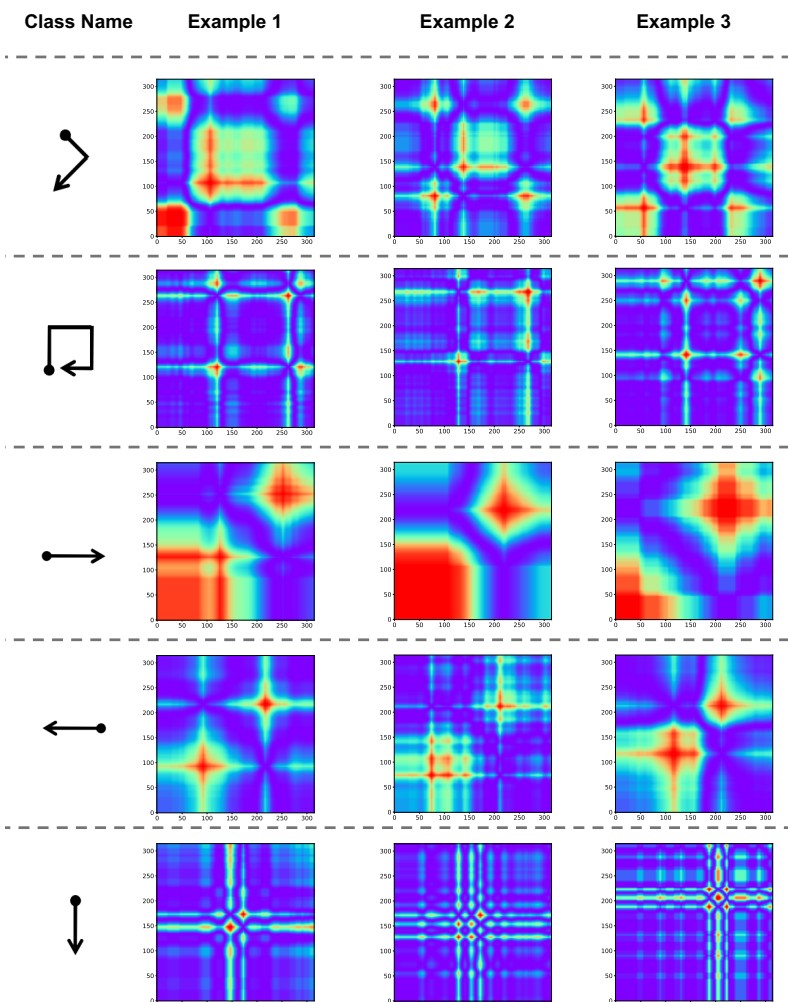

Figure 18: Examples of GAF images on the first channel of multivariate time series with 336 time steps randomly drawn from five classes in the UWaveGestureLibrary dataset.

## D.2  ILLUSTRATION OF AN INDUCTIVE BIAS OF LVMS DURING TSF

As discussed in RQ8, the imaging method UVH can induce an inductive bias to LVMs in TSF toward "forecasting periods" by rendering them to combine the past segments to infer future. To illustrate this, Fig. 19 and Fig. 20 visualize two random examples with varying segment lengths from one period (24 time steps) to two periods (48 time steps) from ETTh1 and Traffic datasets. The blue lines represent the time series in look-back window, the red lines represent the ground truth in prediction horizon, and the green lines represent the forecasted time series by LVMs. The results demonstrate that LVMs perform best when the segment length aligns with the period of the time series, while the performance degrades when the segment length shifts from the period. This implies the inductive bias of combining the past periods as forecasts by LVMs with UVH for TSF.

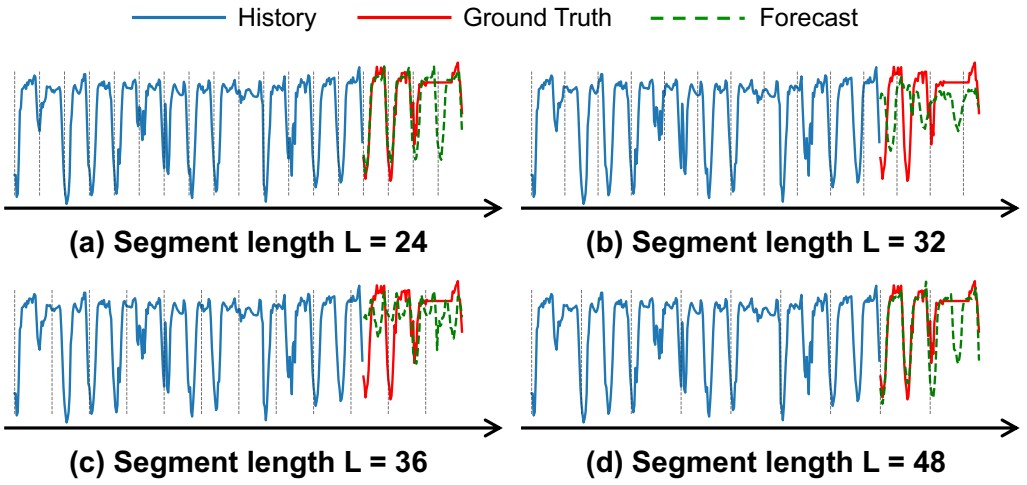

Figure 19: Visualization of LVM's inductive bias during TSF on a random example from the ETTh1 dataset (period is 24 time steps). From (a) to (d), the segment length vary within {24, 32, 36, 48}.

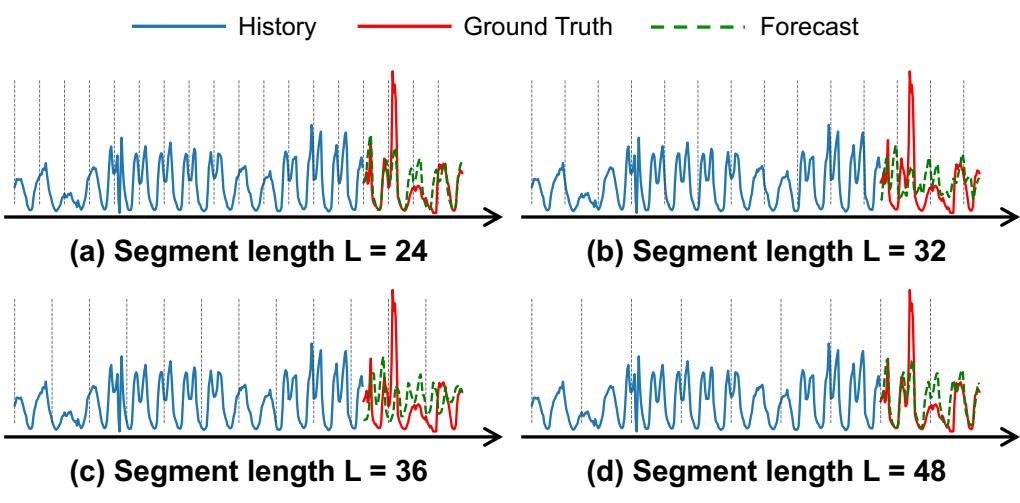

Figure 20: Visualization of LVM's inductive bias during TSF on a random example from the Traffic dataset (period is 24 time steps). From (a) to (d), the segment length vary within {24, 32, 36, 48}.

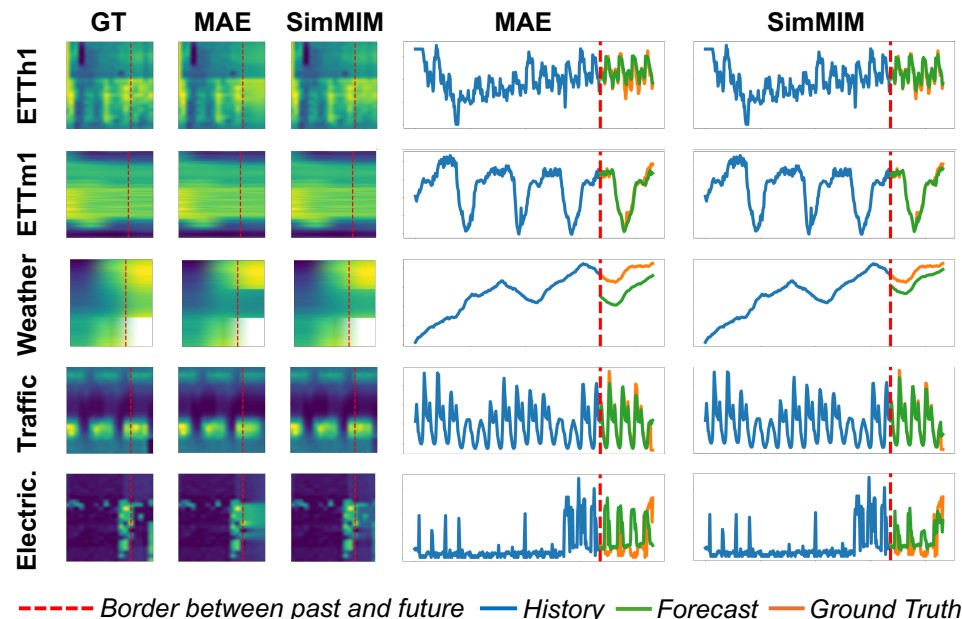

Figure 21: An illustration of the reconstructed images of `MAE`'s decoder and `SimMIM`'s decoder on five TSF benchmark datasets. The red dashed lines separate look-back windows and forecasting horizons. The first column shows the ground truth forecasts. The second and third columns show `MAE`'s forecasts and `SimMIM`'s forecasts, respectively. The fourth and fifth columns show the recovered time series of `MAE`'s forecasts and `SimMIM`'s forecasts, respectively.

## D.3 ANALYSIS OF LVM FORECASTERS' DECODERS

In this section, we analyze the reconstruction abilities of `MAE`'s decoder and `SimMIM`'s decoder. The two encoder-decoder LVMs have different decoders – `MAE` has a 8-layer Transformer decoder and `SimMIM` has a single-layer linear decoder. In Fig. 21, we visualize the reconstructed images of both `MAE` and `SimMIM`, and compare the reconstructions with the ground truths on random samples from 5 TSF benchmark datasets – ETTh1, ETTm1, Weather, Traffic, and Electricity. In Fig. 21, the red dashed lines separate the look-back windows and the forecasts. In addition to the UVH images, we visualize the recovered time series from the images. From Fig. 21, we obverse the reconstruction ability of LVMs' decoders on the masked areas (which corresponds to the forecasting horizons), which tend to be smooth across columns. This confirms our analysis of LVMs' bias toward periodicity in RQ9 (§4.3). It also illustrates the working mechanism of LVMs' decoders in TSF task.

## E  FURTHER DISCUSSIONS

**Related Works on Multimodal Methods for Time Series Analysis**. Among the recent VLM-based methods for time series analysis Wimmer & Rekabsaz (2023); Prithyani et al. (2024); Zhuang et al. (2024); Zhong et al. (2025); Shen et al. (2025), the most relevant includes `Time-VLM` (Zhong et al., 2025) and `DMMV` (Shen et al., 2025). `Time-VLM` builds a forecaster on `ViLT` (Kim et al., 2021) to encode numerical and visual views, along with contextual texts. While integrating rich information with a large model, `Time-VLM` demonstrates promising results in TSF. However, its fusion strategy closely follows the `ViLT` backbone and lacks time-series-specific design, leading to potentially suboptimal performance. `DMMV` integrates LVMs and numerical forecasters (*e.g.*, Transformer) in an adaptive decomposition framework to form a multimodal architecture, which aims to mitigate the bias of LVM forecasters. A recent survey (Jiang et al., 2025) provides a structured discussion on multimodal methods for time series analysis. However, existing works lack fundamental understandings of the effectiveness of sole LVMs in time series analysis. The goal of this work is to fill this gap.

**Further Analysis for RQ8.** The ablation in Fig. 7 validates the importance of LVMs' decoders in forecasting. Our understanding is that this is because LVMs' decoders aim to reconstruct pixel values, while the encoders aim to extract general-purpose features. For a forecasting task, reconstructing pixel values align more with forecasting the numerical values in a time series, thus plays a more important role than the encoders.

**Further Analysis for RQ10.** From Fig. 10, the ineffective use of look-back windows may result from image transformation: fixed-size image in pre-trained LVMs has a pixel limit and may constrain the information captured from excessively long time series. Additionally, the pixels in the fixed-size image are not fully utilized by the SOTA LVM forecasters. In their imaging setup, each column of pixels (i.e., $256 \times 1$) represents an interpolated period with P timesteps (*e.g.*, $P = 24$ for ETTh1), less than 256 timesteps. Also, following the masking strategy in VisionTS (Chen et al., 2025), an alignment constant $c = 0.4$ is applied, which means that over 60% of the pixels are masked. As a result, only about $P \times 0.4 \times 256$ time steps can be effectively used in one image. For example, when $P = 24$, only around 2,400 timesteps can be encoded, which is significantly fewer than the full $256 \times 256$ pixels. Thus the amount of effective pixels is influenced by $P$ and $c$. Other factors such noises and repeated patterns (*i.e.*, redundancy) may further reduce the amount of effective pixels that can inform forecasting. This raises a future direction on how to better utilize image pixels for time series forecasting.

## F ANALYSIS OF DIFFERENT IMAGING METHODS

### F.1 EFFECTIVENESS OF DIFFERENT IMAGING METHODS

Different imaging techniques may lead to different successes of LVMs in time series analysis. This relates to (1) the property of each imaging technique; and (2) the effectiveness of the property when using LVMs.

First, the key property of each imaging technique can be deterministically identified from its definitions as specified in (Ni et al., 2025). In Fig. 22 (and Fig. 1(a)), we've empirically visualized these methods. In the following, we summarize the key property of each imaging technique according to the formal (mathematical) definitions in (Ni et al., 2025). We'd like to refer interested readers to (Ni et al., 2025) for the detailed definitions.

- **Line Plot** (*e.g.*, Fig. 22(a)) uses a line to represent a time series in a 2D image. Its use of pixels is ineffective compared to other imaging techniques because most of the pixels are used for representing the white background.

- **MVH** (*e.g.*, Fig. 22(b)) is the only method that directly visualize all variates in an MTS in a single heatmap image. However, there is no principled way to determine the order of variates on the y-axis. Different orders may lead to different cross-variate patterns learned by an LVM.

- **UVH** (*e.g.*, Fig. 22(c)) visualizes stacked segments of a time series as a heatmap image. As such, it may lead to a bias toward periodicity when the segments are periods.

- **STFT** (*e.g.*, Fig. 22(e)) visualizes the time-frequency space of a time series. However, it uses a fixed-size sliding window which cannot fit varying frequencies in a time series.

- **Wavelet** (*e.g.*, Fig. 22(f)) visualizes the time-frequency space of a time series using wavelets. It addresses STFT's limitation but needs a proper choice of wavelet function.

- **Filterbank** (*e.g.*, Fig. 22(g)) resembles STFT but is associated with several processing filters on a Mel-scale that fit audio signals thus is more often used for imaging audio signals.

- **GAF** (*e.g.*, Fig. 22(d)) is a square $T$-by-$T$ image ($T$ is the length of look-back window). It is good at capturing the temporal correlations and cyclical patterns in a time series.

- **RP** (*e.g.*, Fig. 22(h)) is also a square image but its size is tunable by its hyperparameters. It is good at capturing periodic patterns but is a black-white image thus may lose some fine-grained information.

Among these methods, only UVH and MVH images encode the raw (or normalized) time series values, while other methods visualize different transformations of time series values. This will lead to their different performance in a forecasting (numerical-level) task. Additionally, STFT, Wavelet, and

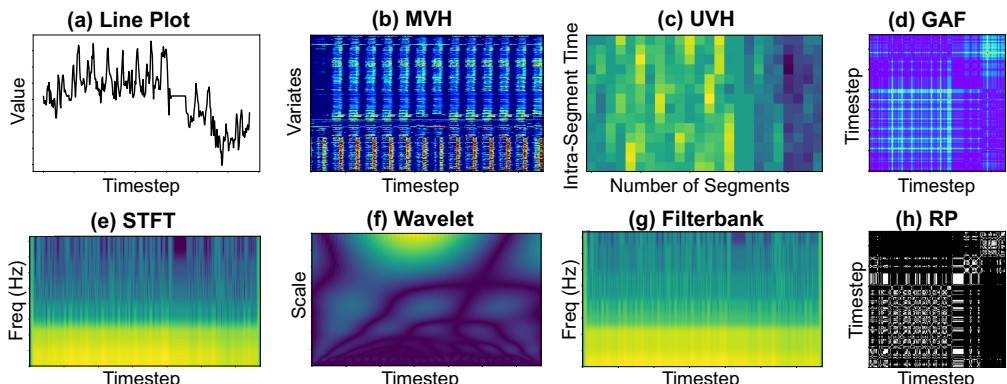

Figure 22: An illustration of different methods for imaging a time series sample (length=336) from the *Electricity* benchmark dataset.

Filterbank produce spectograms. Their difference lies in their different ways of encoding frequencies in a time series.

Second, our experiments for comparing the 8 imaging methods provide insights on the effectiveness of these methods when using LVMs. For classification, the averaged results in Table 11 (the bottom row) suggest that Line Plot are Filterbank are less effective than other methods. This is because of Line Plot's ineffective use of pixels and Filterbank's filter processing that is aligned with high-frequency audio signals. For the other 6 methods, despite their different levels of effectiveness, they all achieve reasonable averaged accuracy ($\sim$70%), suggesting their effectiveness in capturing distinguishable patterns (*e.g.*, GAF, RP, UVH capture periodic patterns; STFT, Wavelet capture time-frequency patterns; MVH captures overall trend patterns).

For forecasting, the results in Table 12 suggest that UVH and MVH are better than other methods. This is because forecasting task requires predicting future time series values. Among the 8 methods, only UVH and MVH encode historical time series values, thus they are more effective than other transformation methods in a numerical-level task, *i.e.*, forecasting. However, MVH underperforms UVH. The limitation of MVH is its mixture of variates in the image without a principled ordering of the variates, which may confuse cross-variate dependency learning.

### F.2 POTENTIAL BIAS OF USING IMAGING METHODS

Transforming time series to an image may introduce some biases. We summarize the key positive bias (pros) and negative bias (cons) of different imaging methods as following. The methods that do not preserve raw time series values (*i.e.*, LinePlot, STFT, Wavelet, Filterbank, GAF, RP) do not fit reconstruction-based LVMs for time series forecasting, leading to their inferior performance in Table 12. From the perspective of LVMs, the key advantage of using imaged time series is that it enables transferring the encoded visual knowledge in LVMs by large-scale pretraining to time series tasks. The key limitations, according to our analysis in RQ1, RQ8 and RQ10, includes (1) ineffective use of window-based local attention; (2) less effective use of encoder; and (3) fixed number of usable pixels (which caps the length of look-back windows).

- **Line Plot** (*e.g.*, Fig. 22(a)): (1) Pros: compact representation of long time series; (2) Cons: pixels are not fully used (large white background pixels).

- **MVH** (*e.g.*, Fig. 22(b)): (1) Pros: direct representation of all variates in a multivariate time series; (2) Cons: random ordering of variates may confuse correlation learning.

- **UVH** (*e.g.*, Fig. 22(c)): (1) Pros: encode periodic patterns of time series; (2) Cons: break continuity across different periods.

- **STFT** (*e.g.*, Fig. 22(e)): (1) Pros: encode time-frequency space; (2) Cons: lose raw time series values (which is important in numerical-level tasks).

- **Wavelet** (*e.g.*, Fig. 22(f)): (1) Pros: encode time-frequency space while fitting varying frequencies; (2) Cons: lose raw time series values.

- **Filterbank** (*e.g.*, Fig. 22(g)): (1) Pros: encode time-frequency space while fitting high-frequency signals: (2) Cons: lose raw time series values.

- **GAF** (*e.g.*, Fig. 22(d)): (1) Pros: encode temporal correlations of pairwise time steps; (2) Cons: lose raw time series values, use more pixels for longer time series ($O(T^2)complexity$)).

- **RP** (*e.g.*, Fig. 22(h)): (1) Pros: encode cyclic patterns of time series; (2) Cons: lose raw time series values, back-white image loses fine-grained information.

## G  COMPARING LVM FORECASTER WITH TSFMS

| Model | Dataset | ETTh1 | | ETTh2 | | ETTm1 | | ETTm2 | | Weather | | Illness | | Traffic | | Electricity | | # Wins |
|---|---|---|---|---|---|---|---|---|---|---|---|---|---|---|---|---|---|---|
| | Metric | MSE | MAE | MSE | MAE | MSE | MAE | MSE | MAE | MSE | MAE | MSE | MAE | MSE | MAE | MSE | MAE | |
| MAE | 96 | 0.356 | 0.383 | 0.297 | 0.341 | 0.284 | 0.333 | 0.173 | 0.258 | 0.146 | 0.191 | 1.977 | 0.921 | 0.346 | 0.232 | 0.127 | 0.217 | |
| | 192 | 0.395 | 0.406 | 0.356 | 0.386 | 0.328 | 0.363 | 0.231 | 0.297 | 0.194 | 0.238 | 1.812 | 0.872 | 0.376 | 0.245 | 0.148 | 0.237 | |
| | 336 | 0.417 | 0.424 | 0.371 | 0.402 | 0.357 | 0.384 | 0.282 | 0.340 | 0.243 | 0.275 | 1.743 | 0.856 | 0.389 | 0.252 | 0.163 | 0.253 | 51 |
| | 720 | 0.467 | 0.463 | 0.403 | 0.430 | 0.411 | 0.417 | 0.386 | 0.413 | 0.318 | 0.328 | 1.816 | 0.881 | 0.432 | 0.293 | 0.199 | 0.293 | |
| | Average | 0.409 | 0.419 | 0.357 | 0.390 | 0.345 | 0.374 | 0.268 | 0.327 | 0.225 | 0.258 | 1.837 | 0.883 | 0.386 | 0.256 | 0.159 | 0.250 | |
| LightGTS | 96 | 0.346 | 0.382 | 0.271 | 0.369 | 0.291 | 0.338 | 0.165 | 0.242 | 0.146 | 0.173 | 3.001 | 1.205 | 0.406 | 0.316 | 0.144 | 0.237 | |
| | 192 | 0.398 | 0.419 | 0.338 | 0.378 | 0.362 | 0.378 | 0.229 | 0.289 | 0.197 | 0.219 | 2.977 | 1.231 | 0.423 | 0.318 | 0.160 | 0.257 | |
| | 336 | 0.403 | 0.423 | 0.359 | 0.401 | 0.416 | 0.419 | 0.295 | 0.346 | 0.243 | 0.257 | 3.110 | 1.295 | 0.439 | 0.326 | 0.183 | 0.283 | 33 |
| | 720 | 0.427 | 0.450 | 0.386 | 0.420 | 0.599 | 0.489 | 0.381 | 0.397 | 0.310 | 0.301 | 3.008 | 1.289 | 0.487 | 0.354 | 0.260 | 0.339 | |
| | Average | 0.394 | 0.419 | 0.339 | 0.392 | 0.417 | 0.406 | 0.268 | 0.319 | 0.224 | 0.238 | 3.024 | 1.255 | 0.439 | 0.329 | 0.187 | 0.279 | |

Table 28: MSE and MAE comparison of LVM forecaster and a TSFM on TSF benchmark datasets.

In this section, we compare the LVM forecaster (*i.e.* MAE) with a time series foundation model (TSFM) – LightGTS (Wang et al., 2025a), which was pre-trained on large-scale time series datasets while is more parameter-efficient than some existing TSFMs such as Chronos Ansari et al. (2025) and Moirai Morid et al. (2023). Meanwhile, LightGTS was demonstrated to outperform TSFMs with larger sizes on TSF benchmark datasets in (Wang et al., 2025a). Table 28 summarizes the comparison on the 8 TSF benchmark datasets. From Table 28, we observe LVM's strong potential in transferring cross-modal knowledge, which appears to be more useful than the intra-modal knowledge in LightGTS (obtained by pre-training on time series datasets). We think this advantage attributes to the much larger size of the available pre-training image datasets than that of the pre-training time series datasets collected by existing methods. The results also suggest a potential research direction of exploring the integration of cross-modal knowledge and intra-modal knowledge for enhancing performance.

