# OpenReview forum: "From Images to Signals: Are Large Vision Models Useful for Time Series Analysis?"
_ICLR.cc/2026/Conference — Submitted to ICLR 2026_

### Official Review · Reviewer_1Fj9 · 2025-10-24

**Soundness:** 2
**Presentation:** 2
**Contribution:** 2
**Rating:** 4
**Confidence:** 4

**Summary:**

This paper benchmarks LVMs for time-series analysis by converting time-series data into image representations. It systematically evaluates 4 LVMs (ViT, Swin, MAE, SimMIM), 8 imaging techniques, and 18 datasets across classification and forecasting tasks. The authors analyze pretraining paradigms, imaging methods, decoder vs. encoder roles, and context length sensitivity.

**Strengths:**

1. First comprehensive study to jointly analyze LVMs across TSC and TSF using diverse image encodings.
2. Provides a valuable reference for researchers exploring multimodal or vision-inspired time-series models.

**Weaknesses:**

1. Most of the forecasting results rely on UVH/MVH imaging applied to relatively periodic datasets. It’s not clear how well these conclusions hold for more irregular, noisy, or non-periodic signals.
2. The claim that decoders matter more than encoders in forecasting is interesting, but the current evidence is mostly correlational. A simple decoder-swapping or partial-finetuning experiment could make this stronger.
3. The authors note that performance drops for longer look-back windows, potentially due to image resolution limits, but this isn’t experimentally verified. It’s unclear whether the weaknesses observed (e.g., poor long-context forecasting) are intrinsic to LVMs or simply due to architectural mismatch, because the approach reuses vision models mostly as-is, with only changes to heads or decoders.
4. The central assumption that converting time series into 2D image structures enables vision transformers to generalize to sequential modeling is interesting but not fully justified. Time and space are fundamentally different dimensions; the study doesn’t discuss what inductive biases are gained or lost by this transformation. For example, local temporal continuity is often disrupted in GAF/UVH encodings, and the benefit of spatial locality learned by ViTs is not clearly transferable to time-dependent modeling.
5. The paper tests many imaging techniques (GAF, RP, UVH, etc.), but it remains unclear why certain encodings align better with LVM features. There’s little analysis of the representational geometry, e.g., whether these encodings preserve correlations, periodicities, or value scales in a meaningful way. The success of UVH/MVH feels more empirical.

**Questions:**

1. How robust are the UVH/MVH findings to non-periodic datasets or those with exogenous inputs (e.g., with irregular seasonality or abrupt regime shifts)?
2. Have you tested simple augmentations (phase shifts, randomized boundaries) to mitigate the UVH bias? You have shown evidence that UVH introduces a copy-period shortcut during forecasting, leading to biased predictions. Did you experiment with simple augmentations to disrupt the implicit periodic alignment? I am wondering whether the bias is structural to the imaging method or sensitive to data alignment choices.
3. The paper concludes that forecasting success in self-supervised LVMs (MAE, SimMIM) mainly stems from pre-trained decoders rather than encoders. Could decoder-swapping experiments (e.g., MAE encoder + SimMIM decoder and vice versa) validate this more directly? Did you observe consistent trends when fine-tuning only the decoder versus only the encoder?
4. The authors report that forecasting performance declines when the input length exceeds roughly 1k steps, hypothesizing this stems from fixed image resolution constraints. Have you tried multi-scale or variable-resolution inputs (e.g., tiled or pyramid representations) to test whether LVMs can better exploit extended contexts when provided with hierarchical image patches?

---

> ### Author Response · Authors · 2025-11-21
> **Response to the comments from reviewer 1Fj9 (Part 1)**
>
> Dear Reviewer 1Fj9,
>
> Thank you so much for the constructive feedback. We sincerely appreciate your valuable suggestions and questions. The following are our responses.
>
> **W1. Most of the forecasting results rely on UVH/MVH imaging applied to relatively periodic datasets. It’s not clear how well these conclusions hold for more irregular, noisy, or non-periodic signals.**
>
> Thanks for this comment, which helps us strength our experiments. Please allow us to start with clarifying our experimental design. A description about the experimental structure is in lines 207-215. Our benchmark study was structured by firstly comparing different LVMs and imaging methods for both TSC (time series classification) and TSF (time series forecasting) tasks in RQ1 and RQ2. Based on the results, we delved into the best LVM forecaster (i.e., MAE+UVH) to analyze its strengths and weakness (as stated in lines 305-307). We focus on the best LVM forecaster because it is more likely to be adopted or to inspire future research, and may help us observe the current cap of LVMs' potential in TSF. Analyzing its limitations (including biases) may be more practically useful than the alternatives. However, this paper still maintains the comparison of all imaging methods in Table 12 as a reference for TSF in Appendix B.2.
>
> Among the 8 TSF benchmark datasets, Weather and Illness don't show strong periodicity (as stated in lines 466-467). To be more comprehensive, we'd like to add two **non-periodic datasets**, i.e., Exchange and Solar, for comparing the best LVM forecaster (i.e., MAE) and the best non-LVM baseline from Table 2 (i.e., PatchTST). The following table summarize the results. In the revised draft, we've included the full results by different forecasting horizons and our analysis in Appendix B.11 (highlighted in blue), along with visualizations of sample time series from these four datasets (Weather, Illness, Exchange, Solar) in Fig. 16, which demonstrates the non-periodic patterns in these datasets. From the results, we can observe that MAE does not show advantage over the non-LVM baseline on non-periodic datasets. This confirms our analysis of LVMs' bias toward periodicity in RQ9 (Section 4.3). We've also referred to the added results in lines 466-468. Furthermore, we call for diversifying benchmark datasets based on our findings in lines 484-485.
>
> | Model       | MAE        |            | PatchTST   |            |
> |-------------|------------|------------|------------|------------|
> | **Dataset** | MSE        | MAE        | MSE        | MAE        |
> | Weather     | **0.225**  | **0.258**  | 0.226      | 0.264      |
> | Illness     | 1.837      | 0.883      | **1.443**  | **0.798**  |
> | Exchange    | 0.405      | 0.432      | **0.383**  | **0.398**  |
> | Solar       | 0.211      | 0.267      | **0.201**  | **0.266**  |
> | # Wins      | 2          |            | **6**      |            |

---

> ### Author Response · Authors · 2025-11-21
> **Response to the comments from reviewer 1Fj9 (Part 2)**
>
> **W2. The claim that decoders matter more than encoders in forecasting is interesting, but the current evidence is mostly correlational. A simple decoder-swapping or partial-finetuning experiment could make this stronger.**
>
> We appreciate the reviewer's recognition of our findings. Following the suggestion, we've investigated the possibility of swapping decoders for MAE and SimMIM, i.e., evaluating MAE encoder + SimMIM decoder and SimMIM encoder + MAE decoder. However, we found it is infeasible for the following reasons.
>
> The output of MAE's encoder is a 3D tensor of size (Batch size, Number of unmasked tokens, Token dimension). The output of SimMIM's encoder is a 4D tensor of size (Batch size, Number of channels, Number of rows in an array of tokens, Number of columns in an array of tokens), where **Number of channels** can be understood as token dimension. Comparing the two outputs, only the **Batch size** can be matched. However, in order to use their pre-trained model parameters, other sizes are pre-defined by the pre-trained models, which are inconsistent across the two LVMs. Specifically, MAE's **Number of unmasked tokens=57** (56 patch tokens + 1 CLS token), **Token dimension=768**, leading to a 3D tensor of size (448, 57, 768) when using 448 as batch size. On the other hand, SimMIM's **Number of channels=1024**, **Number of rows in an array of tokens=6**, **Number of columns in an array of tokens=6**, leading to a 4D tensor of size (448, 1024, 6, 6) when using 448 as batch size. It is noteworthy that, except for batch size, other sizes are pre-defined by the pre-trained LVMs.
>
> The inputs to MAE's decoder (a Transformer decoder) and SimMIM's decoder (a Linear decoder) should have sizes that match the sizes their respective encoders' outputs. Therefore, because of the inconsistency in their encoders' outputs, it is infeasible to swap their decoders. Detailed information about the structures of these two LVMs can be found in [1] and [2].
>
> Actually, our current experiments for RQ8 in Section 4.3 have performed partial fine-tuning: Enc w/o Dec fine-tunes the pre-trained encoder but re-trains the decoder; Dec w/o Enc fine-tunes the pre-trained decoder but re-trains the encoder. We've updated the wordings in lines 424-425 to make our partial fine-tuning efforts more clear (highlighted in blue).
>
> [1] K. He, et al. Masked autoencoders are scalable vision learners. In CVPR, 2022.
>
> [2] Z. Xie, et al. Simmim: A simple framework for masked image modeling. In CVPR, 2022.

---

> ### Author Response · Authors · 2025-11-21
> **Response to the comments from reviewer 1Fj9 (Part 3)**
>
> **W3. The authors note that performance drops for longer look-back windows, potentially due to image resolution limits, but this isn’t experimentally verified. It’s unclear whether the weaknesses observed (e.g., poor long-context forecasting) are intrinsic to LVMs or simply due to architectural mismatch, because the approach reuses vision models mostly as-is, with only changes to heads or decoders.**
>
> Thanks for this comment. The reason for keeping the main architecture of LVMs intact but making a few necessary tweaks for cross-modal adaptation is for faithfully assessing LVMs' innate ability in time series analysis (as stated in lines 131-132 in the paper). This design of research materials is to align with the goal of the benchmark study -- objectively researching LVMs' potential in time series analysis, identifying their strengths and limitations, and providing insights, caveats, and guide for using LVMs in time series tasks.
>
> We think introducing further developments of LVM architectures for making better use of long look-back windows is a non-trivial research problem, and may be out of the scope of this work. It may distract the focus of the paper, especially considering the many possible ways of developments. This can be categorized as further improvements inspired by our findings about the limitations of the basic LVM forecasters. We hope to keep this work focused on analyzing the basic LVMs so that it can inspire future research on various kinds of improvements.
>
> Furthermore, we'd like to extend our analysis (in RQ10) on how LVM forecasters' use of pixels can limit look-back windows: the pixels in the fixed-size image are not fully utilized by the SOTA LVM forecasters. In their imaging setup, each column of pixels (i.e., 256×1) represents an interpolated period with P timesteps (e.g., P=24 for ETTh1), less than 256 timesteps. Also, following the masking strategy in VisionTS [1], an alignment constant c=0.4 is applied, which means that over 60% of the pixels are masked. As a result, only about P×0.4×256 time steps can be effectively used in one image. For example, when P=24, only around 2,400 timesteps can be encoded, which is significantly fewer than the full 256x256 pixels. Thus the amount of effective pixels is influenced by P and c. Other factors such noises and repeated patterns (i.e., redundancy) may further reduce the amount of effective pixels that can inform forecasting. This raises a future direction on how to better utilize image pixels for time series forecasting.
>
> We've added these supplementary understandings to "Further Analysis for RQ10" in Appendix E (highlighted in blue color), and referenced them in the main paper (line 473).
>
> [1] M. Chen, et al. VisionTS: Visual masked autoencoders are free-lunch zero-shot time series forecasters. In ICML, 2025.

---

> ### Author Response · Authors · 2025-11-21
> **Response to the comments from reviewer 1Fj9 (Part 4)**
>
> **W4. The central assumption that converting time series into 2D image structures enables vision transformers to generalize to sequential modeling is interesting but not fully justified. Time and space are fundamentally different dimensions; the study doesn’t discuss what inductive biases are gained or lost by this transformation. For example, local temporal continuity is often disrupted in GAF/UVH encodings, and the benefit of spatial locality learned by ViTs is not clearly transferable to time-dependent modeling.**
>
> Thanks for suggesting the need of discussing the biases (including pros and cons) of using the imaging techniques. Yes, transforming time series to an image may introduce some biases. We summarize the key positive bias (pros) and negative bias (cons) of different imaging methods as following. Since weakness W5 below also asks about comparing different imaging methods, we included a summary of them in our response to W5.
>
> * **Line Plot** (e.g., Fig. 22(a)): (1) Pros: compact representation of long time series; (2) Cons: pixels are not fully used (large white background pixels).
> * **MVH** (e.g., Fig. 22(b)): (1) Pros: direct representation of all variates in a multivariate time series; (2) Cons: random ordering of variates may confuse correlation learning.
> * **UVH** (e.g., Fig. 22(c)): (1) Pros: encode periodic patterns of time series; (2) Cons: break continuity across different periods.
> * **STFT** (e.g., Fig. 22(e)): (1) Pros: encode time-frequency space; (2) Cons: lose raw time series values (which is important in numerical-level tasks).
> * **Wavelet** (e.g., Fig. 22(f)): (1) Pros: encode time-frequency space while fitting varying frequencies; (2) Cons: lose raw time series values.
> * **Filterbank** (e.g., Fig. 22(g)): (1) Pros: encode time-frequency space while fitting high-frequency signals: (2) Cons: lose raw time series values.
> * **GAF** (e.g., Fig. 22(d)): (1) Pros: encode temporal correlations of pairwise time steps; (2) Cons: lose raw time series values, use more pixels for longer time series (O(T^2) complexity)).
> * **RP** (e.g., Fig. 22(h)): (1) Pros: encode cyclic patterns of time series; (2) Cons: lose raw time series values, back-white image loses fine-grained information.
>
> The methods that do not preserve raw time series values (i.e., LinePlot, STFT, Wavelet, Filterbank, GAF, RP) do not fit reconstruction-based LVMs for time series forecasting, leading to their inferior performance in Table 12. This is also discussed in our response to W5.
>
> From the perspective of LVMs, the key advantage of using imaged time series is that it enables transferring the encoded visual knowledge in LVMs obtained by large-scale pre-training to time series tasks. The key limitations, according to our analysis in RQ1, RQ8 and RQ10, includes (1) ineffective use of window-based local attention; (2) less effective use of encoders; and (3) fixed number of usable pixels (which caps the length of look-back windows).
>
> We've added this discussion to Appendix F.2 (highlighted in blue).

---

> ### Author Response · Authors · 2025-11-21
> **Response to the comments from reviewer 1Fj9 (Part 5)**
>
> **W5. The paper tests many imaging techniques (GAF, RP, UVH, etc.), but it remains unclear why certain encodings align better with LVM features. There’s little analysis of the representational geometry, e.g., whether these encodings preserve correlations, periodicities, or value scales in a meaningful way. The success of UVH/MVH feels more empirical.**
>
> Thanks a lot for asking for clarification. We'd like to add the suggested analysis. If our understanding is correct, this question asks why different imaging techniques lead to different successes (or failures) of LVMs in time series analysis. This relates to (1) the property of each imaging technique; and (2) the effectiveness of the property when using LVMs.
>
> First, the key property of each imaging technique can be deterministically identified from its definition as specified in [1]. In Fig. 22 (and Fig. 1(a)), we've empirically visualized these methods. In the following, we summarize the key property of each imaging technique according to the formal (mathematical) definitions in [1]. We'd like to refer interested readers to [1] for the detailed definitions. We've also added the following analysis and the visualizations in Appendix F.1 (highlighted in blue).
>
> * **Line Plot** (e.g., Fig. 22(a)) uses a line to represent a time series in a 2D image. Its use of pixels is ineffective compared to other imaging techniques because most of the pixels are used for representing the white background.
> * **MVH** (e.g., Fig. 22(b)) is the only method that directly visualize all variates in an MTS in a single heatmap image. However, there is no principled way to determine the order of variates on the y-axis. Different orders may lead to different cross-variate patterns learned by an LVM.
> * **UVH** (e.g., Fig. 22(c)) visualizes stacked segments of a time series as a heatmap image. As such, it may lead to a bias toward periodicity when the segments are periods.
> * **STFT** (e.g., Fig. 22(e)) visualizes the time-frequency space of a time series. However, it uses a fixed-size sliding window which cannot fit varying frequencies in a time series.
> * **Wavelet** (e.g., Fig. 22(f)) visualizes the time-frequency space of a time series using wavelets. It addresses STFT's limitation but needs a proper choice of wavelet function.
> * **Filterbank** (e.g., Fig. 22(g)) resembles STFT but is associated with several processing filters on a Mel-scale that fit audio signals thus is more often used for imaging audio signals.
> * **GAF** (e.g., Fig. 22(d)) is a square T-by-T image (T is the length of look-back window). It is good at capturing the temporal correlations and cyclical patterns in a time series.
> * **RP** (e.g., Fig. 22(h)) is also a square image but its size is tunable by its hyperparameters. It is good at capturing periodic patterns but is a black-white image thus may lose some fine-grained information.
>
> Among these methods, only UVH and MVH images encode the raw (or normalized) time series values, while other methods visualize different transformations of time series values. This will lead to their different performance in a forecasting (numerical-level) task. Additionally, STFT, Wavelet, and Filterbank produce spectrograms. Their difference lies in their different ways of encoding frequencies in a time series.
>
> Second, our experiments for comparing the 8 imaging methods provide insights on the effectiveness of these methods when using LVMs. For classification, the averaged results in Table 11 (the bottom row) suggest that Line Plot and Filterbank are less effective than other methods. This is because of Line Plot's ineffective use of pixels and Filterbank's filter processing that is aligned with high-frequency audio signals. For the other 6 methods, despite their different levels of effectiveness, they all achieve reasonable averaged accuracy (~70%), suggesting their effectiveness in capturing distinguishable patterns (e.g, GAF, RP, UVH capture periodic patterns; STFT, Wavelet capture time-frequency patterns; MVH captures overall trend patterns).
>
> For forecasting, the results in Table 12 suggest that UVH and MVH are better than other methods. This is because forecasting task requires predicting future time series values. Among the 8 methods, only UVH and MVH preserve the original time series values in the look-back window, thus they are more effective than other transformation methods in a numerical-level task, i.e., forecasting. However, MVH underperforms UVH. The limitation of MVH is its mixture of variates in a single image without a principled ordering of the variates, which may confuse cross-variate dependency learning.
>
> [1] J. Ni, et al. Harnessing vision models for time series analysis: A survey. In IJCAI, 2025.

---

> ### Author Response · Authors · 2025-11-21
> **Response to the comments from reviewer 1Fj9 (Part 6)**
>
> **Q1. How robust are the UVH/MVH findings to non-periodic datasets or those with exogenous inputs (e.g., with irregular seasonality or abrupt regime shifts)?**
>
> This question is related to weakness W1 above. The response is summarized in the response to W1. For exogenous variables, we think it is necessary to further develop LVMs for jointly modeling both endogenous and exogenous variables, which is an open-question and may need non-trivial developments. Like other possible developments upon LVMs such as modeling longer look-back windows, modeling spatial dependencies, and mitigating bias toward periodicity, this is out of the scope of this benchmark study. By provide an in-depth understanding of LVMs' innate ability in time series analysis, we hope this work can inspire future research on various kinds of developments.
>
>
> **Q2. Have you tested simple augmentations (phase shifts, randomized boundaries) to mitigate the UVH bias? You have shown evidence that UVH introduces a copy-period shortcut during forecasting, leading to biased predictions. Did you experiment with simple augmentations to disrupt the implicit periodic alignment? I am wondering whether the bias is structural to the imaging method or sensitive to data alignment choices.**
>
> Thank you for this suggestion. We think augmentations such as phase shifts may not only disrupt the periodic alignment but also introduce noises such as disrupted temporal patterns. A better method to break the periodic alignment while maintaining the temporal structure of the input time series is to use a non-periodic length for segmenting the input time series when constructing the UVH image.
>
> In RQ9, we did this experiment. Fig. 8 shows the forecasting performance when changing the segment length in various ratios of a period L. As can be seen, when segments don't align with periods, i.e., not L nor 2L, the forecasting performance is worse. This means LVM forecasters is mostly adept at utilizing past periods for forecasting future periods, while cannot optimally forecast when periodic alignment is broken. Therefore, we think this bias is sensitive to the choice of the segment length.
>
>
> **Q3. The paper concludes that forecasting success in self-supervised LVMs (MAE, SimMIM) mainly stems from pre-trained decoders rather than encoders. Could decoder-swapping experiments (e.g., MAE encoder + SimMIM decoder and vice versa) validate this more directly? Did you observe consistent trends when fine-tuning only the decoder versus only the encoder?**
>
> This question is related to weakness W2 above. The response is summarized in the response to W2. For the second sub-question, yes, our experiments for RQ8 have evaluated these two cases. In Fig. 7, Dec w/o Enc is the model that fine-tunes only the pre-trained encoder but re-trains a randomly initialized decoder; Enc w/o Dec is the model that fine-tunes only the encoder but re-trains a randomly initialized decoder. It is noteworthy that random initialization is necessary because we aim to evaluate the impact of removing the pre-trained knowledge in encoder or decoder. Therefore, the results in Fig. 7 are consistent with the case in the second sub-question.
>
> We think our description about these two models in lines 424-425 is not clear enough thus causes this confusion. We've changed the wordings in the revised draft (highlighted in blue).
>
>
> **Q4. The authors report that forecasting performance declines when the input length exceeds roughly 1k steps, hypothesizing this stems from fixed image resolution constraints. Have you tried multi-scale or variable-resolution inputs (e.g., tiled or pyramid representations) to test whether LVMs can better exploit extended contexts when provided with hierarchical image patches?**
>
> Because pre-trained LVMs requires an input image to have a fixed size (256x256x3), which is defined by their pre-training datasets (e.g., Image-Net) and pre-training setup, the resolution cannot be arbitrarily changed in order to re-use their pre-trained knowledge (lines 148-151). This is a general limitation of using pre-trained LVMs, in both computer vision domain and time series domain. However, contemporary LVMs can handle moderately long windows well (1000 may be considered sufficient in many cases). Also, we believe as newer LVMs will be pre-trained and released in the future, longer look-back windows will be effectively modeled. Meanwhile, we hope this work can inspire future research on developing contemporary LVMs for effectively modeling longer look-back windows. This question is also related to weakness W3. Additional analysis is summarized in the response to W3.

---

### Official Review · Reviewer_fb57 · 2025-10-28

**Soundness:** 3
**Presentation:** 3
**Contribution:** 3
**Rating:** 6
**Confidence:** 3

**Summary:**

This paper conducts the comprehensive study on the utility of Large Vision Models for time series analysis, covering classification and forecasting tasks. It evaluates 4 LVMs, 8 imaging methods, 18 datasets, and 26 baselines. Key findings show LVMs excel at TSC by leveraging pre-trained semantic recognition capabilities, with GAF as the optimal imaging method. However, TSF poses greater challenges, self-supervised LVMs such as MAE paired with UVH imaging perform best but suffer from period bias and limited long look-back window utilization. Ablation analyses confirm LVMs transfer pre-trained knowledge, capture temporal order, and rely more on decoders for forecasting. While LVMs trade higher computational cost for superior TSC performance, their TSF effectiveness is constrained by task-specific limitations, providing foundational insights for multimodal time series research.

**Strengths:**

1. It systematic explore large vision models for time series classification and forecasting, covering diverse models, imaging methods, datasets, and baselines to fill existing research gaps.
2. In-depth mechanism analysis reveals key insights and quantifies temporal pattern capture, providing essential support for future optimizations.
3. It identifies optimal task-specific configurations and targeted fine-tuning strategies, enabling efficient real-world implementation.
4. Rigorous benchmark comparisons  and analysis experiments ensure reliable conclusions, establishing a credible reference for subsequent research.

**Weaknesses:**

1. The study mentions that Large Vision Models have numerous limitations in time series forecasting tasks. However, leveraging the feature extraction capability of LVMs for TSF and integrating them with time series through cross-modal fusion may help avoid visual limitations and improve overall performance. The current work lacks an investigation into cross-modal fusion involving visual modalities.
2. The study evaluates 8 imaging methods, and from Table 16, UVH achieves the best performance while MVH ranks second. Other imaging methods perform significantly worse—frequency-domain methods such as Wave and STFT do not yield better results through imaging. Does this indicate that frequency-domain graphs are unsuitable as inputs for LVMs? For MVH and UVH, why does MVH underperform UVH, and would the results differ for datasets with variable dependencies? Additionally, could simultaneous input of MVH and UVH bring performance gains?
3. The study lacks comparisons with state-of-the-art large time series models (e.g., Chronos-2, Moirai2), which may undermine the comprehensiveness of LVMs’ performance evaluation.

**Questions:**

See Weakness.

---

> ### Author Response · Authors · 2025-11-21
> **Response to the comments from reviewer fb57 (Part 1)**
>
> Dear Reviewer fb57,
>
> Thank you so much for the constructive feedback. We sincerely appreciate your valuable suggestions and questions. The following are our responses.
>
> **W1. The study mentions that Large Vision Models have numerous limitations in time series forecasting tasks. However, leveraging the feature extraction capability of LVMs for TSF and integrating them with time series through cross-modal fusion may help avoid visual limitations and improve overall performance. The current work lacks an investigation into cross-modal fusion involving visual modalities.**
>
> Thanks for this insightful forward-looking comment. However, we think there may be a confusion about the position of our work. Please allow us to clarify it.
>
> The key contribution of this work is a benchmark study (as stated in Section 1, lines 74-75), with the goal of objectively researching LVMs' potential in time series analysis, identifying their strengths and limitations, providing insights, caveats, and **guide for using and developing LVMs in time series tasks**. We aim to not only identify whether LVMs can succeed, but also why they succeed or fall short. This work shares similar merits as the benchmark studies in [1] (Transformers for time series forecasting), [2] (LLMs for time series forecasting), and [3] (LLMs for time series anomaly detection). However, to the best of our knowledge, none of the existing benchmark studies explores LVM's potential in time series analysis. Given the growing research attention in this area (i.e., vision for time series), as identified by [4] and its associated GitHub repository (e.g., more and more emerging papers), we think a timely benchmark study can serve as a foundation for future research that may utilize and develop LVMs for proposing novel solutions.
>
> Therefore, the scope of this work lies in the **discovery of LVM's potentials and limitations** in time series analysis. There could be multiple possible ways to develop LVMs -- including the suggested cross-modal fusion -- inspired by the findings of our work, which may be considered as future research directions that benefit from this benchmark study. Actually, we hope our work can draw the community's attention and inspire future solutions for improving LVMs for time series analysis.
>
> [1] A. Zeng, et al. Are transformers effective for time series forecasting? In AAAI, 2023.
>
> [2] M. Tan, et al. Are language models actually useful for time series forecasting? In NeurIPS, 2024.
>
> [3] Z. Zhou, et al. Can LLMs understand time series anomalies? In ICLR, 2025.
>
> [4] J. Ni, et al. Harnessing vision models for time series analysis: A survey. In IJCAI, 2025.

---

> ### Author Response · Authors · 2025-11-21
> **Response to the comments from reviewer fb57 (Part 2)**
>
> **W2. The study evaluates 8 imaging methods, and from Table 16, UVH achieves the best performance while MVH ranks second. Other imaging methods perform significantly worse -- frequency-domain methods such as Wave and STFT do not yield better results through imaging. Does this indicate that frequency-domain graphs are unsuitable as inputs for LVMs? For MVH and UVH, why does MVH underperform UVH, and would the results differ for datasets with variable dependencies? Additionally, could simultaneous input of MVH and UVH bring performance gains?**
>
> From the described question, we think the mentioned results are from Table 12 since Table 16 does not include image comparison. From Table 12, UVH and MVH are better than other imaging methods in time series forecasting (TSF). This is because TSF is a numerical-level task, i.e., it requires the input image to provide concrete time-step-wise values in a look-back window of the history so that the model can accurately forecast the numerical values in future time steps. Among the 8 imaging methods, only UVH and MVH preserves the input time series values in their pixels (after normalization, which is revertible), while other imaging methods transform input time series to pixel values bearing other meanings, which lose the original time series values. For example, in STFT image and Wavelet image, a pixel value represents the strength of a frequency (y-axis) at a time step (x-axis), which loses the historical time series values, making LVMs less effective in inferring future time series values.
>
> Thanks for this comment, we are now aware that, by referring readers to the reference in lines 146-147, our description of the 8 imaging methods may be inadequate and may lead to a lack of understanding. As such, in the revised draft, we've added descriptions of the key properties of each imaging method and their pros and cons in Appendix F.1 and F.2, and referred to them in lines 146-147 (highlighted in blue).
>
> The reason for UVH outperforms MVH is two-fold: (1) MVH mixes all variates in a single image, and there is no principled way to order variates on the y-axis, which may confuse LVMs when learning cross-variate dependencies. This may draw the community's attention for a future research on how to best order variates in MVH; (2) UVH's bias by stacking periods of time series when constructing the image may favor datasets with periodicity (some benchmark datasets have periodicity). This bias resembles the inductive (periodic) bias that was harnessed by some early works [1][2][3] for TSF. As such, we call for diversifying benchmark datasets for TSF as a future direction in the community in line 484 (this has been included in our original submission, here we highlighted it in blue). Moreover, based on reason (1), we think the results may differ with variable dependencies.
>
> Finally, we think simultaneous input of MVH and UVH, and possibly other imaging methods, may provide multiple views that have complementary information, which has a potential to improve TSF performance. We think this may be another insight inspired by our work. By discovering the limitations of LVMs' performance, we hope our benchmark study may inspire many interesting future directions. This is exactly a goal of this work, as stated in lines 21-22 in the abstract.
>
> [1] H. Wu, et al. Autoformer: Decomposition transformers with auto-correlation for long-term series forecasting. In NeurIPS, 2021.
>
> [2] H. Wu, et al. Timesnet: Temporal 2d-variation modeling for general time series analysis. In ICLR, 2023.
>
> [3] S. Lin, et al. Cyclenet: Enhancing time series forecasting through modeling periodic patterns. In NeurIPS, 2024.

---

> ### Author Response · Authors · 2025-11-21
> **Response to the comments from reviewer fb57 (Part 3)**
>
> **W3. The study lacks comparisons with state-of-the-art large time series models (e.g., Chronos-2, Moirai2), which may undermine the comprehensiveness of LVMs’ performance evaluation.**
>
> Thanks for this comment. We were hesitant to include times series foundation models (TSFMs) such as Chronos-2 and Moirai2 because the purpose of this work is to assess whether the pre-trained knowledge from image data in LVMs -- i.e., **cross-modal knowledge** -- can be effectively transferred to time series tasks. As such, we focus on comparing LVMs with the state-of-the-art time series models without pre-trained knowledge, and models with other cross-modal knowledge -- such as language knowledge in Time-LLM and GPT4TS -- for fair comparison of cross-modal knowledge.
>
> TSFMs like Chronos and Moirai mostly encode **intra-modal knowledge**, i.e., pre-trained knowledge from large-scale time series data, which is not orthogonal to cross-modal knowledge. For example, one future research could be fine-tuning image-pretrained LVMs on large-scale time series data so that the resulted LVMs can integrate both cross-modal knowledge and intra-modal knowledge for enhancing performance.
>
> We hope this could clarify our evaluation of baselines. In the revised draft, we've added this clarification in lines 202-204 in Section 4 and lines 796-800 in Appendix A.2.
>
> Moreover, following the suggestion, in the limited time for rebuttal, we implemented an efficient TSFM -- LightGTS [1], which was pre-trained on large-scale time series datasets while is more parameter-efficient than Chronos and Moirai. Meanwhile, LightGTS was demonstrated to outperform Chronos and Moirai on TSF benchmark datasets in [1]. We compared the best LVM forecaster, i.e., MAE, with LightGTS on the 8 benchmark datasets and summarized the results (averaged over different forecasting horizons) in the following table. We've included the results with full horizons in Table 28 in Appendix G in the revised draft. From the results, we observe LVM's strong potential in transferring cross-modal knowledge, which appears to be more useful than the intra-modal knowledge in LightGTS. We think this advantage attributes to the much larger size of the available pre-training image datasets than that of the pre-training time series datasets collected by existing methods. The results also suggest a potential research direction of exploring the integration of cross-modal knowledge and intra-modal knowledge for enhancing performance. We've added this analysis to Appendix G.
>
> |  Model   | ETTh1 |       | ETTh2 |       | ETTm1 |       | ETTm2 |       | Weather |       | Illness |       | Traffic |       | Electricity |       | # Wins |
> | :------: | :---: | :---: | :---: | :---: | :---: | :---: | :---: | :---: | :-----: | :---: | :-----: | :---: | :-----: | :---: | :---------: | :---: | :--: |
> |          |  MSE  |  MAE  |  MSE  |  MAE  |  MSE  |  MAE  |  MSE  |  MAE  |   MSE   |  MAE  |   MSE   |  MAE  |   MSE   |  MAE  |     MSE     |  MAE  |      |
> |   **MAE**    | 0.409 | **0.419** | 0.357 | **0.390** | **0.345** | **0.374** | **0.268** | 0.327 |  0.225  | 0.258 |  **1.837**  | **0.883** | **0.386**  | **0.256** |    **0.159**    | **0.250** |  **11**  |
> | **LightGTS** | **0.394** | **0.419** | **0.339** | 0.392 | 0.417 | 0.406 | **0.268** | **0.319** |  **0.224**  | **0.238** |  3.024  | 1.255 |  0.439  | 0.329 |    0.187    | 0.279 |  7   |
>
>
> [1] Y. Wang, et al. LightGTS: A Lightweight General Time Series Forecasting Model. In ICML, 2025.

---

### Official Review · Reviewer_EKwA · 2025-10-31

**Soundness:** 3
**Presentation:** 3
**Contribution:** 3
**Rating:** 6
**Confidence:** 3

**Summary:**

This paper presents a comprehensive benchmark study investigating the effectiveness of Large Vision Models (LVMs) for time series analysis. The authors evaluate four LVMs (ViT, Swin, MAE, SimMIM) across two representative tasks: time series classification (TSC) and time series forecasting (TSF). Through extensive experiments with 8 imaging methods and 18+ baselines on 18 datasets, the paper finds that LVMs excel at TSC but face significant challenges in TSF due to limitations in encoder utilization, inductive biases, and long look-back window handling. The work provides detailed ablation studies and practical insights for adapting LVMs to time series tasks.

**Strengths:**

(1) The paper covers a substantial experimental scope (4 LVMs, 8 imaging methods, 18 datasets, 26 baselines) with well-designed ablation studies (RQ1-RQ10). The breadth of analysis is commendable and provides genuine value to the community.
(2) The discovery that pre-trained decoders contribute more than encoders in TSF (RQ8) is genuinely interesting and counterintuitive. The analysis of the period-based imaging bias (RQ9) with formal characterization (Lemma 1) provides actionable insights.
(3) The temporal perturbation experiments (RQ6, Table 4) effectively demonstrate that LVMs do capture temporal information, strengthening claims about their utility beyond pattern matching.
(4) The paper is generally well-organized with clear research questions guiding the narrative. Figures 1-3 effectively summarize the methodology and key findings.

**Weaknesses:**

(1) The paper is purely an empirical benchmark study without methodological contributions. While valuable, such studies typically require either exceptional insights or novel proposed solutions.
(2) The paper identifies what fails (encoders, long windows) but provides limited mechanistic understanding of why.
(3) The claim that forecasting is "low-level" and requires numerical inference needs deeper investigation beyond decoder architecture.
(4) The connection to recent multimodal approaches (mentioned briefly) deserves more engagement.
(5) Using ImageNet-derived normalization for time series images may introduce bias. Ablation on normalization strategy is absent.
(6) Why variate-independence assumption for all tasks? For TSC, joint multivariate modeling might capture interactions.

**Questions:**

(1) Decoder Importance (RQ8): Can you provide more analysis on what the decoders learn that aids TSF? Attention visualizations or learned representations would be illuminating. Is this specific to MAE/SimMIM architectures or general?
(2) Periodic Bias Generalization: Lemma 1 is elegant for UVH, but how do other imaging methods induce biases? Can similar formal analysis be provided for GAF or MVH? Does this explain why they underperform for TSF?
(3) Why limit to 8 TSF datasets? Larger evaluation (similar to TSC's 10 datasets) would strengthen claims. Would results change with higher-dimensional time series (e.g., multivariate > 20 dimensions)?
(4) The recommendation to fine-tune only norm layers for TSF seems counterintuitive. Is this an artifact of small datasets or a fundamental limitation of LVMs for low-level tasks?

---

> ### Author Response · Authors · 2025-11-21
> **Response to the comments from reviewer EKwA (Part 1)**
>
> Dear Reviewer EKwA,
>
> Thank you so much for the constructive feedback. We sincerely appreciate your valuable suggestions and questions. The following are our responses.
>
> **W1. The paper is purely an empirical benchmark study without methodological contributions. While valuable, such studies typically require either exceptional insights or novel proposed solutions.**
>
> We are thankful to the reviewer for recognizing the type of this work. Yes, this work is a benchmark study, with the goal of objectively researching LVMs' potential in time series analysis, identifying their strengths and limitations, providing insights, caveats, and guide for using LVMs in time series tasks. We aim to not only identify whether LVMs can succeed, but also why they succeed or fall short. This work shares similar merits as the benchmark studies in [1] (Transformers for time series forecasting), [2] (LLMs for time series forecasting), and [3] (LLMs for time series anomaly detection). However, to the best of our knowledge, none of the existing benchmark studies explores LVM's potential in time series analysis. Given the growing research attention in this area (i.e., vision for time series), as identified by [4] and its associated GitHub repository (e.g., more and more emerging papers), we think a timely benchmark study can serve as a foundation for future research that may utilize and develop LVMs for proposing novel solutions.
>
> As such, like [1][2][3], the focus of this work is on discovering LVMs' capabilities in time series analysis. We think proposing novel solutions could be a distinct topic beyond the scope of this benchmark work. The novelty of this work lies in the identification of the problem, the research method for conducting a comprehensive, structured, and solid study that touches the most concerned questions, and the in-depth analysis about the findings. The exceptional insights include (1) the SOTA performance of certain LVMs on time series benchmark datasets; (2) the different challenges of time series classification (semantic-level task) and forecasting (numerical-level task) when using LVMs; and (3) the disruptive findings (and our analysis) that the SOTA LVM forecasters are limited in using encoders and long context windows, and their strong biases imposed by their imaging method.
>
> [1] A. Zeng, et al. Are transformers effective for time series forecasting? In AAAI, 2023.
>
> [2] M. Tan, et al. Are language models actually useful for time series forecasting? In NeurIPS, 2024.
>
> [3] Z. Zhou, et al. Can LLMs understand time series anomalies? In ICLR, 2025.
>
> [4] J. Ni, et al. Harnessing vision models for time series analysis: A survey. In IJCAI, 2025.

---

> ### Author Response · Authors · 2025-11-21
> **Response to the comments from reviewer EKwA (Part 2)**
>
> **W2. The paper identifies what fails (encoders, long windows) but provides limited mechanistic understanding of why.**
>
> We agree with the reviewer that explaining the observed results is necessary. Therefore, we tried our best to provide concise and key explanations that fit the page limit to each challenge of the LVM forecasters. Thanks to the reviewer's comment, we are now aware that our analysis may be inadequate. We'd like to extend our analysis to RQ8 (limitation of encoders) and RQ10 (ineffective use of long windows) in the paper.
>
> For RQ8, the ablation in Fig. 7 validates the importance of LVMs' decoders in forecasting. Our understanding is that this is because LVMs’ decoders aim to reconstruct pixel values, while the encoders aim to extract general-purpose features. For a forecasting task, reconstructing pixel values aligns more with forecasting the numerical values in a time series, thus the decoders play a more important role than the encoders.
>
> For RQ10, our understanding is that the ineffective use of look-back windows may result from image transformation: fixed-size image in pre-trained LVMs has a pixel limit and may constrain the information captured from excessively long time series. Additionally, the pixels in the fixed-size image are not fully utilized by the SOTA LVM forecasters. In their imaging setup, each column of pixels (i.e., 256×1) represents an interpolated period with P timesteps (e.g., P=24 for ETTh1), less than 256 timesteps. Also, following the masking strategy in VisionTS [1], an alignment constant c=0.4 is applied, which means that over 60% of the pixels are masked. As a result, only about P×0.4×256 time steps can be effectively used in one image. For example, when P=24, only around 2,400 timesteps can be encoded, which is significantly fewer than the full 256x256 pixels. Thus the amount of effective pixels is influenced by P and c. Other factors such noises and repeated patterns (i.e., redundancy) may further reduce the amount of effective pixels that can inform forecasting. This raises a future direction on how to better utilize image pixels for time series forecasting.
>
> We've added these supplementary understandings to Appendix E (highlighted in blue color), and referenced them in the main paper (line 428 and line 473).
>
> [1] M. Chen, et al. VisionTS: Visual masked autoencoders are free-lunch zero-shot time series forecasters. In ICML, 2025.
>
>
> **W3. The claim that forecasting is "low-level" and requires numerical inference needs deeper investigation beyond decoder architecture.**
>
> Thanks for this comment. Please allow us to clarify the use of this term. The term "low-level" is used mainly to describe the numerical-level task on time series forecasting (TSF) for the purpose of distinguishing it from the semantic-level task on time series classification (TSC). TSC requires an LVM to recognize the distinguishable patterns in an imaged time series (like the semantics it convey) for differentiating it from the imaged time series in other classes. This is in contrast to TSF, which requires an LVM to correctly predict the numbers (i.e., pixel values) in the forecasting horizon based on the numbers in the look-back window.
>
>
> **W4. The connection to recent multimodal approaches (mentioned briefly) deserves more engagement.**
>
> Thank you for pointing this out. We've added more discussions about the related works on multimodal methods for time series analysis in Appendix E (highlighted in blue color), and referenced it in the main paper (line 127).
>
>
> **W5. Using ImageNet-derived normalization for time series images may introduce bias. Ablation on normalization strategy is absent.**
>
> Please let us to clarify the reason for using mean-variance standardization as the normalization method on time series images (lines 151-155). Since we aim to make effective use of pre-trained LVMs, there is a need to align the input images with the pre-training datasets of the LVMs. In other words, like the regular adoption of these LVMs (ViT, Swin, MAE, SimMIM) on image data, after transforming time series to images, the images need to be preprocessed in the same way as the pre-training images for the LVMs to avoid distribution shifts. Therefore, we follow this convention of data preprocessing.

---

> ### Author Response · Authors · 2025-11-21
> **Response to the comments from reviewer EKwA (Part 3)**
>
> **W6. Why variate-independence assumption for all tasks? For TSC, joint multivariate modeling might capture interactions.**
>
> This is an insightful comment. Yes, modeling inter-variate interactions is promising for multivariate time series. The main reason for adopting variate-independence assumption in this work is because we aim to assess LVMs' innate ability in time series analysis by keeping the main architecture intact but making a few necessary tweaks for cross-modality adaptation (lines 131-132). We've thought about modeling inter-variate interactions -- for example, by stacking images of multiple variates as multiple channels and using CNNs to aggregate these channels before feeding them to LVMs. However, this will introduce non-trivial developments of the LVM forecasters, which may distract the focus of the paper, especially considering the many possible ways of developments. We think developing LVM-based forecasters for capturing inter-variate interactions can be categorized as a further improvement upon the basic LVM forecasters. Other improvements may include modeling exogenous inputs, integrating multiple imaging methods, mitigating forecasting biases, and so on. We hope to keep this work focused on analyzing the basic LVMs so that it can inspire future research on various kinds of improvements.
>
>
> **Q1. Decoder Importance (RQ8): Can you provide more analysis on what the decoders learn that aids TSF? Attention visualizations or learned representations would be illuminating. Is this specific to MAE/SimMIM architectures or general?**
>
> Thank you for the suggestion. The two encoder-decoder LVMs have different decoders -- MAE has a 8-layer Transformer decoder and SimMIM has a single-layer linear decoder. Therefore, the underlying learning mechanisms for their effectiveness in TSF may be different. Following the suggestion, we've tried to visualize the learned representations of patch tokens before the last reconstruction layers in their decoders. However, the representations have two clusters: (1) unmasked patch embeddings and (2) masked patch embeddings, which cannot effectively illustrate the reconstruction process. Additionally, since SimMIM's decoder has no attention, we've tried to visualize MAE's attention scores in its 8-layer Transformer decoder. However, because of the complex structure of its decoder, the attentions don't show consistent and explainable patterns across the layers.
>
> Therefore, like [1] and [2], we visualized the reconstructed images of both MAE's decoder and SimMIM's decoder, and compared the reconstructions with the ground truths. In the revised draft, we've included such visualizations in Fig. 21 in Appendix D.3 for random samples from 5 datasets -- ETTh1, ETTm1, Weather, Traffic, and Electricity. In Fig. 21, the red dashed lines separate the look-back windows and the forecasts. In addition to the UVH images, we visualized the recovered time series from the images. From Fig. 21, we can obverse the reconstruction ability of LVMs' decoders on the masked areas (which correspond to the forecasting horizons), which tend to be smooth across columns. This confirms our analysis of LVMs' bias toward periodicity in RQ9 (Section 4.3). It also illustrates the working mechanism of LVMs' decoders in TSF task.
>
> Finally, we think this mechanism applies to general LVMs with pre-trained decoders for masked image reconstruction. Both MAE and SimMIM have such pre-trained decoders, but their decoders are different in architectures. Thus they can be considered as representative instances with different realizations for general LVMs with pre-trained decoders.
>
> We've added this analysis to Appendix D.3 in the revised draft.
>
> [1] K. He, et al. Masked autoencoders are scalable vision learners. In CVPR, 2022.
>
> [2] Z. Xie, et al. Simmim: A simple framework for masked image modeling. In CVPR, 2022.

---

> ### Author Response · Authors · 2025-11-21
> **Response to the comments from reviewer EKwA (Part 4)**
>
> **Q2. Periodic Bias Generalization: Lemma 1 is elegant for UVH, but how do other imaging methods induce biases? Can similar formal analysis be provided for GAF or MVH? Does this explain why they underperform for TSF?**
>
> We appreciate the reviewer's recognition of the merit of Lemma 1. The reason for our more in-depth analyses on UVH than other imaging methods lies in the focus of analyzing the best LVM forecaster (i.e., MAE) (as stated in lines 306-307), which uses UVH. As Table 12 suggests, UVH is much better than other imaging methods, which may establish an upper bound of the SOTA LVMs' performance in TSF. Considering the best LVM forecaster is more likely to be adopted or to inspire future research, we think analyzing its limitations (including biases) is more practically useful than the alternatives. Therefore, we concentrate our analysis on UVH.
>
> We think the key limitation of GAF (and RP, STFT, Wavelet, Filterbank, Lineplot) in TSF is because they don't preserve the original time series values in their images. Without knowing historical time series values, LVMs cannot effectively forecasting future values. MVH preserves time series values, thus is better than the above methods. However, it underperforms UVH. The limitation of MVH is its mixture of variates in a single image without a principled ordering of the variates.
>
> We've included this analysis in Appendix B.2 (highlighted in blue color), and referenced it in the main paper (line 263).
>
>
> **Q3. Why limit to 8 TSF datasets? Larger evaluation (similar to TSC's 10 datasets) would strengthen claims. Would results change with higher-dimensional time series (e.g., multivariate > 20 dimensions)?**
>
> We agree with the reviewer that more datasets can provide more reliable results. We used the 8 TSF datasets because they are widely used benchmark datasets. For example, they have been used by the following works on TSF [1][2][3], to name a few. Also, 8 datasets could be considered as sufficiently large. From Table 7, these 8 datasets have a wide coverage of different types of time series (e.g., number of variates ranges from 7 to 862) from different domains (e.g., temperature, electricity, health, weather, traffic), thus can be considered as representative. We think using these datasets aligns well with existing works in this area and can fairly reflect LVMs' potential in TSF when comparing with SOTA methods.
>
> From Table 7, among the datasets, Electricity dataset has 21 variates, Weather dataset has 321 variates, Traffic dataset has 862 variates. As such, we think our analysis has covered high-dimensional time series.
>
> [1] M. Jin, et al. Time-llm: Time series forecasting by reprogramming large language models. In ICLR, 2024
>
> [2] M. Chen, et al. VisionTS: Visual masked autoencoders are free-lunch zero-shot time series forecasters. In ICML, 2025.
>
> [3] S. Zhong, et al. Time-vlm: Exploring multimodal vision-language models for augmented time series forecasting. In ICML, 2025.
>
>
> **Q4. The recommendation to fine-tune only norm layers for TSF seems counterintuitive. Is this an artifact of small datasets or a fundamental limitation of LVMs for low-level tasks?**
>
> We think this observation is caused by the low-level nature of TSF task. However, we'd like to categorize it as an insight rather than a limitation of LVMs. The low-level nature of TSF task can cause the observation of fine-tuning only norm layers in Table 3 because the model needs to predict numerical values, which is largely influenced by normalization layers. The size of the fine-tuning datasets may also be a factor because fine-tuning more parameters than necessary may lead to overfitting. We hope the findings in Table 3 can guide future research that requires fine-tuning LVMs for TSF. In the revised draft, we've also highlighted this explanation in lines 322-355.

---

### Author Response · Authors · 2025-11-21
**Appreciate your attention and time**

Dear Reviewers,

We sincerely appreciate your valuable comments that help us refine our work. If you have more questions or concerns about our response or the current draft, please let us know. We are happy to discuss with you.

---

### Author Response · Authors · 2025-11-27
**Looking forward to hearing from you**

Dear reviewers,

Thank you for reviewing our paper and providing the constructive comments. As the discussion period is approaching to its end, we would appreciate if the reviewers could let us know if our response could address the questions and sufficiently justify our work. We would also appreciate if the reviewers could let us know where we can refine or what concerns you still have, so that we can take the opportunity to clarify them and improve our work. Thank you!

---

### Author Response · Authors · 2025-12-02
**A summary of the author response and paper revisions**

Dear Reviewers and Area Chairs,

We are thankful to your insightful feedback and suggestions for improving the paper. We are glad that the reviewers find our benchmark study (on LVMs for time series) comprehensive, and valuable to subsequent research in the community. We also appreciate the reviewers in recognizing our findings and analysis regarding the impacts of imaging, pre-training, fune-tuning, architecture (encoding/decoding), and biases in this research, which may provide actionable insights for future directions. The questions from the reviewers inspire us to further strengthen our analysis. We tried to address the questions and take the suggestions by **adding appropriate discussions**, **performing additional experiments**, and **revising the paper**. **The revisions are highlighted in blue in the draft**.

**Here please allow us to clarify a potential misunderstanding about the position of this work**. The goal of this work is to objectively research LVMs' potential in time series analysis, identifying their strengths and limitations, providing insights, caveats, and guide for using LVMs in time series tasks. Some reviews present concerns about **why not propose (novel) solutions** to the limitations and biases that were discovered by our benchmark study, covering different aspects of LVMs such as (1) joint modeling of multiple variates, (2) joint modeling of both exogenous and endogenous inputs, (3) integrating multiple imaging methods as the input to LVMs, (4) mitigating the forecasting biases of LVMs, and (5) better modeling of long look-back windows. We really appreciate these forward-looking suggestions, but we humbly think they are **various kinds of future directions inspired by the findings of this work**. Solving these open problems may involve non-trivial developments on LVMs for time series, and may **distract the focus of the paper**, especially considering the many possible ways of developments. As such, they could be considered as distinct research problems that are out of the scope of this work -- which lies in the discovery of LVM's potentials and limitations in time series analysis. Actually, **inspiring further developments of LVMs for better solving time series problems is a goal of this benchmark research**. By providing an in-depth understanding of LVMs' innate ability in time series analysis, we hope this work can serve as a foundation and guide for future research that may utilize and develop LVMs for proposing novel solutions (lines 21-22 in the abstract, lines 50-51 in Sec. 1).

Meanwhile, this benchmark research is not a trivial reflection of existing techniques. **The novelty of this work** lies in the identification of the problem, the research method for conducting a comprehensive, structured, and solid study that touches the most concerned research questions, and the in-depth analysis about the findings. The exceptional insights include (1) the first overview of the SOTA performance of different LVMs on time series benchmark datasets across tasks; (2) the identification of different challenges of time series classification (semantic-level task) and forecasting (numerical-level task) when using LVMs; and (3) the disruptive findings (and our analysis) that the SOTA LVM forecasters are limited in using encoders and long look-back windows, and their strong biases imposed by their imaging method.

We summarize the **key revisions to the draft** as follows. The details are described in our response to each review. Thank you for your attention and time.

1. Extended discussion about related works on multimodal methods for time series analysis (Appendix E lines 1716-1727, referred in main paper at lines 126-127)
2. Extended analysis for RQ8 (Appendix E lines 1728-1732, referred in main paper at line 428)
3. Extended analysis for RQ10 (Appendix E lines 1733-1746, referred in main paper at lines 474-475)
4. Visualization results for analyzing LVM forecasters' decoders (Appendix D.3, Figure 21)
5. Discussion of GAF and MVH's limitation in time series forecasting (TSF) task (Appendix B.2 lines 917-949, referred in main paper at line 263)
6. Detailed description and analysis of the properties, pros and cons of the 8 imaging methods (Appendix F.1, F.2, referred in main paper at lines 146-147)
7. Clarification of the scope of baseline models (main paper lines 202-204 in Section 4, lines 796-800 in Appendix A.2)
8. Addition of a time series foundation model (TSFM) baseline, LightGTS, for its better performance than some other TSFMs like Moirai and Chronos as reported in [1] (Appendix G, Table 28)
9. Evaluation of LVMs on datasets with weak periodicity (Appendix B.11, referred in main paper at lines 466-468)
10. Refined wordings about partial fine-tuning encoders and decoders of LVMs (main paper lines 424-425)
11. Highlighting future directions in Section 5 (main paper lines 477-485)

[1] Y. Wang, et al. LightGTS: A Lightweight General Time Series Forecasting Model. In ICML, 2025.

---

### Meta-Review · Area_Chair_GfBC · 2026-01-04

**Summary:**

This paper presents a large-scale empirical benchmark evaluating Large Vision Models (LVMs) for time series classification (TSC) and time series forecasting (TSF) by transforming time series into image representations. The study is extensive in scope, covering multiple LVMs, imaging methods, datasets, and ablations, and all reviewers agree that it is comprehensive and timely.

However, the reviews reveal a fundamental divergence in how the contribution is interpreted, with concerns clustering around three main issues:
	1.	Nature and depth of contribution
The paper is a pure benchmark study without proposing new models, objectives, or architectural mechanisms. While benchmark papers are valuable, multiple reviewers question whether the work delivers sufficiently deep mechanistic understanding or conceptual synthesis to justify acceptance at ICLR, as opposed to primarily cataloging empirical observations.
	2.	Causal vs. correlational claims
Several central claims—most notably that decoders matter more than encoders for forecasting, and that performance degradation for long look-back windows is intrinsic to LVMs—are supported largely by correlational evidence. Reviewers with higher confidence (notably Reviewer 1Fj9) repeatedly question whether these observations reflect intrinsic properties of LVMs or artifacts of architectural reuse and imaging choices.
	3.	Validity and generality of conclusions
The strongest forecasting results rely heavily on UVH/MVH imaging and on datasets with periodic structure. Reviewers express concern that many conclusions may not generalize to non-periodic, irregular, or exogenous-variable-rich settings, and that the inductive biases introduced by 2D imaging are not fully theorized. This raises questions about how broadly the findings should be interpreted.

A key factor informing my recommendation is that the most critical reviewer (1Fj9) expressed the lowest score but the highest confidence, while the higher scores came from reviewers who explicitly indicated moderate confidence and openness to rejection.

**Reviewer Concerns:**

Concerns that were partially or largely addressed in the rebuttal:
	•	The authors clearly and consistently clarified the scope of the work as a benchmark study, and convincingly argued that proposing new architectures or fixes would be out of scope.
	•	Additional experiments on non-periodic datasets (Exchange, Solar) strengthen the claim that LVMs exhibit a periodicity bias, rather than undermining it.
	•	The rebuttal substantially expands discussion on:
	•	imaging-method-specific biases,
	•	decoder vs. encoder roles,
	•	pixel utilization limits for long look-back windows,
	•	and alignment between imaging properties and forecasting performance.
	•	Comparisons with a strong time series foundation model (LightGTS) help contextualize the empirical findings and improve completeness.

Concerns that remain unresolved or only indirectly addressed:
	•	Causality and mechanistic grounding:
The central findings (decoder dominance, long-context failure) remain interpretive rather than causal. While the authors explain why certain follow-up experiments (e.g., decoder swapping, multi-scale images) are difficult or out of scope, the absence of such tests leaves ambiguity as to whether observed limitations are intrinsic to LVMs or to the chosen adaptation strategy.
	•	Inductive bias of imaging:
Reviewer 1Fj9 raises a deep concern about whether transforming time series into 2D images meaningfully preserves temporal inductive structure, or whether it disrupts temporal locality in ways that undermine conclusions. While the authors provide descriptive pros/cons of imaging methods, this concern remains largely conceptual and unresolved.
	•	Benchmark standard at ICLR:
Multiple reviewers (explicitly EKwA and 1Fj9) note that, unlike recent benchmark papers at ICLR, the work does not clearly articulate a unifying conceptual framework, taxonomy, or formal analysis that elevates it beyond a very thorough empirical survey.

These remaining issues are not minor; they go to the heart of whether the paper’s conclusions should be treated as definitive insights about LVMs or as preliminary observations conditioned on a specific experimental framing.

**Reviewer Scores:**

Based on the rebuttal quality and unresolved issues:
	•	Reviewer EKwA (score: 6, confidence: 3)
Likely to remain at 6. The rebuttal addresses many technical questions but does not change the reviewer’s view that the work lacks deeper mechanistic contribution.
	•	Reviewer fb57 (score: 6, confidence: 3)
Likely to remain at 6. The reviewer is generally positive but explicitly open to rejection and primarily values the work as a reference benchmark.
	•	Reviewer 1Fj9 (score: 4, confidence: 4)
Unlikely to change score. While the authors responded thoroughly, most responses argue that suggested validations are out of scope rather than demonstrating that the conclusions are robust. This reviewer’s high-confidence skepticism remains largely intact.

Overall, while the average score is above threshold, there is no clear convergence toward strong acceptance, and the most confident reviewer remains unconvinced on foundational grounds.

---

### Decision · Program_Chairs · 2026-01-26

Reject